# Robust Bayes–Assisted Conformal Prediction

**Kianoosh Ashouritaklimi** [* 1]   **Stefano Cortinovis** [* 1]   **François Caron** [1]

## Abstract

Bayes–assisted conformal prediction combines the strengths of Bayesian modelling with exact, distribution–free frequentist coverage guarantees. Although conformal validity is preserved even when the Bayesian working model (BWM) is misspecified, the size of the resulting prediction sets can degrade substantially when the prior is poorly aligned with the observed data. We address this limitation by introducing **RoBAS** (**Ro**bust **B**ayes-**A**ssisted **S**hrinkage): a Bayes–assisted framework for constructing robust nonconformity scores, with two instantiations: one induced by a heavy–tailed BWM, and a closed–form empirical Bayes shrinkage score. The resulting scores adapt to the quality of the working information encoded in the prior: when this information is reliable, they exploit it to produce efficient prediction sets; when it is weak or inaccurate, they revert to the Distance–To–Average (DTA) score, a robust non–informative baseline. We evaluate the proposed scores on tabular and image regression tasks where the training distribution may differ from the calibration and test distributions, while the calibration and test data themselves remain exchangeable. We find that they are competitive with widely used scores in the absence of such shift, while substantially reducing interval widths in shifted settings.

## 1. Introduction

Conformal prediction (CP) provides a powerful, distribution–free framework for constructing prediction sets with valid finite–sample frequentist coverage (Vovk et al., 2005). The efficiency of these sets, commonly measured by their expected size (e.g., expected width in regression with a scalar response), depends crucially on the choice of the nonconformity score, which quantifies how unusual a candidate outcome is relative to the observed data. Efficient prediction sets are essential for practical decision making, since overly conservative sets provide limited actionable information. Consequently, a substantial body of work has focused on designing more efficient nonconformity scores (Romano et al., 2019; Sadinle et al., 2019; Romano et al., 2020; Sesia & Romano, 2021; Chernozhukov et al., 2021; Guan, 2023; Seedat et al., 2023; Xie et al., 2024; Kiyani et al., 2024), such as by incorporating estimates of model uncertainty (Romano et al., 2019; Guan, 2023).

Bayes–assisted conformal prediction (Vovk et al., 2005; Wasserman, 2011; Fong & Holmes, 2021; Hoff, 2023; Bersson & Hoff, 2024; Deliu & Liseo, 2025) provides a principled framework for defining nonconformity scores that combine the strengths of Bayesian modelling with the exact frequentist coverage guarantees of conformal prediction. By using the negative posterior predictive density of a *Bayesian working model* (BWM) as the nonconformity score, this approach yields highly efficient prediction sets[1] when the prior information is well aligned with the data–generating process (Hoff, 2023). The ability to incorporate prior knowledge makes this framework particularly useful in settings with limited data, such as small area estimation (Bersson & Hoff, 2024; 2025), where the BWM can borrow strength from external or structural information that is difficult to exploit with purely data–driven methods.

The conformal validity of this approach holds *even* when the BWM is misspecified. However, its efficiency depends critically on the accuracy of the prior information in the model: if the data strongly disagrees with the prior, the resulting prediction sets can grow substantially (e.g., see Figure 1 and Bersson & Hoff, 2024, Figure 4(b)–(c)). In such cases, the benefits of Bayesian modelling are effectively lost.

To address this, we propose **RoBAS** (**Ro**bust **B**ayes-**A**ssisted **S**hrinkage): a Bayes–assisted framework for constructing nonconformity scores that exploit useful working information when it is reliable, while remaining robust when it is not. Unlike previous Bayes–assisted approaches that specify a BWM for the full data–generating process (Fong & Holmes,

---

[*]Equal contribution. [1]Department of Statistics, University of Oxford. Correspondence to: <ashouritaklimi@stats.ox.ac.uk>, <cortinovis@stats.ox.ac.uk>.

*Proceedings of the $43^{rd}$ International Conference on Machine Learning*, Seoul, South Korea. PMLR 306, 2026. Copyright 2026 by the author(s).

---

[1]More precisely, under suitable conditions, the prediction sets have smaller average Lebesgue measure than other prediction sets with the same coverage (Hoff, 2023).

2021; Hoff, 2023; Bersson & Hoff, 2024; 2025; Bhagwat et al., 2025), we place a BWM *only* on the residuals of an underlying predictive model, which itself may or may not be Bayesian. This avoids the requirement of specifying priors over high–dimensional parameter spaces, which can make the inclusion of accurate prior information more challenging (Wenzel et al., 2020; Fortuin et al., 2022; Fortuin, 2022).

The key idea is to design the residual–level BWM so that the resulting score adapts to the quality of the underlying predictor. When the predictor is accurate and its residuals are approximately centred at zero, the working model is well aligned with the data and yields efficient prediction sets. However, when the predictor is inaccurate and its residuals are centred far from zero, the score automatically reverts to the robust Distance–To–Average (DTA) nonconformity score, which we show is a natural choice in this setting. This behaviour ensures that the resulting prediction sets maintain a stable size even when the prior is inaccurate, resolving a key limitation of other Bayes–assisted methods whose efficiency can deteriorate under prior–data conflict.

We develop this idea through two complementary instantiations. First, we introduce a hierarchical residual BWM with heavy–tailed priors, which enjoys the desired robustness property and yields a Bayes–assisted score that provably approaches DTA under strong prior–data conflict. Second, we derive a computationally tractable empirical Bayes version that retains the same qualitative shrinkage behaviour while admitting a simple closed–form nonconformity score. This closed–form score is substantially cheaper to evaluate than Bayes–assisted approaches that require averaging over many MCMC samples in the parameter space (Fong & Holmes, 2021; Bhagwat et al., 2025) and yields provably interval–valued prediction sets. Finally, we provide a grid–free procedure for computing prediction intervals, which avoids the tuning challenges inherent in grid–based approaches.

Empirically, we evaluate the proposed scores on tabular and image regression tasks where the training distribution may differ from the calibration and test distributions, while the calibration and test data themselves remain exchangeable. We find that RoBAS is competitive with widely used nonconformity scores in the absence of such shift, while substantially reducing interval widths in shifted settings.

To summarise, our contributions are as follows:

- We introduce **RoBAS**, a Bayes–assisted framework for constructing robust residual-based nonconformity scores that exploit accurate working information while protecting against prior–data conflict.

- We derive two instantiations of RoBAS: a heavy–tailed residual BWM that induces the desired robustness mechanism, and a closed–form empirical Bayes shrinkage score that preserves this behaviour while yielding

provably interval–valued prediction sets.

- We evaluate the proposed scores on synthetic, tabular, and image regression tasks, showing that they outperform existing methods under training–to–calibration/test distribution shift, while remaining competitive in standard settings.

## 2. Background

In this section, we provide a brief overview of conformal prediction (§2.1–2.2) and Bayes–assisted conformal prediction (§2.3), with additional details given in Appendix E.2. Throughout, we denote random variables and their observed values using capital and lowercase letters, respectively.

### 2.1. Conformal Prediction

Let $(\mathbf{Z}_i)_{i=1}^{n+1}$ be an exchangeable sequence of random variables from some unknown distribution, where $\mathbf{Z}_i = (\mathbf{X}_i, Y_i) \in \mathcal{Z} = \mathcal{X} \times \mathcal{Y}$ is a covariate/response pair, with $\mathcal{X} \subseteq \mathbb{R}^d$ and $\mathcal{Y} \subseteq \mathbb{R}$, for $d \geq 1$. Let $\alpha \in [0, 1)$ be a user–defined error rate. Suppose we observe a dataset $\mathbf{z}_{1:n} = \{\mathbf{z}_i\}_{i=1}^n$ of the first $n$ covariate/response pairs, as well as the covariate $\mathbf{x}_{n+1}$ of a new observation. Conformal prediction (Vovk et al., 2005) allows us to obtain a prediction set $\mathcal{C}_\alpha(\mathbf{x}_{n+1}; \mathbf{z}_{1:n}) \subseteq \mathcal{Y}$ for the unknown response $Y_{n+1}$ that satisfies the $(1-\alpha)$–frequentist marginal coverage guarantee

$$\mathbb{P}(Y_{n+1} \in \mathcal{C}_\alpha(\mathbf{X}_{n+1}; \mathbf{Z}_{1:n})) \geq 1 - \alpha, \qquad (1)$$

where the probability is taken over the exchangeable random sequence $(\mathbf{Z}_i)_{i=1}^{n+1}$.

The prediction set $\mathcal{C}_\alpha$ is constructed via a *nonconformity score function* $\tilde{s} : \mathcal{Z} \times \mathcal{Z}^n \to \mathbb{R}$, which measures how unusual an observation $\mathbf{z}_{n+1} = (\mathbf{x}_{n+1}, y_{n+1})$ is relative to the observed data $\mathbf{z}_{1:n}$, with higher values indicating a more "unusual" observation. CP determines whether a candidate $y \in \mathcal{Y}$ belongs to $\mathcal{C}_\alpha(\mathbf{x}_{n+1}; \mathbf{z}_{1:n})$ by first computing the augmented nonconformity scores $\{s_i(y)\}_{i=1}^{n+1}$:

$$s_i(y) = \tilde{s}(\mathbf{z}_i, \mathbf{z}_{1:n,-i} \cup \{(\mathbf{x}_{n+1}, y)\}), \;\; i = 1, \ldots, n,$$
$$s_{n+1}(y) = \tilde{s}((\mathbf{x}_{n+1}, y), \mathbf{z}_{1:n}), \qquad (2)$$

where $\mathbf{z}_{1:n,-i} = \mathbf{z}_{1:n} \backslash \{\mathbf{z}_i\}$.

The candidate $y$ is then tested by comparing the rank of $s_{n+1}(y)$ among the augmented scores. The corresponding *conformal p-value* is

$$\rho(y) = \frac{1}{n+1} \sum_{i=1}^{n+1} \mathbb{I}\{s_i(y) \geq s_{n+1}(y)\}, \qquad (3)$$

and $y$ is accepted if $\rho(y) > \alpha$. The full prediction set is

$$\mathcal{C}_\alpha(\mathbf{x}_{n+1}; \mathbf{z}_{1:n}) = \{y \in \mathcal{Y} \mid \rho(y) > \alpha\}. \qquad (4)$$

When evaluated at the true response $Y_{n+1}$, the augmented nonconformity scores $(S_i(Y_{n+1}))_{i=1}^{n+1}$ are exchangeable, providing the set in (4) with the desired guarantee in (1).

## 2.2. Residual–Based Nonconformity Score Functions

We consider here the case where we have access to a fixed predictive model $f : \mathcal{X} \to \mathcal{Y}$, typically trained on data independent of the calibration/test set. In this setting, the nonconformity score function is often based on the residuals $r_i = y_i - f(\mathbf{x}_i)$, $i = 1, \dots, n+1$, through a residual–based nonconformity score function $s : \mathbb{R} \times \mathbb{R}^n \to \mathbb{R}$, such that

$$\tilde{s}(\mathbf{z}_{n+1}, \mathbf{z}_{1:n}) = s(r_{n+1}, \mathbf{r}_{1:n}). \qquad (5)$$

The nonconformity scores (2) then reduce to

$$s_i(y) = s(r_i, \mathbf{r}_{1:n,-i} \cup \{y - f(\mathbf{x}_{n+1})\}), \;\; i = 1, \dots, n,$$
$$s_{n+1}(y) = s(y - f(\mathbf{x}_{n+1}), \mathbf{r}_{1:n}), \qquad (6)$$

where $\mathbf{r}_{1:n,-i} = \mathbf{r}_{1:n} \backslash \{r_i\}$.

Two common residual–based nonconformity scores are the *Distance–To–Origin* (DTO) and *Distance–To–Average* (DTA) scores:

$$s^{\mathrm{DTO}}(r_{n+1}, \mathbf{r}_{1:n}) = |r_{n+1}|, \qquad (7)$$
$$s^{\mathrm{DTA}}(r_{n+1}, \mathbf{r}_{1:n}) = |r_{n+1} - \bar{r}_n|, \qquad (8)$$

where $\bar{r}_n = \frac{1}{n} \sum_{i=1}^{n} r_i$. The DTO score treats zero–centred residuals as the reference, effectively *trusting* that $f$ is (approximately) unbiased on the calibration/test distribution. When this assumption is correct, especially with small calibration sets, DTO can be highly efficient because it avoids the extra variability introduced by estimating a centering term. However, DTO is sensitive to systematic bias: if the residuals are shifted away from zero, $|r_{n+1}|$ is uniformly inflated and the resulting prediction sets can become unnecessarily large. By contrast, DTA recentres the residuals by $\bar{r}_n$, making it translation–invariant and therefore more robust to mean shifts. This robustness comes at the cost of estimating $\bar{r}_n$, which can be noisy for small $n$ and can widen prediction sets even when the true residual mean is close to zero. This trade–off suggests that neither DTO nor DTA is uniformly preferable, motivating an adaptive score that interpolates between them by shrinking towards DTO when residuals are plausibly centred near zero and reverting toward DTA when the data indicate a substantial mean shift.

## 2.3. Bayes–Assisted Conformal Prediction

While the validity of conformal prediction holds for any nonconformity score function, its efficiency – the expected size of the resulting prediction sets – depends critically on this choice. *Bayes–assisted conformal prediction* (Vovk et al., 2005; Wasserman, 2011; Fong & Holmes, 2021;

Hoff, 2023; Bersson & Hoff, 2024; Deliu & Liseo, 2025) defines the score using a Bayesian working model (BWM). In the standard formulation, a BWM is specified for the conditional data–generating process and the nonconformity score is taken to be the negative posterior predictive density:

$$\tilde{s}(\mathbf{z}_{n+1}, \mathbf{z}_{1:n}) = -p(y_{n+1} \mid \mathbf{x}_{n+1}, \mathbf{z}_{1:n}) \qquad (9)$$
$$= -\int p(y_{n+1} \mid \mathbf{x}_{n+1}, \theta) p(\theta \mid \mathbf{z}_{1:n}) \, d\theta,$$

where $\theta$ denotes the parameters of the BWM. Previous approaches (Fong & Holmes, 2021; Bhagwat et al., 2025) typically require MCMC sampling to compute the posterior predictive, although certain BWMs admit closed–form solutions (Bersson & Hoff, 2024; 2025). To reduce the cost of repeated leave–one–out posterior predictive evaluations for each candidate $y$, the *add–one–in* (AOI) importance sampling trick from Fong & Holmes (2021) is normally used.

The benefits of the Bayes–assisted approach are twofold. First, the use of a BWM allows prior or side information to be incorporated into the nonconformity score, which is especially useful in small–data regimes (Bersson & Hoff, 2024). Second, it can be shown (Hoff, 2023) that, when the BWM is well aligned with the data–generating process, the posterior-predictive score yields an efficient, *Bayes–optimal* conformal procedure:

**Theorem 2.1** (Hoff, 2023, Thm. 4.1, Bayes–optimality; informal). *Let $P = \{P_\theta \mid \theta \in \Theta\}$ be a family of conditional probability distributions on $Y \mid \mathbf{X}$, and let $\pi$ be a prior on $\Theta$. Given a conformal procedure $\mathcal{C}_\alpha$ for a specified error rate $\alpha$, let $\mathcal{R}_\pi(\mathcal{C}_\alpha)$ be the associated Bayes risk, defined as*

$$\mathcal{R}_\pi(\mathcal{C}_\alpha) = \int_\Theta \mathcal{R}_\theta(\mathcal{C}_\alpha) \, \pi(d\theta) = \int_\Theta \mathbb{E}_{P_\theta}[\lambda\{\mathcal{C}_\alpha(\mathbf{X})\}] \, \pi(d\theta),$$

*where $\lambda$ is a volume measure on $\mathcal{Y}$. Then, under mild regularity conditions, the conformal procedure defined by the score (9) minimises Bayes risk among conformal procedures with equal or greater coverage.*

Theorem 2.1 implies that, when the BWM prior is accurate and correctly assigns probability mass to parameters describing the observed data, the CP procedure induced by the score (9) yields lower average set size than other scores. However, as we will see later, efficiency can deteriorate substantially when the prior is inaccurate, motivating the robust Bayes–assisted construction in the next section.

## 3. Robust Bayes–Assisted Conformal Prediction

In this section, we develop a conformal prediction approach that (i) retains the efficiency benefits of Bayes–assisted procedures, but (ii) remains robust when the prior information

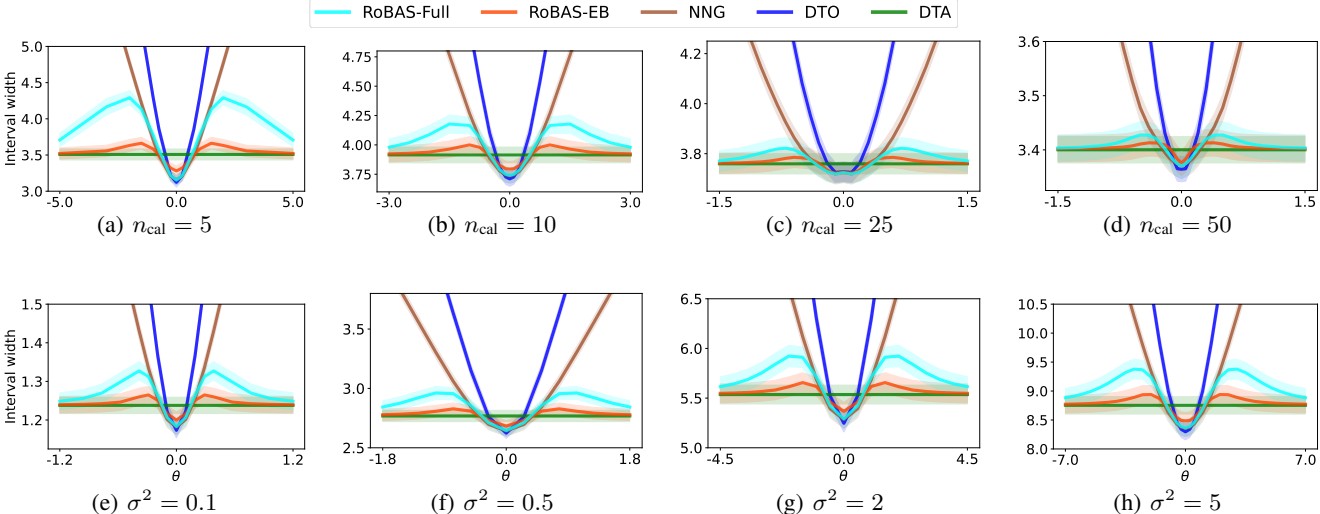

*Figure 1.* Interval width results for different nonconformity scores for data distributed as $\mathcal{N}(\theta, \sigma^2)$. **Top Row:** Results across different calibration sizes, $n_{\text{cal}}$, with $\sigma^2 = 1$. **Bottom Row:** Results across different noise levels, $\sigma^2$, with a fixed calibration size $n_{\text{cal}} = 10$. To make the differences between the different methods clear, the y–axis has been cut. Results show the mean $\pm$ standard err. over 300 trials.

is inaccurate while (iii) maintaining computation tractability. Although we primarily instantiate our approach for residual–based conformal prediction (§2.2), our robust score construction applies more generally (§2.3). Our core message is that, under suitable residual–level BWMs, Bayes–assisted scores induce prediction sets that interpolate between DTO when residuals are centred near zero, and DTA when the residual mean is far from zero. We refer to this overall framework as **RoBAS** (**Ro**bust **B**ayes–**A**ssisted **S**hrinkage).

We begin by formally motivating and describing our approach (§3.1). We then introduce a robust hierarchical working model based on heavy–tailed priors (**RoBAS–Full**, §3.2), which forms the basis for deriving a more computationally efficient empirical Bayes score (**RoBAS–EB**, §3.3). Finally, we describe a grid–free method for computing prediction intervals via bracketed search and root–finding (§3.4). We defer proofs and additional details to Appendices D and E, respectively.

### 3.1. Residual–Based, Bayes–Assisted Conformal Prediction

Let $f : \mathcal{X} \to \mathcal{Y}$ be a fixed predictive model, typically trained on a dataset separate from the calibration/test set, and define the residuals $r_i$ as in §2.2. We consider a BWM on the *residuals* of $f$, as opposed to the conditional data–generating process used in previous works (Burnaev & Vovk, 2014; Fong & Holmes, 2021; Bersson & Hoff, 2024; 2025; Bhagwat et al., 2025). Formally, we define our residual–based nonconformity score function through the corresponding

posterior predictive:

$$s\left(r_{n+1}, \mathbf{r}_{1:n}\right) = -p\left(r_{n+1} \mid \mathbf{r}_{1:n}\right)$$
$$= -\int p(r_{n+1} \mid \phi) p\left(\phi \mid \mathbf{r}_{1:n}\right) \, d\phi, \quad (10)$$

where $\phi$ denotes the parameters of the BWM. The associated nonconformity score function and nonconformity scores are given by (5)–(6). As $f$ is fixed, exchangeability of $(\mathbf{Z}_i)_{i=1}^{n+1}$ implies exchangeability of the residuals $(R_i)_{i=1}^{n+1}$, and the usual finite–sample marginal coverage guarantee (1) holds.

Residual–level BWMs are attractive for two main reasons. First, they induce a substantially more scalable conformal procedure: as the model is defined for scalar residuals that are independent of $\mathbf{X}$, $\phi$ is low–dimensional (e.g., scalar mean and variance), making posterior evaluation cheap and sometimes available in closed form. Second, residual BWMs make it considerably easier to incorporate *domain knowledge*. Informative priors for a full conditional model $Y \mid \mathbf{X}, \theta$ typically require beliefs over high–dimensional objects (e.g., neural network weights), which are rarely available (Wenzel et al., 2020; Fortuin et al., 2022; Fortuin, 2022). By contrast, a residual–level BWM only requires prior beliefs about low–dimensional scalar quantities.

Throughout this work, we use zero as the working prior centre for the residual mean, encoding the belief that $f$ is approximately unbiased on the calibration/test distribution. The construction naturally extends to any fixed nonzero centre $\mu_0$ by applying the same method to the shifted residuals $r_i - \mu_0$, or equivalently by using the shifted predictor $f'(\mathbf{x}_i) = f(\mathbf{x}_i) + \mu_0$.

A natural conjugate residual BWM is the zero–centred

Normal–Normal–Gamma model

$$R \mid \theta \sim \mathcal{N}\left(\theta, \sigma^2\right),$$
$$\theta \sim \mathcal{N}\left(0, \tau^2\sigma^2\right),$$
$$1/\sigma^2 \sim \text{Gamma}(a/2, b/2), \tag{11}$$

where $\tau^2, a, b$ are fixed hyperparameters. BWM (11) represents a zero–centred residual version of the model used by Bersson & Hoff (2024) and admits both closed–form nonconformity scores and prediction intervals (see Appendix E.4 for further details), making it a computationally appealing choice. When the zero–centre prior belief is correct, the resulting Bayes–assisted prediction sets can be highly efficient. However, when $f$ is highly biased, such a BWM leads to prediction sets that, while satisfying the coverage guarantee, can be arbitrarily large (see Figure 1). This motivates a BWM that remains informative near zero but enjoys greater robustness to prior–data conflict.

### 3.2. Robust Bayesian Working Model

We propose the following zero–centred hierarchical BWM for the residuals (Gelman, 2006; Carvalho et al., 2010):

$$R \mid \theta \sim \mathcal{N}(\theta, \sigma^2),$$
$$\theta \mid \tau^2 \sim \mathcal{N}(0, \gamma\tau^2),$$
$$\tau^2 \sim g_{\tau^2}(\tau^2), \tag{12}$$

where $\gamma, \sigma > 0$ are fixed hyperparameters and $g_{\tau^2}$ is a heavy–tailed density. Specifically, we assume $g_{\tau^2}$ is regularly varying at infinity: $g_{\tau^2}(\tau^2) \sim C(\tau^2)^{-\delta}$ as $\tau^2 \to \infty$, for some $C > 0$ and $\delta > 1$.

When $f$ is accurate and residuals are centred near zero (so $\bar{r}_n \approx 0$), the zero–centred prior is well aligned with the data and we expect efficient prediction sets similar to DTO. On the other hand, when $f$ is inaccurate – for instance under distribution shift between training and calibration/test data – the sample mean $\bar{r}_n$ can deviate substantially from zero. In this misspecified regime, the behaviour of the nonconformity score induced by the heavy–tailed BWM (12) differs greatly from other seemingly natural BWM choices, such as (11). Intuitively, the heavy–tailed prior on $\tau^2$ enables the posterior to place non–negligible mass on large prior variances for $\theta$, effectively down–weighting the influence of the zero–centred prior when the data indicate a large mean shift. As a result, the induced Bayes–assisted nonconformity score becomes asymptotically equivalent to the DTA score up to a monotone transformation.

**Theorem 3.1** (Asymptotic Robustness of Heavy–Tailed BWM). *Fix $n \geq 1$. Let $(\mathbf{r}_{1:n}^{(m)})_{m \geq 1}$, with $\mathbf{r}_{1:n}^{(m)} \in \mathbb{R}^n$, be a sequence of residuals such that $|\bar{r}_n^{(m)}| \to \infty$ as $m \to \infty$. Under BWM (12), the score function (10) satisfies, for every fixed $\Delta \in \mathbb{R}$,*

$$s(\bar{r}_n^{(m)} + \Delta, \mathbf{r}_{1:n}^{(m)}) = -p(\bar{r}_n^{(m)} + \Delta \mid \mathbf{r}_{1:n}^{(m)}) \to h_n(|\Delta|)$$

*as $m \to \infty$, where $h_n : [0, \infty) \to \mathbb{R}$ is the strictly increasing function*

$$h_n(u) = -\frac{1}{\sqrt{2\pi\sigma^2(1+1/n)}} \exp\left(-\frac{u^2}{2\sigma^2(1+1/n)}\right).$$

Theorem 3.1 states that, when $|\bar{r}_n|$ is large, $s(r_{n+1}, \mathbf{r}_{1:n}) \simeq h_n(|r_{n+1} - \bar{r}_n|)$ for $r_{n+1} = \bar{r}_n + \Delta$. Since $h_n$ is strictly monotone increasing, this implies that the Bayes–assisted score function behaves similarly to the DTA score function. In practice, this behaviour is highly desirable: when the underlying predictor is poor and the residual mean is far from the working prior mean of zero, it is preferable to rely more heavily on the calibration data rather than on unrealistic prior information.

Theorem 3.1 also implicitly unveils an insightful connection between the DTA score and Bayes–assisted conformal prediction. In particular, as the next result shows, the DTA score is equivalent to the Bayes–assisted nonconformity score corresponding to a non–informative prior on the mean.

**Proposition 3.2.** *The DTA nonconformity score is equivalent to the Bayes–assisted nonconformity score for the following BWM:*

$$R \mid \theta \sim \mathcal{N}(\theta, \sigma^2),$$
$$\theta \sim \pi(\theta) \propto 1$$

*where $\sigma > 0$ can take any arbitrary, fixed value.*

Together, Theorem 3.1 and Proposition 3.2 formalise our robustness goal: our heavy–tailed BWM (12) induces efficient Bayes–assisted prediction sets when residuals are near zero, but automatically reverts to stable, data–driven DTA prediction sets when the residual mean becomes large.

In §4, we instantiate BWM (12) by choosing $g_{\tau^2}$ so that $\tau \sim C^+(0, 1)$, which induces a horseshoe prior on $\theta$ (Carvalho et al., 2010), and setting $\gamma = \sigma^2/n$, in the spirit of Piironen & Vehtari (2017). The resulting model induces the Bayes–assisted nonconformity score

$$s(r_{n+1}, \mathbf{r}_{1:n}) = \exp\left(-\frac{ns_n^2}{2\sigma^2}\right) {}_1F_1\left(1; \frac{3}{2}; -\frac{n\bar{r}_n^2}{2\sigma^2}\right) \tag{13}$$

via Lemma D.1, where $s_n^2 = \frac{1}{n}\sum_{i=1}^n (r_i - \bar{r}_n)^2$ and ${}_1F_1$ is the confluent hypergeometric function of the first kind; see Appendix E.3 for a detailed derivation. In all experiments, we use score (13), which we refer to as **RoBAS–Full**, with $\sigma^2$ estimated from the augmented residual vector $\mathbf{r}_{1:n+1}$; an extension of Theorem 3.1 to this plug–in setting is provided in Appendix D.5.

### 3.3. Empirical Bayes Nonconformity Score

While (12) yields the desired robustness behaviour, computing the corresponding Bayes–assisted score (10) may

require integrating over $\tau^2$ or evaluating special functions, which can be computationally expensive when scores must be evaluated repeatedly to construct prediction sets through (4); see Appendix E.3. To avoid this difficulty while retaining the same qualitative shrinkage behaviour, we also consider an alternative empirical Bayes (EB, Efron & Morris, 1973) construction.

The starting point is a conjugate Normal–Normal approximation to BWM (12), obtained by replacing the variance–mixture prior on $\theta$ with a single prior variance parameter. Specifically, we consider the BWM

$$
\begin{aligned}
R \mid \theta &\sim \mathcal{N}(\theta, \sigma^2), \\
\theta \mid \upsilon^2 &\sim \mathcal{N}(0, \upsilon^2).
\end{aligned}
\tag{14}
$$

We first record the fixed–hyperparameter Bayes–assisted score induced by BWM (14).

**Proposition 3.3.** *Consider BWM* (14) *with fixed* $\sigma^2, \upsilon^2 > 0$. *Then, the induced Bayes–assisted score is a strictly monotone transformation of the score*

$$
s(r_{n+1}, \mathbf{r}_{1:n}) = \left| r_{n+1} - a(\sigma^2, \upsilon^2)\, \bar{r}_n \right|,
\tag{15}
$$

*where* $\bar{r}_n = \frac{1}{n} \sum_{i=1}^{n} r_i$ *and*

$$
a(\sigma^2, \upsilon^2) = \frac{\upsilon^2}{\upsilon^2 + \sigma^2/n}.
\tag{16}
$$

That is, for fixed variance parameters, BWM (14) induces an absolute–deviation score centred at the posterior mean $a(\sigma^2, \upsilon^2)\, \bar{r}_n$. In practice, rather than fixing $\sigma^2$ and $\upsilon^2$ a priori, we choose them adaptively via EB and plug the resulting estimates into the fixed–hyperparameter score. Specifically, we define the nonconformity score as

$$
s(r_{n+1}, \mathbf{r}_{1:n}) = \left| r_{n+1} - \widehat{a}(\mathbf{r}_{1:n})\, \bar{r}_n \right|,
\tag{17}
$$

where $\widehat{a}(\mathbf{r}_{1:n}) = a(\widehat{\sigma}^2(\mathbf{r}_{1:n}), \widehat{\upsilon}^2(\mathbf{r}_{1:n}))$ for

$$
\widehat{\sigma}^2(\mathbf{r}_{1:n}) = \frac{1}{n} \sum_{i=1}^{n} (r_i - \bar{r}_n)^2,
\tag{18}
$$

$$
\widehat{\upsilon}^2(\mathbf{r}_{1:n}) = \max\left( \bar{r}_n^2 - \frac{\widehat{\sigma}^2(\mathbf{r}_{1:n})}{n},\, 0 \right),
\tag{19}
$$

and with the convention that $\widehat{a}(\mathbf{r}_{1:n}) = 0$ when its denominator is zero. We refer to this score as **RoBAS–EB** and provide a derivation of the EB estimates (18)–(19) in Appendix D.3.

The $\widehat{a}(\mathbf{r}_{1:n})\bar{r}_n$ term in (17) is a shrinkage estimator of the mean (Efron & Morris, 1975; Morris, 1983; Copas, 1983). Similarly to the score induced by BWM (12), score (17) reverts to DTA when $|\bar{r}_n|$ is large, provided that $\widehat{\sigma}^2(\mathbf{r}_{1:n})$ does not grow as fast as $|\bar{r}_n|$. This is summarised in the following result, giving an EB analogue of Theorem 3.1.

**Proposition 3.4** (Asymptotic Robustness of EB Score). *Fix* $n \geq 1$. *Let* $(\mathbf{r}_{1:n}^{(m)})_{m \geq 1}$, *with* $\mathbf{r}_{1:n}^{(m)} \in \mathbb{R}^n$, *be a sequence of residuals such that* $|\bar{r}_n^{(m)}| \to \infty$ *and* $\widehat{\sigma}^2(\mathbf{r}_{1:n}^{(m)})/|\bar{r}_n^{(m)}| \to 0$ *as* $m \to \infty$. *Then, the score* (17) *satisfies, for every fixed* $\Delta \in \mathbb{R}$,

$$
s(\bar{r}_n^{(m)} + \Delta, \mathbf{r}_{1:n}^{(m)}) \to |\Delta| \quad \text{as } m \to \infty.
$$

On the other hand, as $\bar{r}_n \to 0$, the mean estimate shrinks to zero and we recover DTO. Thus, the EB approximation preserves the desirable properties of BWM (12) while providing an interpretable, closed–form expression for the nonconformity score. Moreover, this form implies that the "bias–corrected" prediction $y = f(\mathbf{x}_{n+1}) + \widehat{a}(\mathbf{r}_{1:n})\bar{r}_n$ gives $s_{n+1}(y) = 0$ and, as a result, it is always contained in the resulting prediction set.

## 3.4. Computation of Prediction Intervals

Recall that the exact conformal prediction set (4) is given by

$$
\mathcal{C}_\alpha(\mathbf{x}_{n+1}; \mathbf{z}_{1:n}) = \{y \in \mathcal{Y} \mid \rho(y) > \alpha\},
$$

where $\rho(y)$ is the conformal $p$-value (3) for candidate label $y$. Except for specific nonconformity scores that admit closed–form expressions for their prediction sets, standard approaches (Fong & Holmes, 2021; Bhagwat et al., 2025) approximate this set with a *prediction interval* by evaluating the $p$-values over a fine grid of candidate labels and returning the boundaries of the grid–based acceptance set. This, however, requires careful selection of the grid range and resolution to avoid under/over coverage, which induces a trade–off between computational cost and discretisation error.

Motivated by this issue, we instead compute a prediction interval by using a grid–free search for the endpoints of the acceptance set, rather than evaluating the conformal $p$-values over a fixed grid. We first find an accepted centre $y^\star$ by approximately maximising $\rho(y)$ using bracketed search (Brent, 1973; Le, 1985). Then, starting from $y^\star$, we search outwards for the transition between accepted and rejected candidates using a bracketed root–search routine applied to the acceptance criterion $\rho(y) > \alpha$. The full procedure is summarised in Algorithm 1; see Appendix E.5 for additional details on its computational complexity.

For a generic nonconformity score, the exact conformal set (4) need not be an interval. In that case, Algorithm 1 should be interpreted as returning an interval approximation to the exact conformal set. When the exact conformal set is an interval, however, the procedure recovers its endpoints up to numerical tolerance. The following result shows that this favourable case holds for the RoBAS–EB score (17).

**Theorem 3.5** (Interval Property of RoBAS–EB). *Assume* $n \geq 4$ *and consider the nonconformity score* (17). *Then, for any* $\alpha \in [0, 1)$, *the corresponding conformal prediction set* $\mathcal{C}_\alpha$ *in* (4) *is an interval, up to intersection with* $\mathcal{Y}$.

**Algorithm 1** Grid–free computation of prediction intervals

**Require:** Calibration data $\mathbf{z}_{1:n} = \{\mathbf{z}_i\}_{i=1}^n$, new input $\mathbf{x}_{n+1}$, score function $s(\cdot, \cdot)$, predictor $f$, error level $\alpha \in [0, 1)$.
**Ensure:** Interval approximation $[l, u]$ to $\mathcal{C}_\alpha(\mathbf{x}_{n+1}; \mathbf{z}_{1:n})$
1: Define conformal $p$–value $\rho(y)$ from augmented scores $\{s_i(y)\}_{i=1}^n \cup \{s_{n+1}(y)\}$ as in (3) and (6)
2: Find $y^\star \leftarrow \arg\max_y \rho(y)$ via bracketed search
3: Approximate $u \leftarrow \inf\{y \geq y^\star \mid \rho(y) \leq \alpha\}$ via bracketed root–search
4: Approximate $l \leftarrow \sup\{y \leq y^\star \mid \rho(y) \leq \alpha\}$ via bracketed root–search
5: **return** $[l, u]$

## 4. Experiments

In this section, we demonstrate the benefits of our approach in a synthetic setting and on several real–world datasets. Given a fixed $f$ trained on a proper training set, we apply full conformal prediction to the calibration residuals. The calibration and test sets are assumed exchangeable, which guarantees coverage validity, but we consider the scenario where there is a shift between the training and calibration/test data that impacts the performance of $f$. We focus on the small–calibration size setting like in Hoff (2023) and ablate with standard calibration sizes in Appendix F.2. We summarise the key details of our setup below and provide full details in Appendix C. All experiments were run for 300 trials. Code for reproducing the experiments is available at https://github.com/kiaashour/RoBAS.

### 4.1. Synthetic Experiments

We first demonstrate the benefits and potential limitations of our approach in a synthetic setting.

**Data:** We generate calibration data $\epsilon_i \sim \mathcal{N}(\theta, \sigma^2)$ with a test set of size $n_{\text{test}} = 1000$. We compare different nonconformity scores for different calibration sizes $n_{\text{cal}} \in \{5, 10, 25, 50\}$, and different values of $\theta$ and $\sigma^2$. We use a nominal error rate of $\alpha = 0.1$; when $n_{\text{cal}}$ is too small to satisfy the nominal error rate, we set $\alpha$ to the smallest possible error rate of $1/(n_{\text{cal}} + 1)$.

This simpler setting allows us to simulate the effect of varying model quality on the width of the prediction intervals. Indeed, our data $\epsilon_i$ can be considered as the residuals of some model $f$ where the mean of these residuals diverges from zero as the model's performance worsens.

**Nonconformity scores:** We compare the two variants of **RoBAS** (**–Full** and **–EB**) with **NNG**, **DTA** and **DTO**, where **NNG** refers to the nonconformity score corresponding to the **N**ormal–**N**ormal–**G**amma BWM in (11). For **NNG**, we set $\tau^2 = 1/n_{\text{cal}}$, which equally weights the influence of the

prior and data and avoids the need for extra held–out data (see Appendix C.1 for additional details).

**Results:** From Figure 1, we see that when $\theta \approx 0$ (i.e., $f$ is accurate), **DTO**, **NNG** and **RoBAS** perform similarly, providing the tightest prediction intervals. **DTO** benefits from its inherent bias in assuming the residuals are centred at 0, while **RoBAS** and **NNG**'s BWM leverage strong prior information concentrated near zero. In contrast, **DTA** performs the worst, a consequence of the mean estimate's high variance at small sample sizes. Moreover, **RoBAS–Full** slightly outperforms **RoBAS–EB**, which is consistent with the full hierarchical model's horseshoe prior inducing stronger shrinkage towards the prior mean of zero when the calibration residuals are themselves close to zero.

When the residuals are far from zero (i.e., the model is inaccurate), **DTO** and **NNG** show significantly increased widths, whereas **RoBAS** remains robust, performing similarly to **DTA** by adaptively reverting to its nonconformity score. We observe that **RoBAS–Full** transitions more gradually to the **DTA**–like regime. This is expected as **RoBAS–EB** point–estimates the prior variance of $\theta$ as $\hat{v}^2 = \max(\bar{r}_n^2 - \hat{\sigma}^2/n, 0)$, which grows quadratically in $|\bar{r}_n|$ and drives the shrinkage factor $\hat{a}$ to one as soon as $|\bar{r}_n|$ exceeds the noise scale, whereas **RoBAS–Full** instead marginalises this variance under the horseshoe prior, whose spike at zero sustains **DTO**–like shrinkage until $|\bar{r}_n|$ is large enough for the heavy tail to drive reversion to **DTA** (Theorem 3.1).

Moreover, we also observe that the gains at $\theta = 0$ of both **RoBAS** variants diminish as $n_{\text{cal}}$ increases and $\sigma^2$ decreases, suggesting that our approach is most beneficial in high–noise settings or where limited calibration data is available.

### 4.2. Real Datasets

We now demonstrate the benefits of our approach on real–world tabular and image regression datasets. We consider the setting where there is a distribution shift between the data used to train $f$ and the calibration/test set. This provides a natural testbed for our approach, as we expect our predictor to be accurate in the absence of distribution shift, but for performance to degrade when shifts occur.

**Tabular datasets:** For tabular datasets we consider the setting where there exists covariate shift between the data used to train our predictor $f$ and our calibration/test data. This can occur in many settings when, for example, we have a black–box predictor that has been pretrained on a broad, general dataset, but we wish to calibrate and deploy it for a particular sub–population.

We consider standard UCI datasets (Kelly et al.) used in previous works (Romano et al., 2019; Tibshirani et al., 2019; Sesia & Romano, 2021; Zaffran et al., 2023; Plassier et al., 2025): Facebook comment volume (facebook_1),

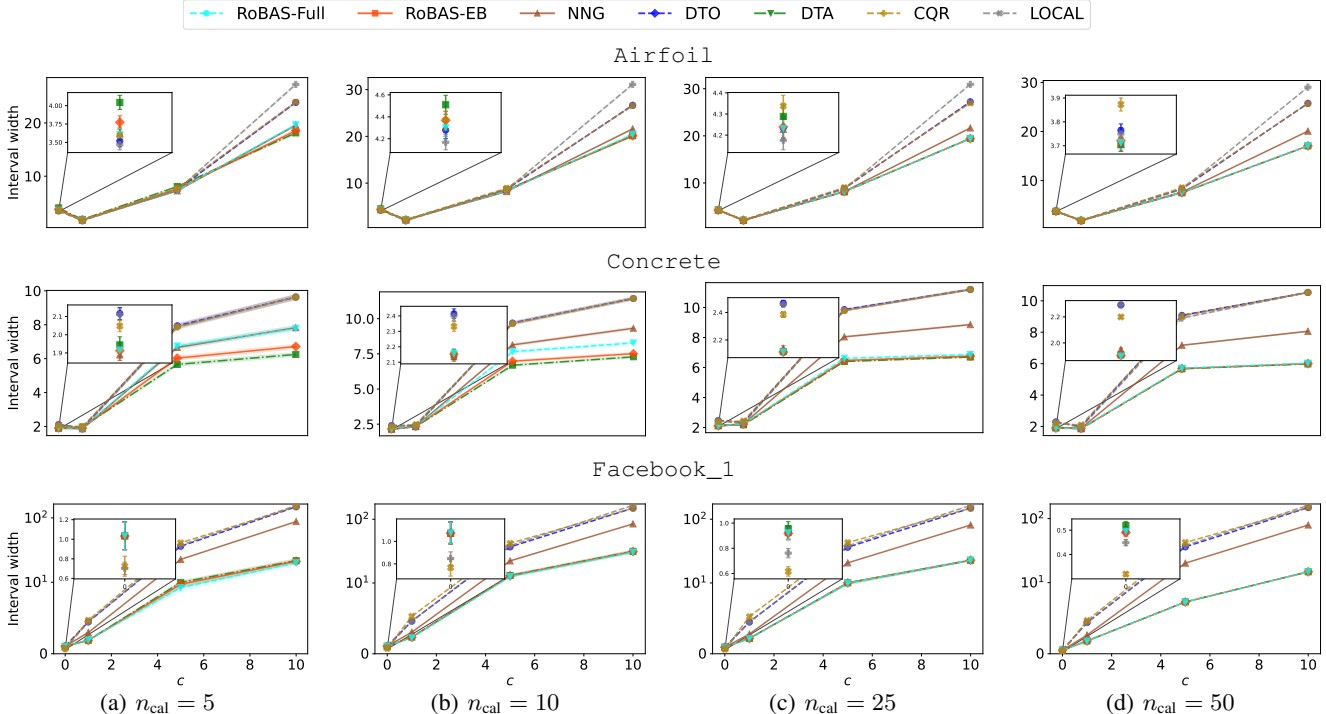

*Figure 2.* Interval width results for different nonconformity scores for different datasets with different levels of covariate shift. Results for different calibration sizes, $n_{\text{cal}}$ are shown. Each of the plots also zooms in on the results for $c = 0$ to better illustrate the differences under lower levels of covariate shift. The `Facebook_1` results are displayed on a symlog scale to more clearly highlight the differences between the methods. Results show mean $\pm$ standard err. over 300 trials.

airfoil self–noise (`airfoil`), concrete compressive strength (`concrete`). We focus on small–calibration sets of sizes $n_{\text{cal}} \in \{5, 10, 25, 50\}$ and a nominal error rate of $0.1$; when our calibration size is too small to satisfy this we use the smallest possible error rate of $1/(n_{\text{cal}} + 1)$.

To simulate covariate shift for these datasets, we follow Tibshirani et al. (2019) and sample training points with replacement, using probabilities proportional to:

$$w(\mathbf{x}) = \exp(\mathbf{x}^T \boldsymbol{\beta}), \ \ \boldsymbol{\beta} = \underbrace{(-c, 0, \ldots, 0, c)}_{d},$$

where $c \in \mathbb{R}^+$, $\mathbf{x}, \boldsymbol{\beta} \in \mathbb{R}^d$, $d$ is the number of features in the dataset. We consider varying levels of covariate shift by varying $c$, where we use $c = 0$ to denote no covariate shift.

Following Romano et al. (2019), we use $20\%$ of the data for testing, and $80\%$ for training. The training set is split equally into a calibration set and a proper training set, where the proper training set is sampled as above.

**Image datasets:** For our image regression datasets, we consider the setting where our calibration/test data are in/out–of–distribution relative to the data used to train $f$. We use the `UTKFaces` (Zhang et al., 2017) and `VentricularVolume` datasets from Gustafsson et al. (2023). For the `UTKFaces` dataset, we subset the data so

that the in–distribution data includes ages between 18–50, and the out–of–distribution data includes ages larger than 50; for the `VentricularVolume` dataset we use the in–distribution and out–of–distribution subsets provided. We report only the results for `UTKFaces` here and defer the results of `VentricularVolume` to Appendix F.1.

Again, we focus on small–calibration sets of sizes $n_{\text{cal}} \in \{5, 10, 25, 50\}$. We set our error rate in the same way as before and use the test and training sets provided. We partition the provided test set into a calibration set and a final test set using an 80/20 split; this ensures that the calibration and test data are identically distributed.

**Nonconformity scores:** We compare both variants of **RoBAS** with **NNG**, **DTA**, **DTO**, **CQR** (Romano et al., 2019), **LOCAL** (Guan, 2023), **CB** (Fong & Holmes, 2021) and **CBMA** (Bhagwat et al., 2025). We emphasise that while **CB** and **CBMA** are also Bayes–assisted approaches, they are not directly comparable to ours, as they conformalise the posterior predictive of a fully Bayesian model for $Y|\mathbf{X}$; in contrast, our method treats $f$ as fixed, places a BWM *only* on its residuals, and also accommodates a much broader class of predictors. We thus report the results for all other baselines here and defer the comparisons with **CB** and **CBMA** to Appendix F. For **NNG**, we set $\tau^2 = 1/n_{\text{cal}}$ as before. We use the default hyperparameters from the

*Table 1.* Interval widths for different nonconformity scores at different calibration sizes, $n_{cal}$, for the `UTKFaces` dataset. Results show the mean $\pm$ standard err. over 300 trials on the in–distribution and out–of–distribution sets of the dataset.

| Scores | IN | | | | OUT | | | |
|---|---|---|---|---|---|---|---|---|
| | $n_{cal} = 5$ | $n_{cal} = 10$ | $n_{cal} = 25$ | $n_{cal} = 50$ | $n_{cal} = 5$ | $n_{cal} = 10$ | $n_{cal} = 25$ | $n_{cal} = 50$ |
| **DTO** | $2.834 \pm 0.052$ | $3.388 \pm 0.045$ | $3.387 \pm 0.035$ | $3.141 \pm 0.023$ | $7.554 \pm 0.028$ | $7.861 \pm 0.023$ | $7.870 \pm 0.015$ | $7.717 \pm 0.009$ |
| **DTA** | $3.282 \pm 0.070$ | $3.629 \pm 0.054$ | $3.464 \pm 0.036$ | $3.147 \pm 0.022$ | $2.674 \pm 0.054$ | $2.919 \pm 0.040$ | $2.726 \pm 0.024$ | $2.463 \pm 0.015$ |
| **NNG** | $2.919 \pm 0.058$ | $3.450 \pm 0.049$ | $3.402 \pm 0.035$ | $3.137 \pm 0.023$ | $4.998 \pm 0.028$ | $5.022 \pm 0.023$ | $4.885 \pm 0.015$ | $4.677 \pm 0.009$ |
| **LOCAL** | $2.805 \pm 0.051$ | $3.361 \pm 0.045$ | $3.339 \pm 0.034$ | $3.091 \pm 0.022$ | $7.558 \pm 0.026$ | $7.839 \pm 0.019$ | $7.847 \pm 0.012$ | $7.728 \pm 0.007$ |
| **CQR** | $2.822 \pm 0.061$ | $3.470 \pm 0.052$ | $3.440 \pm 0.040$ | $3.139 \pm 0.026$ | $7.169 \pm 0.031$ | $7.454 \pm 0.024$ | $7.420 \pm 0.018$ | $7.241 \pm 0.009$ |
| **RoBAS–Full** | $2.907 \pm 0.056$ | $3.436 \pm 0.049$ | $3.403 \pm 0.035$ | $3.139 \pm 0.023$ | $2.782 \pm 0.063$ | $2.864 \pm 0.038$ | $2.702 \pm 0.023$ | $2.452 \pm 0.015$ |
| **RoBAS–EB** | $3.069 \pm 0.063$ | $3.498 \pm 0.051$ | $3.421 \pm 0.035$ | $3.140 \pm 0.023$ | $2.670 \pm 0.053$ | $2.901 \pm 0.040$ | $2.716 \pm 0.023$ | $2.457 \pm 0.015$ |

original papers for all other nonconformity scores.

**Models:** For the predictor $f$, we use a random forest with the default hyperparameters from `Scikit-learn` (Pedregosa et al., 2011). For the image datasets, we first map the images to a lower–dimensional latent space using a pretrained encoder. For `UTKFaces`, we use a ViT (Dosovitskiy et al., 2021) pretrained for facial recognition, and for `VentricularVolume` we use an ImageNet–pretrained ResNet34 (He et al., 2016).

**Results:** From Figure 2, we make two key observations. First, we observe that the widths of all standard and Bayes–assisted approaches expand with increasing covariate shift. The notable exception is **RoBAS**, which remains robust and performs similarly to **DTA**, achieving the smallest widths. On the other hand, at lower levels of covariate shift, where $f$ is a better fit for the calibration set, **RoBAS** performs competitively or matches the methods with the smallest widths.

We make a similar observation for the image regression datasets from Table 1. On the in–distribution subset, our method is competitive with those with the smallest widths, while on the out–of–distribution subset, it remains robust, attaining the smallest widths alongside **DTA**.

Like in the synthetic case, we note that **RoBAS–Full** often achieves slightly smaller widths than **RoBAS–EB** while transitioning more gradually towards the **DTA**–like regime.

## 5. Related Work

**Bayes–assisted conformal prediction.** Bayes–assisted conformal prediction (Vovk et al., 2005; Wasserman, 2011; Hoff, 2023; Deliu & Liseo, 2025) uses BWMs to design nonconformity scores, most commonly via the negative posterior predictive density. Fong & Holmes (2021) extend this idea beyond conjugate settings through add–one–in importance sampling, while Hoff (2023) show that, under mild conditions, the posterior predictive density score is Bayes–optimal among conformal procedures achieving the same (or higher) coverage. Building on this, Bersson & Hoff (2024) develop a Normal–Normal–Gamma BWM with closed–

form prediction intervals for small–area estimation, and Bhagwat et al. (2025) aggregate posterior predictive scores across multiple BWMs via Bayesian model averaging.

**Robust conformal prediction.** Robustness in conformal prediction has largely focused on maintaining coverage under weakened assumptions, such as distribution shift or corrupted calibration data. Under covariate shift, Tibshirani et al. (2019) propose weighted conformal methods that restore validity when the calibration/test density ratio can be estimated, while Gibbs & Candes (2021) develop adaptive online procedures that maintain coverage as the test distribution evolves. For corrupted labels, Feldman et al. (2023) give conditions under which CP sets remain approximately valid under dispersive label noise. The robustness addressed in our work is different in kind: we retain the usual finite–sample marginal guarantee under exchangeability, but target robustness of *efficiency* to the quality of working prior information. This addresses a failure mode specific to Bayes–assisted methods, where validity persists but efficiency can deteriorate sharply when prior information is inaccurate.

See Appendix A for an extended related work discussion.

## 6. Discussion

In this paper, we introduced **RoBAS**, a Bayes–assisted framework for constructing robust nonconformity scores based on residual BWMs. By design, the resulting scores adapt to the quality of the prior information: when the residuals are concentrated near zero, they yield the efficient, DTO–like prediction sets characteristic of Bayes–assisted approaches with accurate priors, while they provably revert toward the DTA score as the residual mean drifts away from zero. Across both synthetic experiments and real-world tabular and image regression tasks with distribution shift between the training data of $f$ and the calibration/test distribution, the proposed scores remain competitive with widely used nonconformity scores in–distribution and produce substantially tighter intervals under shift.

## Acknowledgments

KA and SC are supported by the EPSRC Centre for Doctoral Training in Modern Statistics and Statistical Machine Learning (EP/S023151/1). The authors are grateful to Guneet Singh Dhillon for helpful discussions.

## Impact Statement

Our work advances the reliability of conformal–based uncertainty quantification methods by making them robust to the bias of the underlying predictive model. This can improve the trustworthiness of uncertainty estimates in high–stakes applications such as healthcare, autonomous systems, and scientific decision–making, where reliable measures of predictive confidence are important for downstream decisions. At the same time, improved uncertainty quantification does not remove risks arising from biased data or incorrect modelling assumptions, and overly confident use of such methods could create a false sense of safety in real–world applications. As with other reliability methods, the societal impact depends on careful use, transparent reporting of assumptions, and evaluation in the specific domain where the method is applied.

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

# Appendix Table of Contents

# A. Extended Related Work

## A.1. Conformal Prediction and Efficient Prediction Sets

Conformal prediction (CP) is a general framework for producing prediction intervals with *finite–sample, distribution–free* marginal coverage under exchangeability (Vovk et al., 2005). A central practical consideration is *efficiency*: among methods with the same target coverage, tighter prediction intervals are typically more useful for decision making, while overly conservative intervals can be uninformative. In CP, efficiency is driven largely by the choice of the nonconformity score, which determines how candidate labels are ranked against calibration examples; consequently, a large body of the literature has focused on designing scores that adapt to heteroscedasticity, local difficulty, or uncertainty estimates in order to reduce average widths while maintaining validity.

A number of methods seek tighter prediction intervals by incorporating input–dependent structure into the score. For regression, Romano et al. (2019) propose Conformalised Quantile Regression, which fits lower and upper conditional quantiles and conformalises the resulting residual–like errors to achieve finite-sample coverage; when the quantile model captures heteroscedasticity, intervals can be significantly shorter than those based on absolute residuals. Guan (2023) develops a localised CP framework that alters the calibration comparison (e.g., by localisation/weighting), producing tighter intervals in regions of the covariate space where the predictor is more accurate. Similarly, for the case of classification Romano et al. (2020) develop adaptive prediction sets that use estimated class probabilities to shrink sets on "easy" inputs while maintaining coverage guarantees.

Recent work also explores improving efficiency by strengthening representations or combining multiple conformal procedures. Seedat et al. (2023) use self–supervised learning to improve adaptive CP, leveraging improved representations to sharpen uncertainty estimates and reduce widths. Xie et al. (2024) propose boosted conformal prediction intervals, combining/boosting procedures to reduce widths while preserving validity. Kiyani et al. (2024) study width optimisation in CP, directly targeting expected widths subject to coverage constraints. Finally, complementary theoretical work by Dhillon et al. (2024) quantifies the expected set size in the split conformal setting by decomposing it into the nonconformity score distribution and a volume–translating multiplicative factor.

## A.2. Bayes–Assisted Conformal Prediction

Bayes–assisted conformal prediction (Vovk et al., 2005; Wasserman, 2011; Hoff, 2023; Deliu & Liseo, 2025) uses Bayesian models to design nonconformity scores – most commonly, the negative posterior predictive density under a Bayesian working model (BWM). This combination is appealing for two reasons. First, Bayesian modelling provides a principled way to incorporate prior or side information (including hierarchical structure), which is especially valuable in small–data regimes such as small area estimation (Bersson & Hoff, 2024; 2025); second, when the BWM is a good description of the data, posterior predictive scores can yield highly efficient prediction intervals on average.

A foundational perspective on reconciling Bayesian procedures with frequentist guarantees appears in Wasserman (2011) ("Frasian" inference), emphasising procedures that preserve desirable Bayesian behaviour while controlling frequentist error. Burnaev & Vovk (2014) provide a theoretical analysis of conformalised ridge regression, proving that under standard Gaussian assumptions, the method is asymptotically efficient – producing prediction intervals that converge to the optimal Bayesian intervals – while retaining validity guarantees under the weaker i.i.d. assumption. In the conformal context, Fong & Holmes (2021) develop Conformal Bayesian Computation, using posterior predictive quantities to construct conformal scores (with add–one–in sampling to amortise leave–one–out computations), allowing for Bayes–assisted CP beyond conjugate settings.

A key theoretical result for Bayes–assisted CP is due to Hoff (2023): among conformal procedures achieving the same (or higher) coverage, the posterior predictive density score is *Bayes-risk optimal* (i.e., it minimises expected interval width under the assumed prior and model, under mild conditions). This shows that Bayes–assisted scores can produce substantially shorter prediction intervals when the working prior/model is accurate. However, this same dependence on the prior/model highlights an important practical limitation: while conformal validity is robust to misspecification, *efficiency need not be*. When the working prior is poorly aligned with the observed data, Bayes–assisted intervals can become significantly wider (Bersson & Hoff, 2024), undermining the very motivation for using Bayes–assisted scores.

Recent work by Bersson & Hoff (2024) develops Bayes–assisted CP by introducing a Normal–Normal–Gamma BWM that admits a closed–form expression for its prediction intervals. Applying their method to small–area estimation, they demonstrate that it effectively utilises side information to achieve lower widths on average than the Distance–To–Average

score. In a similar vein, Bersson & Hoff (2025) construct prediction sets for species abundance by encoding indirect information from neighbouring areas into a BWM. On a different note, Bhagwat et al. (2025) present a Bayes–assisted score that aggregates posterior predictive density scores across multiple Bayesian models via Bayesian model averaging. They show that the aggregated score is itself a valid conformity score, and prove that if the true model is contained in the candidate model class, the resulting prediction interval converges to the optimal conformal Bayes interval (and hence achieves asymptotically optimal expected width).

Our contribution is complementary to these works: instead of specifying a BWM for the typically high–dimensional conditional data–generating process, we apply a BWM only to the *scalar residuals* of a fixed predictive model and design a BWM that adapts to the mean of those residuals.

### A.3. Robust Conformal Prediction

Robustness in CP has largely focused on maintaining coverage guarantees under weakened assumptions, such as departures from exchangeability, distribution shift, or corrupted calibration data due to label noise or missingness. Under distribution shift, an important setting is covariate shift between the data used for calibration and testing. Tibshirani et al. (2019) study CP under covariate shift and propose weighted conformal methods (based on importance weights) that restore validity when the density ratio between test and calibration distributions can be estimated. Complementarily, Gibbs & Candes (2021) develop adaptive conformal inference under distribution shift, proposing online procedures that aim to maintain coverage as the test distribution evolves over time or varies across environments.

Beyond distribution shift, robustness has also been studied for corrupted calibration labels. Feldman et al. (2023) analyse the effect of dispersive label noise and provide conditions under which CP sets remain approximately valid despite mislabelled data. Penso & Goldberger (2024) propose an alternative conformal score designed to reduce sensitivity to label noise, improving empirical robustness of prediction sets when calibration labels are unreliable. Finally, Zaffran et al. (2023) study CP with missing values and introduce procedures that handle incomplete covariates while retaining coverage guarantees under assumptions on the missingness mechanism.

The robustness addressed in our work is different in kind. We do *not* modify the conformal validity guarantee itself: our coverage is the usual finite–sample marginal guarantee under exchangeability of calibration/test pairs. Instead, we focus on robustness of *efficiency* to the *quality of working information in the prior* used to define our Bayes–assisted score. In our setup, prior quality is linked to the quality of the underlying predictor through the mean of its residuals. This robustness to the working information in the prior is complementary to (rather than a substitute for) robustness to non–exchangeability or robustness to label noise: it targets a failure mode specific to Bayes–assisted conformal methods, where validity persists but efficiency can deteriorate sharply when prior information is wrong.

# B. Further Discussion

Here, we describe limitations of our work and interesting directions for future work.

**Robustness is currently targeted to mean misalignment.** A limitation of the current robustness guarantee is that it is specifically tied to the *mean of the residuals* (equivalently, systematic bias in the predictor's mean prediction). This is formulated in terms of the residual average $\bar{r}_n$: as $|\bar{r}_n| \to \infty$, the Bayes–assisted score becomes equivalent to the Distance–To–Average score (Theorem 3.1 and Proposition 3.4). This means RoBAS is most useful in scenarios where model degradation manifests as a shift in the residual mean – for example, when a pretrained black–box predictor becomes biased on a target subpopulation, increasing the average residual away from zero.

An interesting direction for future work is to broaden robustness beyond mean effects. In many applications, model degradation may instead (or additionally) appear through changes in the *variance* or tail behaviour of the residuals, even when the residual mean is close to zero. Since RoBAS's adaptive behaviour is designed around down–weighting an informative prior on the *mean* when $\bar{r}_n$ conflicts with that prior, it may not protect against width inflation when the dominant misspecification is in the residual *scale*. A natural extension would be to introduce analogous hierarchical/heavy–tailed adaptivity for scale parameters (e.g., priors or empirical Bayes shrinkage for $\sigma^2$), so that the score can revert to an appropriate robust baseline not only under mean shift but also under variance shift.

**Computational efficiency: toward fully closed–form endpoints.** While both RoBAS–Full and RoBAS–EB admit simple expressions for their scores (eliminating the MCMC averaging required by other Bayes–assisted approaches), the prediction interval endpoints are still computed via an iterative procedure, relying on evaluations of the conformal $p$–value. In contrast, for the Normal–Normal–Gamma model there exists a *closed–form expression* for the prediction interval (Bersson & Hoff, 2024). Ideally, we would also like to calculate RoBAS intervals using direct formulas, avoiding the need for root–finding.

Currently, the scores for both RoBAS–Full and RoBAS–EB do not appear to admit the same simplifying structure exploited in the Normal–Normal–Gamma model to obtain fully closed–form prediction intervals. It would be interesting to explore whether RoBAS–Full or RoBAS–EB can be approximated to yield closed–form endpoints without losing their key qualitative properties. One direction is to search for transformations or surrogate scores that are (approximately) order–equivalent, since conformal sets are invariant to strictly monotone transformations of the score.

**Additional practical considerations.** The paper's validity results (as standard in split conformal) require exchangeability between calibration and test pairs and treat the predictor $f$ as fixed; the "distribution shift" studied is between the training data used to fit $f$ and the calibration/test distribution, rather than between calibration and test, so coverage is not itself challenged by non–exchangeability. Extending the robustness concept to settings where calibration and test are not exchangeable would be an interesting direction for future work.

# C. Experimental Setup

We provide the full details of our experimental setup in this section. We describe the setup for each of the datasets in turn.

## C.1. Synthetic Dataset

**Dataset.**    We generate data $\epsilon_i \sim \mathcal{N}(\theta, \sigma^2)$ with a test set of size $n_{\text{test}} = 1000$. We compare different nonconformity scores for different calibration sizes $n_{\text{cal}} \in \{5, 10, 25, 50\}$, and different values of $\theta$ and $\sigma^2$.

Note that we do *not* have a training set here. Indeed, our data $\epsilon_i$ can be considered as the residuals of some model $f$ where the mean of these residuals diverges from zero as the model's performance decreases. This setting allows us to simulate the effect of varying model quality on the width of the prediction intervals.

**Nonconformity scores.**    We compare **RoBAS–Full** and **RoBAS–EB** with **NNG**, **DTA** and **DTO**. We chose these nonconformity scores to highlight the strengths and limitations of our approach across different regimes. Specifically, in this setting **DTO** serves as an oracle score when $\theta = 0$ (i.e. when we have an unbiased predictor), while **DTA** acts as a robust baseline when $|\theta| \gg 0$ (i.e. when we have a biased predictor).

For **NNG**, we compute the prediction intervals using the closed–form expression in Theorem 2 of Bersson & Hoff (2024), which requires specifying the hyperparameter $\tau^2$. This can be done using a separate validation set or available prior information. Given the limited access to both of these in the standard conformal prediction setting, we instead set $\tau^2 = 1/n_{\text{cal}}$, which equally weights the influence of the prior and data. This is achieved by considering the posterior of $\theta$ for BWM (11). Indeed, the posterior of $\theta$ for observed data $r_{1:n} = \{r_i\}_{i=1}^n$ is a Student's t–distribution with location given by:

$$\mu_{\text{post}} = \frac{W_{\text{data}}\bar{r}_{1:n}}{W_{\text{prior}} + W_{\text{data}}},$$

where $W_{\text{prior}} = 1/\tau^2$ and $W_{\text{data}} = n$. Balancing these weights naturally gives $\tau^2 = 1/n_{\text{cal}}$.

**Prediction intervals.**    We use a nominal error rate of $\alpha = 0.1$; when $n_{\text{cal}}$ is too small to satisfy the nominal error rate, we set $\alpha$ to the smallest possible error rate of $1/(n_{\text{cal}} + 1)$. We compute our prediction intervals using Algorithm 1. For **NNG**, we use the closed–form expression for the prediction interval in Bersson & Hoff (2024). All experiments are repeated for 300 trials.

**Models.**    As discussed earlier, we do not require a model for this setting.

**Computational resources.**    All experiments were run on a machine with an Intel Xeon Gold 6132 (28 logical CPU cores) and an NVIDIA GeForce RTX 2080 Ti GPU.

## C.2. Tabular Datasets

**Datasets.**    For tabular datasets we consider the setting where there exists covariate shift between the data used to train our predictor $f$ and our calibration/test data. This occurs in many settings when, for example, we have a black–box predictor that has been pretrained on a broad, general dataset, but we wish to calibrate and deploy it for a particular sub–population.

We consider standard UCI datasets (Kelly et al.) used in previous works (Romano et al., 2019; Tibshirani et al., 2019; Sesia & Romano, 2021; Zaffran et al., 2023; Plassier et al., 2025): Facebook comment volume (`facebook_1`), airfoil self–noise (`airfoil`), concrete compressive strength (`concrete`). We provide a brief description of these datasets below.

- `Facebook_1`: This dataset involves predicting the volume of comments a post will receive within a specific timeframe. The features ($d = 54$) are derived from metadata and page popularity metrics, such as the number of likes and the length of time the post has been published.

- `Airfoil`: Obtained from NASA, this dataset tasks the predictor with estimating the scaled sound pressure level (in decibels) of an airfoil. The $d = 5$ features describe aerodynamic properties, including the frequency, angle of attack, chord length, free–stream velocity, and suction side displacement thickness.

- `Concrete`: This dataset relates the composition of concrete mixtures to their structural integrity. The goal is to regress the compressive strength (in MPa) based on $d = 8$ input variables representing the age of the concrete and the quantities of ingredients such as cement, blast furnace slag, fly ash, water, and superplasticizer.

To simulate covariate shift for these datasets, we follow Tibshirani et al. (2019) and sample training points with replacement, using probabilities proportional to:

$$w(\mathbf{x}) = \exp(\mathbf{x}^T \boldsymbol{\beta}), \ \ \boldsymbol{\beta} = \underbrace{(-c, 0, \dots, 0, c)}_{d},$$

where $c \in \mathbb{R}^+$, $\mathbf{x}, \boldsymbol{\beta} \in \mathbb{R}^d$, and $d$ is the number of features in the dataset. This corresponds to sampling from an *exponentially tilted* covariate distribution, where data points with smaller values for the first continuous feature and larger values for the last are more likely to be sampled as $c$ increases. We consider different levels of covariate shift by varying $c$, where we use $c = 0$ to denote no covariate shift.

Following Romano et al. (2019), we use 20% of the data for testing, and 80% for training. The training set is split equally into a proper training set used to train our predictor and a calibration set, where the proper training set is sampled as above. This ensures that our calibration/test sets follow the same distribution. To make our experiments computationally tractable on our machine, we cap the number of training points to 5000. Moreover, we vary the number of calibration points by sampling $n_{\text{cal}} \in \{5, 10, 25, 50\}$ from the calibration set. We also present the results using the full calibration set in Appendix F.

**Nonconformity scores.** We compare **RoBAS–EB** and **RoBAS–Full** with **NNG**, **DTA**, **DTO**, **CQR** (Romano et al., 2019), **LOCAL** (Guan, 2023), **CB** (Fong & Holmes, 2021) and **CBMA** (Bhagwat et al., 2025). We emphasise that while **CB** and **CBMA** are also Bayes–assisted approaches, they are not directly comparable to ours as they conformalise the posterior predictive of a full Bayesian model for $Y|\mathbf{X}$, whereas our method treats $f$ as fixed and places a BWM *only* on its residuals. In particular, their approaches strictly requires the use of Bayesian models. Nonetheless, we report their results in Appendix F for completeness.

We set the $\tau^2$ parameter of **NNG** in the same way as Appendix C.1. For all remaining methods, we adopt the hyperparameter settings from their respective citations.

**Prediction intervals.** For all the nonconformity scores, we use the training set, $\mathcal{D}_{\text{train}}$, to fit a predictor $f : \mathcal{X} \rightarrow \mathcal{Y}$. To make the comparisons with **CB** and **CBMA** fair, we fit its posterior on the training set. Below we provide further details:

- **CQR**, **LOCAL**: For these nonconformity scores, we use our training set to learn a predictor $f : \mathcal{X} \rightarrow \mathcal{Y}$. We then use this learned predictor to compute our nonconformity scores on the calibration set. The prediction intervals are computed following the same procedure as in the original papers.

- **RoBAS–Full**, **RoBAS–EB NNG**, **DTO**, **DTA**: For these nonconformity scores, we use our training set to learn a predictor $f : \mathcal{X} \rightarrow \mathcal{Y}$. We then use this learned predictor to compute our nonconformity scores on the calibration set. We compute our prediction intervals using Algorithm 1. For **NNG**, we use the closed–form expression for the prediction interval in Bersson & Hoff (2024).

- **CB, CBMA**: To make comparisons fair and make use of the available training data for all other approaches, we modify the nonconformity score to be the training–conditional density, $-p(y|\mathbf{x}, \mathcal{D}_{\text{train}})$, with the calibration set being used exclusively for computing the nonconformity scores. We compute the prediction intervals using a fine grid of candidate $y$ values in a similar way to **CB**.[2]

**Models.** Firstly, note that **CB** and **CBMA** require strictly Bayesian models, while all other nonconformity scores can use any arbitrary predictor $f : \mathcal{X} \rightarrow \mathcal{Y}$. We detail the model choices for the different nonconformity scores below:

- **CQR**, **LOCAL**, **DTO**, **DTA**, **NNG**, **RoBAS–Full**, **RoBAS–EB**: We follow **CQR**, and use a random forest model. We choose this over their kernel ridge regression and neural network model as it is a simple out–of–the–box choice and

---

[2]We found that the original grid of 100 equally spaced candidates in $\left[\min(\{y_i^{\text{cal}}\}_{i=1}^{n_{\text{cal}}}) - 2, \max(\{y_i^{\text{cal}}\}_{i=1}^{n_{\text{cal}}}) + 2\right]$ where $y_i^{\text{cal}}$ is the $i$th calibration response value, failed to provide us with the desired coverage guarantee. We therefore implemented an adaptive gridding method which expanded the grid outward until the extreme candidate values were rejected.

because it has been implemented for both the **CQR** and **LOCAL** nonconformity scores in their codebase. We choose the default hyperparameters from `scikit-learn` (Pedregosa et al., 2011).

- **CB, CBMA**: For **CB**, we use a Bayesian linear regression model like in Fong & Holmes (2021). Like Bhagwat et al. (2025), we fit four different Bayesian linear regression models for **CBMA**, where model $i \in \{1, 2, 3, 4\}$ uses the first $d \times i/4$ features. For **CB**, we run 4 MCMC chains to sample from the posterior, where we take 100 samples from each chain. We do the same for **CBMA**, except that we now have to run 4 MCMC chains for each model.

**Computational resources.** All experiments were run on a machine with an Intel Xeon Gold 6132 (28 logical CPU cores) and an NVIDIA GeForce RTX 2080 Ti GPU.

### C.3. Image Datasets

**Datasets.** For our image regression datasets, we consider the setting where we have in/out–of–distribution training data. We use the `UTKFaces` (Zhang et al., 2017) and `VentricularVolume` datasets from Gustafsson et al. (2023). We provide a brief description of these datasets below:

- `UTKFaces`: This dataset consists of over 20,000 aligned face images covering a large diversity of ages, genders, and ethnicities. The regression task is to predict the *age* of an individual given their face image.

- `VentricularVolume`: This is a medical imaging dataset consisting of 5,088 cardiac MRI scans derived from the UK Biobank. The regression task involves predicting the size of the left ventricle based on the MRI scan.

For the `UTKFaces` dataset, we first subset the data so that the in–distribution data includes ages between 18–50, and the out–of–distribution data includes ages larger than 50. Our training data always comes from the in–distribution subset, while the calibration/test data are either the in–distribution or out–of–distribution subset. For the `VentricularVolume` dataset we use the in–distribution and out–of–distribution subsets provided.

For both datasets, we use the training and test sets provided. We split the provided test set equally into a calibration set and a final test set; this ensures that the calibration and test data are identically distributed. We vary the number of calibration points by sampling $n_{\text{cal}} \in \{5, 10, 25, 50\}$ from the calibration set. We also present the results using the full calibration set in Appendix F. As with the tabular datasets, we cap the training set at 5,000 points to keep the experiments computationally tractable on our machine.

**Nonconformity scores.** We use the same nonconformity scores with the same setup as in Appendix C.2.

**Prediction intervals.** We compute our prediction intervals in the same way as in Appendix C.2.

**Models.** We use the same models as in Appendix C.2, where we now first map our high–dimensional inputs into a lower–dimensional latent space using a pretrained encoder. Specifically, for `UTKFaces` we use a ViT (Dosovitskiy et al., 2021) pretrained for facial recognition, and for `VentricularVolume` we use an ImageNet–pretrained ResNet34 (He et al., 2016)[3].

**Computational resources.** All experiments were run on a machine with an Intel Xeon Gold 6132 (28 logical CPU cores) and an NVIDIA GeForce RTX 2080 Ti GPU.

---

[3]More specifically, for `UTKFaces` we use the ViT/Ti-8 from https://github.com/gau-nernst/timm-face, while for `VentricularVolume` we use the ImageNet–pretrained ResNet34 model from `torchvision` (TorchVision maintainers and contributors, 2016).

# D. Proofs

Appendix D.1 presents three useful results: Lemma D.1 shows the equivalence of the Bayes–assisted nonconformity score (9) and the leave–one–out marginal likelihood score, Lemma D.2 provides a sufficient condition for the conformal prediction set to be an interval, and Lemma D.3 establishes a compact–uniform version of the tail–transfer property for normal scale mixtures with regularly varying mixing density. The latter two results are used in the proof of Theorem 3.5 and Theorem D.4, respectively. The following sections provide proofs for results mentioned in the main body of the paper. Throughout, we follow the same setup as in §2.1.

## D.1. Auxiliary Results

**Lemma D.1.** *For any $i, j \in \{1, \ldots, n+1\}$, the ordering of posterior predictive scores is equivalent to the ordering of marginal likelihoods of the leave–one–out data. That is,*

$$p(\mathbf{z}_i | \mathbf{z}_{1:n+1,-i}) \geq p(\mathbf{z}_j | \mathbf{z}_{1:n+1,-j}) \quad \text{if and only if} \quad p(\mathbf{z}_{1:n+1,-j}) \geq p(\mathbf{z}_{1:n+1,-i}).$$

*Proof.* Let $\mathbf{z}_{1:n+1} = \{\mathbf{z}_i\}_{i=1}^{n+1}$, and, for any index $i \in \{1, \ldots, n+1\}$, write $\mathbf{z}_{1:n+1,-i} = \mathbf{z}_{1:n+1} \setminus \{\mathbf{z}_i\}$. Fix any $i, j \in \{1, \ldots, n+1\}$. Then,

$$
\begin{aligned}
p(\mathbf{z}_i \mid \mathbf{z}_{1:n+1,-i}) \geq p(\mathbf{z}_j \mid \mathbf{z}_{1:n+1,-j}) &\iff \frac{p(\mathbf{z}_{1:n+1})}{p(\mathbf{z}_{1:n+1,-i})} \geq \frac{p(\mathbf{z}_{1:n+1})}{p(\mathbf{z}_{1:n+1,-j})} \\
&\iff \frac{1}{p(\mathbf{z}_{1:n+1,-i})} \geq \frac{1}{p(\mathbf{z}_{1:n+1,-j})} \\
&\iff p(\mathbf{z}_{1:n+1,-j}) \geq p(\mathbf{z}_{1:n+1,-i}).
\end{aligned}
$$

This proves the equivalence of the two orderings. In particular, this lemma shows that the Bayes–assisted nonconformity score, $-p(\mathbf{z}_i \mid \mathbf{z}_{1:n+1,-i})$, is equivalent (up to a strictly monotone transformation) to the leave–one–out marginal likelihood score, $p(\mathbf{z}_{1:n+1,-i})$. $\qquad \square$

**Lemma D.2** (Bersson & Hoff, 2024, Lemmas 1–2). *Fix $\mathbf{x}_{n+1}$ and $\mathbf{z}_{1:n}$, and let $s_1(y), \ldots, s_{n+1}(y)$ denote the nonconformity scores associated with a candidate response $y \in \mathbb{R}$. Define*

$$I_i = \{y \in \mathbb{R} \mid s_i(y) \geq s_{n+1}(y)\}, \qquad i = 1, \ldots, n+1.$$

*If each $I_i$ is an interval and there exists $y_0 \in \mathbb{R}$ such that $y_0 \in I_i$ for all $i = 1, \ldots, n+1$, then, for every $\alpha \in [0, 1)$,*

$$\{y \in \mathbb{R} : \rho(y) > \alpha\}$$

*is an interval. Consequently, the conformal prediction set*

$$C_\alpha(\mathbf{x}_{n+1}; \mathbf{z}_{1:n}) = \{y \in \mathcal{Y} : \rho(y) > \alpha\}$$

*is an interval, up to intersection with $\mathcal{Y}$.*

**Lemma D.3** (Compact–uniform local tail equivalence). *Assume that the mixing density satisfies*

$$g_{\tau^2}(w) \sim C w^{-\delta}, \qquad w \to \infty,$$

*for some constants $C > 0$ and $\delta > 1$. Let $\mu \in \mathbb{R}$ and $\gamma > 0$ be fixed. For $a > 0$, define*

$$f_a(z) = \int_0^\infty \varphi(z; \mu, a + \gamma w) \, g_{\tau^2}(w) \, dw,$$

*where $\varphi(\cdot; \mu, \nu^2)$ denotes the density of $\mathcal{N}(\mu, \nu^2)$. Then, for every compact $K \subset (0, \infty)$ and every bounded $B \subset \mathbb{R}$,*

$$\sup_{a \in K} \sup_{\xi \in B} \left| \frac{f_a(x + \xi)}{f_a(x)} - 1 \right| \to 0 \qquad \text{as } |x| \to \infty.$$

*Proof.* This is a compact–uniform version of the standard tail–transfer property for normal scale mixtures with regularly varying mixing density; it uses the Uniform Convergence Theorem and Potter bounds for regularly varying functions, see Bingham et al. (1987, Theorems 1.5.2 and 1.5.6). We give the details for completeness.

Let
$$a_- = \inf K > 0, \qquad a_+ = \sup K < \infty,$$
and write
$$y = |z - \mu|.$$
We first prove the uniform tail asymptotic
$$\sup_{a \in K} \left| \frac{f_a(z)}{A|z-\mu|^{1-2\delta}} - 1 \right| \to 0 \qquad \text{as } |z| \to \infty,$$
where
$$A = C\gamma^{\delta-1}(2\pi)^{-1/2}2^{\delta-1/2}\Gamma\left(\delta - \frac{1}{2}\right).$$

Fix $T > 0$ and decompose
$$f_a(z) = I_{a,T}(z) + J_{a,T}(z),$$
where $I_{a,T}$ integrates over $w \in (0, T)$ and $J_{a,T}$ integrates over $w \in [T, \infty)$. Since $a + \gamma w \in [a_-, a_+ + \gamma T]$ for $a \in K$ and $w \in (0, T)$,
$$I_{a,T}(z) \le (2\pi a_-)^{-1/2} \exp\left\{-\frac{y^2}{2(a_+ + \gamma T)}\right\} \int_0^T g_{\tau^2}(w)\, dw.$$

Therefore
$$I_{a,T}(z) = o(y^{1-2\delta}) \qquad \text{uniformly over } a \in K.$$

It remains to study $J_{a,T}$. Use the change of variables
$$w = \frac{y^2 r}{\gamma}, \qquad dw = \frac{y^2}{\gamma}\, dr.$$

Then
$$J_{a,T}(z) = \frac{y}{\gamma\sqrt{2\pi}} \int_{\gamma T/y^2}^{\infty} \left(r + \frac{a}{y^2}\right)^{-1/2} \exp\left\{-\frac{1}{2(r + a/y^2)}\right\} g_{\tau^2}\left(\frac{y^2 r}{\gamma}\right) dr.$$

Define
$$R(u) = \frac{g_{\tau^2}(u)}{Cu^{-\delta}}.$$

By assumption, $R(u) \to 1$ as $u \to \infty$. Hence
$$g_{\tau^2}\left(\frac{y^2 r}{\gamma}\right) = C\gamma^\delta y^{-2\delta} r^{-\delta} R\left(\frac{y^2 r}{\gamma}\right).$$

Thus
$$\frac{J_{a,T}(z)}{C\gamma^{\delta-1}(2\pi)^{-1/2}y^{1-2\delta}} = \int_{\gamma T/y^2}^{\infty} \left(r + \frac{a}{y^2}\right)^{-1/2} \exp\left\{-\frac{1}{2(r + a/y^2)}\right\} r^{-\delta} R\left(\frac{y^2 r}{\gamma}\right) dr.$$

Let $\varepsilon > 0$. Choose $T$ sufficiently large that
$$T \ge \frac{a_+}{\gamma} \qquad \text{and} \qquad |R(u) - 1| \le \varepsilon \quad \text{for all } u \ge T.$$

For $r \ge \gamma T/y^2$ and $a \in K$,
$$\frac{a}{y^2} \le \frac{a_+}{\gamma T} r \le r,$$

so

$$r \le r + \frac{a}{y^2} \le 2r.$$

Consequently,

$$\left(r + \frac{a}{y^2}\right)^{-1/2} \exp\left\{-\frac{1}{2(r + a/y^2)}\right\} r^{-\delta} \le r^{-\delta-1/2} \exp\left(-\frac{1}{4r}\right).$$

The right-hand side is integrable on $(0, \infty)$: the exponential term controls the origin, and $\delta > 1$ controls infinity. Moreover, for each fixed $r > 0$,

$$\left(r + \frac{a}{y^2}\right)^{-1/2} \exp\left\{-\frac{1}{2(r + a/y^2)}\right\} r^{-\delta} \to r^{-\delta-1/2} \exp\left(-\frac{1}{2r}\right)$$

uniformly over $a \in K$. Dominated convergence therefore gives

$$\int_{\gamma T/y^2}^{\infty} \left(r + \frac{a}{y^2}\right)^{-1/2} \exp\left\{-\frac{1}{2(r + a/y^2)}\right\} r^{-\delta} \, dr \to \int_0^{\infty} r^{-\delta-1/2} \exp\left(-\frac{1}{2r}\right) \, dr$$

uniformly over $a \in K$. The replacement of $R(y^2 r/\gamma)$ by 1 has limsup bounded by

$$\varepsilon \int_0^{\infty} r^{-\delta-1/2} \exp\left(-\frac{1}{4r}\right) \, dr,$$

which is finite. Since $\varepsilon > 0$ is arbitrary,

$$\frac{J_{a,T}(z)}{C\gamma^{\delta-1}(2\pi)^{-1/2} y^{1-2\delta}} \to \int_0^{\infty} r^{-\delta-1/2} \exp\left(-\frac{1}{2r}\right) \, dr$$

uniformly over $a \in K$. Finally, with the change of variables $q = 1/(2r)$,

$$\int_0^{\infty} r^{-\delta-1/2} \exp\left(-\frac{1}{2r}\right) \, dr = 2^{\delta-1/2} \Gamma\left(\delta - \frac{1}{2}\right).$$

Combining this with the negligible small-$w$ contribution proves

$$f_a(z) \sim A|z - \mu|^{1-2\delta}$$

uniformly over $a \in K$.

We now deduce the desired local ratio property. Let

$$M = \sup_{\xi \in B} |\xi| < \infty.$$

Since $B$ is bounded, $|x + \xi - \mu| \to \infty$ uniformly over $\xi \in B$ as $|x| \to \infty$. Hence the preceding asymptotic gives, uniformly over $a \in K$ and $\xi \in B$,

$$f_a(x + \xi) = A|x + \xi - \mu|^{1-2\delta}\{1 + o(1)\},$$

and, uniformly over $a \in K$,

$$f_a(x) = A|x - \mu|^{1-2\delta}\{1 + o(1)\}.$$

Therefore

$$\frac{f_a(x + \xi)}{f_a(x)} = \frac{1 + o(1)}{1 + o(1)} \left(\frac{|x + \xi - \mu|}{|x - \mu|}\right)^{1-2\delta}$$

uniformly over $a \in K$ and $\xi \in B$. Finally,

$$\sup_{\xi \in B} \left|\frac{|x + \xi - \mu|}{|x - \mu|} - 1\right| \le \frac{M}{|x - \mu|} \to 0.$$

The power term therefore converges to 1 uniformly over $\xi \in B$, proving

$$\sup_{a \in K} \sup_{\xi \in B} \left|\frac{f_a(x + \xi)}{f_a(x)} - 1\right| \to 0.$$

$\square$

## D.2. Proofs of Section 3.2

**Theorem 3.1** (Asymptotic Robustness of Heavy–Tailed BWM). *Fix $n \geq 1$. Let $(\mathbf{r}_{1:n}^{(m)})_{m \geq 1}$, with $\mathbf{r}_{1:n}^{(m)} \in \mathbb{R}^n$, be a sequence of residuals such that $|\bar{r}_n^{(m)}| \to \infty$ as $m \to \infty$. Under BWM (12), the score function (10) satisfies, for every fixed $\Delta \in \mathbb{R}$,*

$$s(\bar{r}_n^{(m)} + \Delta, \mathbf{r}_{1:n}^{(m)}) = -p(\bar{r}_n^{(m)} + \Delta \mid \mathbf{r}_{1:n}^{(m)}) \to h_n(|\Delta|)$$

*as $m \to \infty$, where $h_n : [0, \infty) \to \mathbb{R}$ is the strictly increasing function*

$$h_n(u) = -\frac{1}{\sqrt{2\pi\sigma^2(1 + 1/n)}} \exp\left(-\frac{u^2}{2\sigma^2(1 + 1/n)}\right).$$

*Proof.* For notational convenience, write

$$\bar{r}_m := \bar{r}_n^{(m)}, \qquad \mathbf{r}_m := \mathbf{r}_{1:n}^{(m)}.$$

Under BWM (12), the likelihood depends on $\mathbf{r}_m$ through $\bar{r}_m$. We first show that

$$\theta - \bar{r}_m \mid \mathbf{r}_m \Rightarrow \mathcal{N}\left(0, \frac{\sigma^2}{n}\right) \qquad \text{as } m \to \infty.$$

That is, for large $m$, the posterior distribution of $\theta$ given $\mathbf{r}_m$ is approximately $\mathcal{N}(\bar{r}_m, \frac{\sigma^2}{n})$. Closely related large-observation posterior robustness results go back to Dawid (1973); see also Pericchi & Smith (1992); Pericchi & Sansó (1995).

Write the model in canonical form with natural parameter

$$\eta = \frac{n}{\sigma^2}\theta.$$

Following Tweedie's formula for exponential families (see Efron (2011, §2)), the posterior cumulant generating function of $\eta$ given $\mathbf{r}_m$ is

$$\kappa_m(\xi) = \log \mathbb{E}\left[e^{\eta\xi} \mid \mathbf{r}_m\right] = \lambda(\bar{r}_m + \xi) - \lambda(\bar{r}_m),$$

where

$$\lambda(z) = \log \frac{f(z)}{f_0(z)}, \qquad f_0(z) = \mathcal{N}\left(z; 0, \frac{\sigma^2}{n}\right), \qquad f(z) = \int_0^\infty \mathcal{N}\left(z; 0, \frac{\sigma^2}{n} + \gamma\tau^2\right) g_{\tau^2}(\tau^2)\, d\tau^2.$$

Consider the centred natural parameter

$$\eta_m^\star = \eta - \frac{n}{\sigma^2}\bar{r}_m.$$

Its conditional log–mgf is

$$\begin{aligned}
\log \mathbb{E}\left[e^{\eta_m^\star \xi} \mid \mathbf{r}_m\right] &= \kappa_m(\xi) - \frac{n}{\sigma^2}\bar{r}_m\xi \\
&= \log \frac{f(\bar{r}_m + \xi)}{f(\bar{r}_m)} - \log \frac{f_0(\bar{r}_m + \xi)}{f_0(\bar{r}_m)} - \frac{n}{\sigma^2}\bar{r}_m\xi \\
&= \log \frac{f(\bar{r}_m + \xi)}{f(\bar{r}_m)} + \frac{n}{2\sigma^2}\xi^2,
\end{aligned}$$

where the final equality uses the explicit Gaussian form of $f_0$.

By the regular-variation assumption on $g_{\tau^2}$, the induced marginal density $f$ is regularly varying as $|z| \to \infty$ (see Cortinovis & Caron (2024, Proposition 4.3)). Therefore, for each fixed $\xi \in \mathbb{R}$,

$$\frac{f(\bar{r}_m + \xi)}{f(\bar{r}_m)} \to 1 \qquad \text{as } m \to \infty,$$

because $|\bar{r}_m| \to \infty$, and so $\log(f(\bar{r}_m + \xi)/f(\bar{r}_m)) \to 0$. Consequently,

$$\log \mathbb{E}\left[e^{\eta_m^\star \xi} \mid \mathbf{r}_m\right] \longrightarrow \frac{n}{2\sigma^2}\xi^2 \qquad \text{as } m \to \infty.$$

The limit $\frac{n}{2\sigma^2}\xi^2$ is exactly the cumulant generating function of $\mathcal{N}(0, n/\sigma^2)$. Hence,

$$\eta_m^\star \mid \mathbf{r}_m \Rightarrow \mathcal{N}\left(0, \frac{n}{\sigma^2}\right) \qquad \text{as } m \to \infty.$$

Transforming back to $\theta = (\sigma^2/n)\eta$ gives

$$\theta - \bar{r}_m = \frac{\sigma^2}{n}\eta_m^\star \mid \mathbf{r}_m \Rightarrow \mathcal{N}\left(0, \frac{\sigma^2}{n}\right) \qquad \text{as } m \to \infty.$$

Let $\varphi(\cdot; \mu, \nu^2)$ denote the density of $\mathcal{N}(\mu, \nu^2)$. Under BWM (12), for fixed $\Delta \in \mathbb{R}$,

$$\begin{aligned} p(\bar{r}_m + \Delta \mid \mathbf{r}_m) &= \mathbb{E}\left[p(\bar{r}_m + \Delta \mid \theta) \mid \mathbf{r}_m\right] \\ &= \mathbb{E}\left[\varphi(\bar{r}_m + \Delta; \theta, \sigma^2) \mid \mathbf{r}_m\right] \\ &= \mathbb{E}\left[\varphi(\Delta - (\theta - \bar{r}_m); 0, \sigma^2) \mid \mathbf{r}_m\right]. \end{aligned}$$

Since the map $z \mapsto \varphi(\Delta - z; 0, \sigma^2)$ is bounded and continuous, we have that

$$p(\bar{r}_m + \Delta \mid \mathbf{r}_m) \to \mathbb{E}\left[\varphi(\Delta - Z; 0, \sigma^2)\right] \qquad \text{as } m \to \infty,$$

where the expectation is taken with respect to $Z \sim \mathcal{N}(0, \sigma^2/n)$. This gives

$$s(\bar{r}_m + \Delta, \mathbf{r}_m) = -p(\bar{r}_m + \Delta \mid \mathbf{r}_m) \to h_n(|\Delta|) \qquad \text{as } m \to \infty,$$

where

$$h_n(u) = -\frac{1}{\sqrt{2\pi\sigma^2(1 + 1/n)}}\exp\left(-\frac{u^2}{2\sigma^2(1 + 1/n)}\right), \quad u \geq 0.$$

As $h_n$ is strictly monotone increasing, the limiting score function is equivalent to the DTA score function. $\qquad\square$

**Proposition 3.2.** *The DTA nonconformity score is equivalent to the Bayes–assisted nonconformity score for the following BWM:*

$$\begin{aligned} R \mid \theta &\sim \mathcal{N}(\theta, \sigma^2), \\ \theta &\sim \pi(\theta) \propto 1 \end{aligned}$$

*where $\sigma > 0$ can take any arbitrary, fixed value.*

*Proof.* Fix $\sigma > 0$. The likelihood for the given BWM is:

$$L(\theta; \mathbf{r}_{1:n}) \propto \exp\left(-\frac{1}{2\sigma^2}\sum_{i=1}^{n}(r_i - \theta)^2\right) = \exp\left(-\frac{n}{2\sigma^2}(\theta - \bar{r}_n)^2\right),$$

Since the prior is constant, the posterior is proportional to the likelihood, giving

$$\theta \mid \mathbf{r}_{1:n} \sim \mathcal{N}\left(\bar{r}_n, \frac{\sigma^2}{n}\right).$$

The posterior predictive of our BWM is therefore:

$$p(r_{n+1} \mid \mathbf{r}_{1:n}) = \int p(r_{n+1} \mid \theta)\, p(\theta \mid \mathbf{r}_{1:n})\, d\theta = \int \mathcal{N}(r_{n+1}; \theta, \sigma^2)\mathcal{N}\left(\theta; \bar{r}_n, \frac{\sigma^2}{n}\right)d\theta.$$

This is a convolution of Gaussians, so

$$r_{n+1} \mid \mathbf{r}_{1:n} \sim \mathcal{N}\left(\bar{r}_n,\ \sigma^2 + \frac{\sigma^2}{n}\right) = \mathcal{N}\left(\bar{r}_n,\ \sigma^2\left(1 + \frac{1}{n}\right)\right).$$

The Bayes–assisted nonconformity score is therefore:

$$-p(r_{n+1} \mid \mathbf{r}_{1:n}) = -\frac{1}{\sqrt{2\pi\sigma^2(1 + 1/n)}}\exp\left(-\frac{(r_{n+1} - \bar{r}_n)^2}{2\sigma^2(1 + 1/n)}\right).$$

As a function of $|r_{n+1} - \bar{r}_n|$, the right-hand side is a strictly monotone transformation. Thus, the Bayes–assisted score for the given BWM is equivalent to the DTA score.

$\square$

### D.3. Proofs of Section 3.3

**Proposition 3.3.** *Consider BWM* (14) *with fixed* $\sigma^2, \upsilon^2 > 0$. *Then, the induced Bayes–assisted score is a strictly monotone transformation of the score*

$$s(r_{n+1}, \mathbf{r}_{1:n}) = \left|r_{n+1} - a(\sigma^2, \upsilon^2)\,\bar{r}_n\right|, \tag{15}$$

*where* $\bar{r}_n = \frac{1}{n}\sum_{i=1}^{n} r_i$ *and*

$$a(\sigma^2, \upsilon^2) = \frac{\upsilon^2}{\upsilon^2 + \sigma^2/n}. \tag{16}$$

*Proof.* Let $\mathbf{z}_{1:n+1} = \{\mathbf{z}_i\}_{i=1}^{n+1}$ and $\mathbf{r}_{1:n+1} = \{r_i\}_{i=1}^{n+1}$. The likelihood for BWM (14) is given by:

$$p(\mathbf{r}_{1:n} \mid \theta, \sigma^2) \propto \exp\left(-\frac{1}{2\sigma^2}\sum_{i=1}^{n}(r_i - \theta)^2\right) \propto_\theta \exp\left(-\frac{n}{2\sigma^2}(\theta - \bar{r}_n)^2\right),$$

so conjugacy yields the posterior

$$\theta \mid \mathbf{r}_{1:n}, \sigma^2, \upsilon^2 \sim \mathcal{N}(m_n, V_n), \qquad V_n = \left(\frac{n}{\sigma^2} + \frac{1}{\upsilon^2}\right)^{-1}, \qquad m_n = V_n \cdot \frac{n}{\sigma^2}\bar{r}_n.$$

Writing

$$a(\sigma^2, \upsilon^2) := \frac{\upsilon^2}{\upsilon^2 + \sigma^2/n},$$

we have $m_n = a(\sigma^2, \upsilon^2)\,\bar{r}_n$.

The predictive density for $r_{n+1}$ is obtained by integrating out $\theta$:

$$p(r_{n+1} \mid \mathbf{r}_{1:n}, \sigma^2, \upsilon^2) = \int \mathcal{N}(r_{n+1}; \theta, \sigma^2)\,\mathcal{N}(\theta; m_n, V_n)\,d\theta = \mathcal{N}(r_{n+1}; m_n, \sigma^2 + V_n).$$

Therefore the Bayes–assisted nonconformity score is given by:

$$-p(r_{n+1} \mid \mathbf{r}_{1:n}, \sigma^2, \upsilon^2) = -\frac{1}{\sqrt{2\pi(\sigma^2 + V_n)}}\exp\left(-\frac{(r_{n+1} - m_n)^2}{2(\sigma^2 + V_n)}\right).$$

Noting that the above is a strictly monotone transformation of $|r_{n+1} - m_n|$, we can conclude that the Bayes–assisted nonconformity score is equivalent to:

$$s(r_{n+1}, \mathbf{r}_{1:n}) = \left|r_{n+1} - m_n\right| = \left|r_{n+1} - a(\sigma^2, \upsilon^2)\,\bar{r}_n\right|.$$

$\square$

**Empirical Bayes estimates of hyperparameters.** Consider BWM (14). We derive the empirical Bayes estimates (18)–(19) for the hyperparameters $\sigma^2$ and $\upsilon^2$ based on the observed residuals $\mathbf{r}_{1:n}$.

Define the residual sum of squares about $\bar{r}_n$ by

$$S_{1:n} = \sum_{i=1}^{n} (r_i - \bar{r}_n)^2.$$

Maximising the Gaussian likelihood $\prod_{i=1}^{n} \mathcal{N}(r_i; \theta, \sigma^2)$ over $\theta$ gives $\hat{\theta} = \bar{r}_n$, and substituting back yields the usual MLE

$$\hat{\sigma}^2(r_{1:n}) = \frac{1}{n} S_{1:n} = \frac{1}{n} \sum_{i=1}^{n} (r_i - \bar{r}_n)^2.$$

Next, under the prior $\theta \sim \mathcal{N}(0, \upsilon^2)$ and conditional model, the sample mean satisfies

$$\bar{r}_n \mid \upsilon^2, \sigma^2 \sim \mathcal{N}\left(0, \upsilon^2 + \frac{\sigma^2}{n}\right).$$

Using the plug–in $\sigma^2 = \hat{\sigma}^2(\mathbf{r}_{1:n})$, the (type-II) marginal likelihood for $\upsilon^2$ is proportional to

$$\left(\upsilon^2 + \frac{\hat{\sigma}^2(\mathbf{r}_{1:n})}{n}\right)^{-1/2} \exp\left(-\frac{\bar{r}_n^2}{2\left(\upsilon^2 + \hat{\sigma}^2(\mathbf{r}_{1:n})/n\right)}\right),$$

whose maximiser over $\upsilon^2 \geq 0$ is

$$\hat{\upsilon}^2(\mathbf{r}_{1:n}) = \max\left(\bar{r}_n^2 - \frac{\hat{\sigma}^2(\mathbf{r}_{1:n})}{n}, 0\right).$$

**Remark on the plug–in empirical Bayes predictive score.** The equivalence in Proposition 3.3 relies on treating $\sigma^2$ and $\upsilon^2$ as fixed. Indeed, its proof shows that the posterior predictive variance is $\sigma^2 + V_n$, where

$$V_n = \left(\frac{n}{\sigma^2} + \frac{1}{\upsilon^2}\right)^{-1}.$$

For fixed $\sigma^2$ and $\upsilon^2$, this quantity is common across the augmented scores, and hence the Bayes–assisted score based on the posterior predictive density is equivalent to the absolute–deviation score in Proposition 3.3.

This equivalence no longer holds if one instead plugs the EB estimates directly into the posterior predictive density. To see this, define

$$\widehat{V}_n(\mathbf{r}_{1:n}) = \left(\frac{n}{\hat{\sigma}^2(\mathbf{r}_{1:n})} + \frac{1}{\hat{\upsilon}^2(\mathbf{r}_{1:n})}\right)^{-1},$$

with the usual limiting convention when $\hat{\upsilon}^2(\mathbf{r}_{1:n}) = 0$. The plug–in posterior predictive density then has mean $\hat{a}(\mathbf{r}_{1:n})\bar{r}_n$ and variance

$$\hat{\sigma}^2(\mathbf{r}_{1:n}) + \widehat{V}_n(\mathbf{r}_{1:n}).$$

Applying a strictly increasing transformation to the negative predictive density, the corresponding plug–in empirical Bayes predictive score is equivalent to

$$\frac{(r_{n+1} - \hat{a}(\mathbf{r}_{1:n})\bar{r}_n)^2}{\hat{\sigma}^2(\mathbf{r}_{1:n}) + \widehat{V}_n(\mathbf{r}_{1:n})} + \log\left(\hat{\sigma}^2(\mathbf{r}_{1:n}) + \widehat{V}_n(\mathbf{r}_{1:n})\right).$$

Equivalently, using the RoBAS–EB score (17), this can be written as

$$\frac{s(r_{n+1}, \mathbf{r}_{1:n})^2}{\hat{\sigma}^2(\mathbf{r}_{1:n}) + \widehat{V}_n(\mathbf{r}_{1:n})} + \log\left(\hat{\sigma}^2(\mathbf{r}_{1:n}) + \widehat{V}_n(\mathbf{r}_{1:n})\right).$$

Thus RoBAS–EB should be viewed as plugging EB estimates into the fixed–hyperparameter Bayes–assisted score, rather than as the exact score obtained from the full EB posterior predictive density. In the latter case, the predictive scale depends on the reference residual vector and therefore cannot, in general, be cancelled across the augmented conformal scores.

**Proposition 3.4** (Asymptotic Robustness of EB Score). *Fix $n \geq 1$. Let $(\mathbf{r}_{1:n}^{(m)})_{m \geq 1}$, with $\mathbf{r}_{1:n}^{(m)} \in \mathbb{R}^n$, be a sequence of residuals such that $|\bar{r}_n^{(m)}| \to \infty$ and $\widehat{\sigma}^2(\mathbf{r}_{1:n}^{(m)})/|\bar{r}_n^{(m)}| \to 0$ as $m \to \infty$. Then, the score* (17) *satisfies, for every fixed $\Delta \in \mathbb{R}$,*

$$s(\bar{r}_n^{(m)} + \Delta, \mathbf{r}_{1:n}^{(m)}) \to |\Delta| \quad \text{as } m \to \infty.$$

*Proof.* For notational convenience, write

$$\bar{r}_m := \bar{r}_n^{(m)}, \qquad \mathbf{r}_m := \mathbf{r}_{1:n}^{(m)}.$$

Since $|\bar{r}_m| \to \infty$ and $\widehat{\sigma}^2(\mathbf{r}_m)/|\bar{r}_m| \to 0$, we also have

$$\frac{\widehat{\sigma}^2(\mathbf{r}_m)}{n\bar{r}_m^2} \to 0 \quad \text{as } m \to \infty.$$

Hence, for $m$ sufficiently large,

$$\widehat{v}^2(\mathbf{r}_m) = \bar{r}_m^2 - \frac{\widehat{\sigma}^2(\mathbf{r}_m)}{n},$$

and therefore

$$\widehat{a}(\mathbf{r}_m) = \frac{\bar{r}_m^2 - \widehat{\sigma}^2(\mathbf{r}_m)/n}{\bar{r}_m^2} = 1 - \frac{\widehat{\sigma}^2(\mathbf{r}_m)}{n\bar{r}_m^2}.$$

Consequently,

$$(1 - \widehat{a}(\mathbf{r}_m))\bar{r}_m = \frac{\widehat{\sigma}^2(\mathbf{r}_m)}{n\bar{r}_m} \to 0 \quad \text{as } m \to \infty.$$

For any fixed $\Delta \in \mathbb{R}$, the RoBAS score (17) evaluated at $r_{n+1} = \bar{r}_m + \Delta$ is given by

$$s(\bar{r}_m + \Delta, \mathbf{r}_m) = |\bar{r}_m + \Delta - \widehat{a}(\mathbf{r}_m)\bar{r}_m| = |\Delta + (1 - \widehat{a}(\mathbf{r}_m))\bar{r}_m| \to |\Delta| \quad \text{as } m \to \infty,$$

which is the DTA score evaluated at $r_{n+1} = \bar{r}_m + \Delta$. $\qquad\square$

### D.4. Proofs of Section 3.4

**Theorem 3.5** (Interval Property of RoBAS–EB). *Assume $n \geq 4$ and consider the nonconformity score* (17). *Then, for any $\alpha \in [0, 1)$, the corresponding conformal prediction set $\mathcal{C}_\alpha$ in* (4) *is an interval, up to intersection with $\mathcal{Y}$.*

*Proof.* By Lemma D.2, it suffices to show that the pairwise regions

$$I_i = \{y \in \mathbb{R} : s_i(y) \geq s_{n+1}(y)\}, \qquad i = 1, \ldots, n+1,$$

are intervals containing a common point $y_0 \in \mathbb{R}$.

We prove this in residual coordinates. Let

$$r = y - f(\mathbf{x}_{n+1})$$

be the candidate residual corresponding to the candidate response $y$. Since $f(\mathbf{x}_{n+1})$ is fixed, the map $y \mapsto r$ is a translation. Thus, if the corresponding residual–space pairwise regions are intervals and contain a common residual value $r_0 \in \mathbb{R}$, then the response–space regions $I_i$ are intervals containing the corresponding translated common point $y_0 = r_0 + f(\mathbf{x}_{n+1})$.

In particular, let $s : \mathbb{R} \times \mathbb{R}^n \to \mathbb{R}$ denote the RoBAS residual–based nonconformity score (17). Then, as in Equation (6), the nonconformity scores associated with a candidate response $y$ reduce to

$$s_i(y) = s(r_i, \mathbf{r}_{1:n,-i} \cup \{y - f(\mathbf{x}_{n+1})\}), \qquad i = 1, \ldots, n,$$
$$s_{n+1}(y) = s(y - f(\mathbf{x}_{n+1}), \mathbf{r}_{1:n}),$$

where $r_i = y_i - f(\mathbf{x}_i)$ and $\mathbf{r}_{1:n,-i} = \mathbf{r}_{1:n} \setminus \{r_i\}$. With a slight abuse of notation, we write the same scores as functions of the candidate residual $r$:

$$s_i(r) := s(r_i, \mathbf{r}_{1:n,-i} \cup \{r\}), \qquad i = 1, \ldots, n,$$
$$s_{n+1}(r) := s(r, \mathbf{r}_{1:n}).$$

Define the residual–space pairwise regions

$$J_i = \{r \in \mathbb{R} : s_i(r) \ge s_{n+1}(r)\}, \qquad i = 1, \ldots, n+1.$$

Then,

$$I_i = \{f(\mathbf{x}_{n+1}) + r : r \in J_i\}, \qquad i = 1, \ldots, n+1.$$

Hence, it is enough to show that each $J_i$ is an interval and that all the sets $J_i$ contain a common residual value $r_0 \in \mathbb{R}$.

For any residual vector $\mathbf{v}_{1:n} \in \mathbb{R}^n$, the RoBAS centre

$$m(\mathbf{v}_{1:n}) = \widehat{a}(\mathbf{v}_{1:n})\bar{\mathbf{v}}_{1:n} = a(\widehat{\sigma}^2(\mathbf{v}_{1:n}), \widehat{v}^2(\mathbf{v}_{1:n}))\bar{\mathbf{v}}_{1:n}$$

depends on $\mathbf{v}_{1:n}$ only through its empirical mean and variance, and is thus invariant to permutations of $\mathbf{v}_{1:n}$. We therefore also write

$$m(A) = \widehat{a}(A)\bar{A} = a(\widehat{\sigma}^2(A), \widehat{v}^2(A))\bar{A},$$

for an $n$–element residual multiset $A$ to mean $m(\mathbf{v}_{1:n})$ for any ordering $\mathbf{v}_{1:n}$ of the elements of $A$.

Let $r_0 = m(\mathbf{r}_{1:n})$. Then, the test score is given by $s_{n+1}(r) = |r - r_0|$. For $i = 1, \ldots, n$, define the leave–one–out augmented residual multiset

$$A_i(r) = \mathbf{r}_{1:n,-i} \cup \{r\},$$

and write

$$h_i(r) = m(A_i(r)).$$

The corresponding calibration score is given by

$$s_i(r) = |r_i - h_i(r)|.$$

We first show that each map $h_i$ is a contraction in $r$. Fix $i \in \{1, \ldots, n\}$ and write the fixed residuals in $\mathbf{r}_{1:n,-i}$ as $v_1, \ldots, v_{n-1}$. Let

$$S = \sum_{j=1}^{n-1} v_j, \qquad Q = \sum_{j=1}^{n-1} v_j^2, \qquad u(r) = S + r,$$

and define the multiset

$$B(r) = \{v_1, \ldots, v_{n-1}, r\}.$$

Then,

$$\bar{B}(r) = \frac{u(r)}{n}, \qquad \widehat{\sigma}^2(B(r)) = \frac{Q + r^2}{n} - \frac{u(r)^2}{n^2}.$$

The positive part in $\widehat{v}^2(B(r))$ is active if and only if

$$\bar{B}(r)^2 > \frac{\widehat{\sigma}^2(B(r))}{n}.$$

or, equivalently, if and only if

$$D(r) > 0, \quad \text{where} \quad D(r) = (n+1)u(r)^2 - n(Q + r^2).$$

When $D(r) \le 0$, the RoBAS centre is zero. We therefore call $D(r) > 0$ the active region and $D(r) \le 0$ the inactive region.

Consider the active region $D(r) > 0$. There, the RoBAS centre is given by

$$m(B(r)) = \bar{B}(r) - \frac{\widehat{\sigma}^2(B(r))}{n\bar{B}(r)} = \frac{D(r)}{n^2 u(r)},$$

which is well–defined since $u(r) = 0$ implies $D(r) = -n(Q + r^2) \le 0$. Differentiating with respect to $r$ gives

$$\frac{d}{dr}m(B(r)) = \frac{u(r)^2 + n(Q + S^2)}{n^2 u(r)^2}.$$

Set

$$t = \frac{S}{u(r)}, \qquad w = \frac{Q}{u(r)^2}.$$

Then,

$$\frac{d}{dr} m(B(r)) = \frac{1}{n^2} + \frac{w + t^2}{n}. \tag{20}$$

By the Cauchy–Schwarz inequality,

$$Q \geq \frac{S^2}{n-1},$$

and hence

$$w \geq \frac{t^2}{n-1}. \tag{21}$$

On the other hand, the active region condition $D(r) > 0$ can be rewritten as

$$D(r) = u(r)^2 + n(S^2 + 2S(u(r) - S) - Q) = u(r)^2 + nu(r)^2(-t^2 + 2t - w) > 0,$$

which implies

$$w < \frac{1}{n} + 2t - t^2. \tag{22}$$

Combining inequalities (21)–(22) yields

$$\frac{t^2}{n-1} < \frac{1}{n} + 2t - t^2,$$

which in turn implies that $t$ is smaller than the larger root of the corresponding quadratic, namely

$$t < t_+ := \frac{n-1}{n} \left( 1 + \sqrt{\frac{n}{n-1}} \right). \tag{23}$$

Applying bounds (22)–(23) to the derivative (20) gives

$$\frac{d}{dr} m(B(r)) \overset{(22)}{<} \frac{1}{n^2} + \frac{1/n + 2t}{n} \overset{(23)}{<} \frac{2}{n^2} + \frac{2t_+}{n} = \frac{2\left(n + \sqrt{n(n-1)}\right)}{n^2} =: L_n.$$

and, for $n \geq 4$, we have $L_n < 1$.

The function $D(r)$ is a quadratic polynomial in $r$, and so the active and inactive regions form a finite partition of $\mathbb{R}$. On the inactive region $D(r) \leq 0$, $m(B(r)) = 0$ and thus $\frac{d}{dr} m(B(r)) = 0$ for $D(r) < 0$. Moreover, $m(B(r))$ is continuous across the boundaries $D(r) = 0$. Applying the mean-value theorem piecewise gives

$$|m(B(r_2)) - m(B(r_1))| \leq L_n |r_2 - r_1|, \qquad r_1, r_2 \in \mathbb{R}.$$

Consequently, each leave–one–out RoBAS centre

$$h_i(r) = m(A_i(r))$$

is $L_n$-Lipschitz, with $L_n < 1$.

Now define

$$F_i(r) = s_i(r) - s_{n+1}(r) = |r_i - h_i(r)| - |r - r_0|.$$

Since $h_i$ is $L_n$-Lipschitz and the absolute value map is one–Lipschitz, the map

$$r \mapsto |r_i - h_i(r)|$$

is also $L_n$-Lipschitz. Let $r' < r''$ with $r', r'' \geq r_0$, then

$$
\begin{aligned}
F_i(r'') - F_i(r') &= (|r_i - h_i(r'')| - |r_i - h_i(r')|) - (|r'' - r_0| - |r' - r_0|) \\
&= (|r_i - h_i(r'')| - |r_i - h_i(r')|) - (r'' - r') \\
&\leq L_n(r'' - r') - (r'' - r') \\
&= -(1 - L_n)(r'' - r') < 0.
\end{aligned}
$$

Thus, $F_i$ is strictly decreasing on $[r_0, \infty)$. Similarly, if $r' < r''$ with $r', r'' \leq r_0$, then

$$
\begin{aligned}
F_i(r'') - F_i(r') &= (|r_i - h_i(r'')| - |r_i - h_i(r')|) - \{|r'' - r_0| - |r' - r_0|\} \\
&= \{|r_i - h_i(r'')| - |r_i - h_i(r')|\} + (r'' - r') \\
&\geq -L_n(r'' - r') + (r'' - r') \\
&= (1 - L_n)(r'' - r') > 0.
\end{aligned}
$$

Therefore $F_i$ is strictly increasing on $(-\infty, r_0]$. At $r = r_0$, the test score is zero,

$$
s_{n+1}(r_0) = |r_0 - r_0| = 0,
$$

and hence

$$
F_i(r_0) = s_i(r_0) - s_{n+1}(r_0) = s_i(r_0) \geq 0.
$$

Since $F_i$ is increasing on $(-\infty, r_0]$ and decreasing on $[r_0, \infty)$, for each $i = 1, \ldots, n$, the upper level set

$$
J_i = \{r \in \mathbb{R} : s_i(r) \geq s_{n+1}(r)\} = \{r \in \mathbb{R} : F_i(r) \geq 0\}
$$

is an interval containing $r_0$.

For $i = n + 1$,

$$
J_{n+1} = \{r \in \mathbb{R} : s_{n+1}(r) \geq s_{n+1}(r)\} = \mathbb{R},
$$

which is also an interval containing $r_0$.

Thus, all residual–space pairwise regions $J_1, \ldots, J_{n+1}$ are intervals containing the common residual value $r_0$, completing the proof. □

### D.5. Additional Results

**Theorem D.4** (Asymptotic robustness of heavy–tailed BWM with estimated variance). *Fix $n \geq 2$. Let $(\mathbf{r}_{1:n}^{(m)})_{m \geq 1}$, with $\mathbf{r}_{1:n}^{(m)} \in \mathbb{R}^n$, be a sequence of residual vectors such that*

$$
|\bar{r}_n^{(m)}| \to \infty \qquad \text{as } m \to \infty.
$$

*Define*

$$
\widehat{\varsigma}_m^2 = \frac{1}{n} \sum_{i=1}^n \left(r_i^{(m)} - \bar{r}_n^{(m)}\right)^2,
$$

*and suppose that*

$$
\widehat{\varsigma}_m^2 \to \varsigma^2 \in (0, \infty) \qquad \text{as } m \to \infty.
$$

*For fixed $\Delta \in \mathbb{R}$, set*

$$
r_{n+1}^{(m)} = \bar{r}_n^{(m)} + \Delta,
$$

*and define the augmented plug-in variance*

$$
\widehat{\sigma}_{m,\Delta}^2 = \frac{1}{n} \sum_{i=1}^{n+1} \left(r_i^{(m)} - \bar{r}_{n+1}^{(m)}\right)^2, \qquad \bar{r}_{n+1}^{(m)} = \frac{1}{n+1} \sum_{i=1}^{n+1} r_i^{(m)}.
$$

*Under BWM (12), with $\widehat{\sigma}_{m,\Delta}^2$ plugged in for $\sigma^2$, assume*

$$
\theta \mid \tau^2 \sim \mathcal{N}(0, \gamma \tau^2), \qquad g_{\tau^2}(w) \sim C w^{-\delta}, \quad w \to \infty,
$$

*for some fixed $\gamma > 0$, $C > 0$, and $\delta > 1$. Then, for every fixed $\Delta \in \mathbb{R}$, the score function (10) satisfies*

$$
s(\bar{r}_n^{(m)} + \Delta, \mathbf{r}_{1:n}^{(m)}) = -p_{\widehat{\sigma}_{m,\Delta}^2}(\bar{r}_n^{(m)} + \Delta \mid \mathbf{r}_{1:n}^{(m)}) \to h_{n,\varsigma}(|\Delta|)
$$

*as $m \to \infty$, where $h_{n,\varsigma} : [0, \infty) \to \mathbb{R}$ is the strictly increasing function*

$$
h_{n,\varsigma}(u) = -\frac{1}{\sqrt{2\pi(1 + 1/n)\left(\varsigma^2 + \frac{u^2}{n+1}\right)}} \exp\left(-\frac{u^2}{2(1 + 1/n)\left(\varsigma^2 + \frac{u^2}{n+1}\right)}\right).
$$

*Proof.* Fix $\Delta \in \mathbb{R}$. For notational convenience, write

$$\bar{r}_m := \bar{r}_n^{(m)}, \qquad \mathbf{r}_m := \mathbf{r}_{1:n}^{(m)}.$$

We first decompose the augmented plug–in variance. Since

$$\bar{r}_{n+1}^{(m)} = \frac{1}{n+1} \left( \sum_{i=1}^{n} r_i^{(m)} + \bar{r}_m + \Delta \right) = \bar{r}_m + \frac{\Delta}{n+1},$$

and since $\sum_{i=1}^{n} (r_i^{(m)} - \bar{r}_m) = 0$,

$$
\begin{aligned}
\widehat{\sigma}_{m,\Delta}^2 &= \frac{1}{n} \sum_{i=1}^{n} \left( r_i^{(m)} - \bar{r}_m - \frac{\Delta}{n+1} \right)^2 + \frac{1}{n} \left( \Delta - \frac{\Delta}{n+1} \right)^2 \\
&= \frac{1}{n} \sum_{i=1}^{n} \left( r_i^{(m)} - \bar{r}_m \right)^2 + \frac{1}{n} \cdot \frac{n\Delta^2}{(n+1)^2} + \frac{1}{n} \cdot \frac{n^2 \Delta^2}{(n+1)^2} \\
&= \widehat{\varsigma}_m^2 + \frac{\Delta^2}{n+1}.
\end{aligned}
$$

Hence

$$\widehat{\sigma}_{m,\Delta}^2 \to \varsigma_\Delta^2 := \varsigma^2 + \frac{\Delta^2}{n+1} \in (0, \infty).$$

Now consider BWM (12) with $\widehat{\sigma}_{m,\Delta}^2$ plugged in for $\sigma^2$. Conditional on this plug-in value, the likelihood of $\mathbf{r}_m$ depends on $\theta$ only through $\bar{r}_m$; the remaining within–sample factor is independent of $\theta$. The conditional model for $\bar{r}_m$ is therefore

$$\bar{r}_m \mid \theta \sim \mathcal{N}\left( \theta, \frac{\widehat{\sigma}_{m,\Delta}^2}{n} \right).$$

Write this one-dimensional Gaussian exponential family with natural parameter

$$\eta_m = \frac{n}{\widehat{\sigma}_{m,\Delta}^2} \theta.$$

For $\xi \in \mathbb{R}$, Tweedie's formula for exponential families (Efron, 2011, §2) gives

$$\kappa_m(\xi) := \log \mathbb{E}\left[ e^{\eta_m \xi} \mid \mathbf{r}_m \right] = \lambda_m(\bar{r}_m + \xi) - \lambda_m(\bar{r}_m),$$

where

$$\lambda_m(z) = \log \frac{f_m(z)}{f_{0,m}(z)},$$

$$f_{0,m}(z) = \varphi\left( z; 0, \frac{\widehat{\sigma}_{m,\Delta}^2}{n} \right),$$

and

$$f_m(z) = \int_0^\infty \varphi\left( z; 0, \frac{\widehat{\sigma}_{m,\Delta}^2}{n} + \gamma w \right) g_{\tau^2}(w) \, dw.$$

Define the centred natural parameter

$$\eta_m^\star = \eta_m - \frac{n}{\widehat{\sigma}_{m,\Delta}^2} \bar{r}_m.$$

For every fixed $\xi \in \mathbb{R}$,

$$
\begin{aligned}
\log \mathbb{E}\left[ e^{\eta_m^\star \xi} \mid \mathbf{r}_m \right] &= \kappa_m(\xi) - \frac{n}{\widehat{\sigma}_{m,\Delta}^2} \bar{r}_m \xi \\
&= \log \frac{f_m(\bar{r}_m + \xi)}{f_m(\bar{r}_m)} - \log \frac{f_{0,m}(\bar{r}_m + \xi)}{f_{0,m}(\bar{r}_m)} - \frac{n}{\widehat{\sigma}_{m,\Delta}^2} \bar{r}_m \xi \\
&= \log \frac{f_m(\bar{r}_m + \xi)}{f_m(\bar{r}_m)} + \frac{n}{2\widehat{\sigma}_{m,\Delta}^2} \xi^2.
\end{aligned}
$$

The final equality follows from the explicit Gaussian form of $f_{0,m}$.

We next show that the first term on the right–hand side vanishes. Since $\widehat{\sigma}^2_{m,\Delta} \to \varsigma^2_\Delta \in (0,\infty)$, there exists a compact set $K_\Delta \subset (0,\infty)$ such that

$$\frac{\widehat{\sigma}^2_{m,\Delta}}{n} \in K_\Delta$$

for all sufficiently large $m$. Applying Lemma D.3 with this compact set and with any bounded set containing the fixed value $\xi$ gives

$$\frac{f_m(\bar{r}_m + \xi)}{f_m(\bar{r}_m)} \to 1,$$

because $|\bar{r}_m| \to \infty$. Therefore,

$$\log \mathbb{E}\Big[e^{\eta^\star_m \xi} \mid \mathbf{r}_m\Big] \to \frac{n}{2\varsigma^2_\Delta}\xi^2 \qquad \text{for every } \xi \in \mathbb{R}.$$

The limiting cumulant generating function is that of $\mathcal{N}(0, n/\varsigma^2_\Delta)$. By the convergence theorem for moment–generating functions,

$$\eta^\star_m \mid \mathbf{r}_m \Rightarrow \mathcal{N}\left(0, \frac{n}{\varsigma^2_\Delta}\right).$$

Since

$$\theta - \bar{r}_m = \frac{\widehat{\sigma}^2_{m,\Delta}}{n}\eta^\star_m$$

and $\widehat{\sigma}^2_{m,\Delta} \to \varsigma^2_\Delta$, Slutsky's theorem gives

$$\theta - \bar{r}_m \mid \mathbf{r}_m \Rightarrow \mathcal{N}\left(0, \frac{\varsigma^2_\Delta}{n}\right).$$

Let

$$Z_\Delta \sim \mathcal{N}\left(0, \frac{\varsigma^2_\Delta}{n}\right).$$

The posterior predictive density at $\bar{r}_m + \Delta$ is

$$p_{\widehat{\sigma}^2_{m,\Delta}}(\bar{r}_m + \Delta \mid \mathbf{r}_m) = \mathbb{E}\big[\varphi(\bar{r}_m + \Delta; \theta, \widehat{\sigma}^2_{m,\Delta}) \mid \mathbf{r}_m\big]$$
$$= \mathbb{E}\big[\varphi(\Delta - (\theta - \bar{r}_m); 0, \widehat{\sigma}^2_{m,\Delta}) \mid \mathbf{r}_m\big].$$

Since $\widehat{\sigma}^2_{m,\Delta} \to \varsigma^2_\Delta \in (0,\infty)$,

$$\sup_{z \in \mathbb{R}} \big|\varphi(\Delta - z; 0, \widehat{\sigma}^2_{m,\Delta}) - \varphi(\Delta - z; 0, \varsigma^2_\Delta)\big| \to 0.$$

Moreover, the map

$$z \mapsto \varphi(\Delta - z; 0, \varsigma^2_\Delta)$$

is bounded and continuous, and

$$\theta - \bar{r}_m \mid \mathbf{r}_m \Rightarrow Z_\Delta.$$

Hence

$$p_{\widehat{\sigma}^2_{m,\Delta}}(\bar{r}_m + \Delta \mid \mathbf{r}_m) \to \mathbb{E}\big[\varphi(\Delta - Z_\Delta; 0, \varsigma^2_\Delta)\big].$$

The right–hand side is a Gaussian convolution:

$$\mathbb{E}\big[\varphi(\Delta - Z_\Delta; 0, \varsigma^2_\Delta)\big] = \varphi\left(\Delta; 0, \varsigma^2_\Delta\left(1 + \frac{1}{n}\right)\right).$$

Using

$$\varsigma^2_\Delta = \varsigma^2 + \frac{\Delta^2}{n+1},$$

we obtain

$$s(\bar{r}_m + \Delta, \mathbf{r}_m) = -p_{\widehat{\sigma}^2_{m,\Delta}}(\bar{r}_m + \Delta \mid \mathbf{r}_m) \to h_{n,\varsigma}(|\Delta|),$$

where

$$h_{n,\varsigma}(u) = -\frac{1}{\sqrt{2\pi(1+1/n)\left(\varsigma^2 + \frac{u^2}{n+1}\right)}} \exp\left(-\frac{u^2}{2(1+1/n)\left(\varsigma^2 + \frac{u^2}{n+1}\right)}\right).$$

It remains to prove that $h_{n,\varsigma}$ is strictly increasing on $[0, \infty)$. Let

$$c_n = 1 + \frac{1}{n}, \qquad A(u) = \varsigma^2 + \frac{u^2}{n+1},$$

and write

$$h_{n,\varsigma}(u) = -q(u),$$

where

$$q(u) = \frac{1}{\sqrt{2\pi c_n A(u)}} \exp\left(-\frac{u^2}{2c_n A(u)}\right).$$

For $u > 0$,

$$\frac{d}{du}\log q(u) = -\frac{u}{(n+1)A(u)} - \frac{u\varsigma^2}{c_n A(u)^2} < 0.$$

Thus $q$ is strictly decreasing on $(0, \infty)$, and therefore $h_{n,\varsigma} = -q$ is strictly increasing on $(0, \infty)$. Since $h_{n,\varsigma}$ is continuous at 0, it is strictly increasing on $[0, \infty)$. $\square$

# E. Additional Details

## E.1. Algorithm Blocks

Algorithms 2 and 3 provide the procedure for finding the conformal prediction intervals for RoBAS–Full and RoBAS–EB, respectively.

---

**Algorithm 2** RoBAS–Full conformal prediction interval

---

**Require:** Calibration data $\mathbf{z}_{1:n} = \{\mathbf{z}_i\}_{i=1}^n$, predictor $f$, test covariate $\mathbf{x}_{n+1}$, level $\alpha \in [0, 1)$.
**Ensure:** Interval $[l, u]$

1: **function** $\tilde{s}^{\text{RoBAS–Full}}(\mathbf{z}_{n+1}, \mathbf{z}_{1:n})$
2:     $r_i \leftarrow y_i - f(\mathbf{x}_i)$ for $i = 1, \dots, n+1$
3:     $\bar{r}_n \leftarrow \frac{1}{n} \sum_{i=1}^n r_i$;    $s_n^2 \leftarrow \frac{1}{n} \sum_{i=1}^n (r_i - \bar{r}_n)^2$
4:     $\bar{r}_{n+1} \leftarrow \frac{1}{n+1} \sum_{i=1}^{n+1} r_i$;    $\hat{\sigma}^2 \leftarrow \frac{1}{n} \sum_{i=1}^{n+1} (r_i - \bar{r}_{n+1})^2$
5:     **return** $\exp\left(-\frac{n s_n^2}{2\hat{\sigma}^2}\right) \cdot {}_1F_1\left(1; \frac{3}{2}; -\frac{n \bar{r}_n^2}{2\hat{\sigma}^2}\right)$
6: **end function**
7: $[l, u] \leftarrow$ **Algorithm** 1$(\mathbf{z}_{1:n}, \mathbf{x}_{n+1}, s^{\text{RoBAS–Full}}, f, \alpha)$
8: **return** $[l, u]$

---

---

**Algorithm 3** RoBAS–EB conformal prediction interval

---

**Require:** Calibration data $\mathbf{z}_{1:n} = \{\mathbf{z}_i\}_{i=1}^n$, predictor $f$, test covariate $\mathbf{x}_{n+1}$, level $\alpha \in [0, 1)$.
**Ensure:** Interval $[l, u]$

1: **function** $\tilde{s}^{\text{RoBAS–EB}}(\mathbf{z}_{n+1}, \mathbf{z}_{1:n})$
2:     $r_i \leftarrow y_i - f(\mathbf{x}_i)$ for $i = 1, \dots, n+1$
3:     $\bar{r}_n \leftarrow \frac{1}{n} \sum_{i=1}^n r_i$;    $\hat{\sigma}^2 \leftarrow \frac{1}{n} \sum_{i=1}^n (r_i - \bar{r}_n)^2$
4:     $\hat{v}^2 \leftarrow \max\{\bar{r}_n^2 - \hat{\sigma}^2/n, 0\}$;    $a \leftarrow \hat{v}^2/(\hat{v}^2 + \hat{\sigma}^2/n)$
5:     **return** $|r_{n+1} - a\bar{r}_n|$
6: **end function**
7: $[l, u] \leftarrow$ **Algorithm** 1$(\mathbf{z}_{1:n}, \mathbf{x}_{n+1}, s^{\text{RoBAS–EB}}, f, \alpha)$
8: **return** $[l, u]$

---

## E.2. Additional Details on Variants of Conformal Prediction

Here, we provide further details on the different variants of conformal prediction and clarify how our approach relates to them. We use the same setup and notation as §2.

**Full conformal prediction.**    *Full conformal prediction* (full CP, Vovk et al. (2005)) determines whether a candidate $y \in \mathcal{Y}$ belongs to $\mathcal{C}_\alpha(\mathbf{x}_{n+1}; \mathbf{z}_{1:n})$ by first computing the set of nonconformity scores $\{s_i(y)\}_{i=1}^{n+1}$:

$$s_i(y) = \tilde{s}(\mathbf{z}_i, \mathbf{z}_{1:n,-i} \cup \{(\mathbf{x}_{n+1}, y)\}), \quad i = 1, \dots, n$$
$$s_{n+1}(y) = \tilde{s}((\mathbf{x}_{n+1}, y), \mathbf{z}_{1:n}),$$

where $\mathbf{z}_{1:n,-i} = \mathbf{z}_{1:n} \setminus \{\mathbf{z}_i\}$. Secondly, a hypothesis test is used to test whether $s_{n+1}(y)$ is compatible with $\{s_i(y)\}_{i=1}^n$ by comparing its rank with the rank of the remaining scores. The *conformal p-value* for this test is given by

$$\rho(y) = \frac{1}{n+1} \sum_{i=1}^{n+1} \mathbb{I}\{s_i(y) \geq s_{n+1}(y)\}, \tag{24}$$

and $y$ is accepted if $\rho(y) > \alpha$. The full prediction set is therefore given by:

$$C_\alpha(\mathbf{x}_{n+1}; \mathbf{z}_{1:n}) = \{y \in \mathcal{Y} \mid \rho(y) > \alpha\}. \tag{25}$$

As the random variables $(\mathbf{Z}_i)_{i=1}^{n+1}$ form an exchangeable sequence, the associated random nonconformity scores $(S_i)_{i=1}^{n+1}$ are also exchangeable, providing the interval in (25) with the desired guarantee in (1).

In practice, the nonconformity scores are often based on the residuals $r_i = y_i - f(\mathbf{x}_i)$ of a predictive model $f : \mathcal{X} \to \mathcal{Y}$, where, in full CP, the predictive model is trained in a leave–one–out fashion for each nonconformity score.

**Example 1.** *The Distance–To–Origin (DTO) nonconformity score is given by:*

$$s^{\mathrm{DTO}}(\mathbf{z}_{n+1}, \mathbf{z}_{1:n}) = |r_{n+1}| = |y_{n+1} - f_{\mathbf{z}_{1:n}}(\mathbf{x}_{n+1})|,$$

*where $f_{\mathbf{z}_{1:n}}$ denotes that $f$ has been trained on $\mathbf{z}_{1:n}$.*

**Remark E.1.** *Note that alternative formulations of full CP exist, although this is the original formulation as in* Vovk et al. *(2005). One popular formulation is based on the following definition of the nonconformity scores:*

$$s_i(y) = \tilde{s}(\mathbf{z}_i, \mathbf{z}_{1:n+1}^y), \ \ i = 1, \ldots, n$$
$$s_{n+1}(y) = \tilde{s}((\mathbf{x}_{n+1}, y), \mathbf{z}_{1:n+1}^y),$$

*where $\mathbf{z}_{1:n+1}^y = \mathbf{z}_{1:n} \cup \{(\mathbf{x}_{n+1}, y)\}$.*

**Split conformal prediction.** As full CP typically involves retraining a model $n + 1$ times for each candidate $y$, a more computationally efficient alternative called *split conformal prediction* (split CP) is often used. Split CP first partitions the dataset $\mathbf{z}_{1:n}$ into two subsets of size $m$ and $k$, with $m + k = n$, called the *training* and *calibration* set:

$$\mathbf{z}_{1:n} = \mathbf{z}_{1:m}^{\mathrm{train}} \cup \mathbf{z}_{1:k}^{\mathrm{cal}}, \quad \text{with } \mathbf{z}_{1:m}^{\mathrm{train}} \cap \mathbf{z}_{1:k}^{\mathrm{cal}} = \emptyset,$$

where $\mathbf{z}_{1:m}^{\mathrm{train}} = \{\mathbf{z}_i^{\mathrm{train}}\}_{i=1}^m$ and $\mathbf{z}_{1:k}^{\mathrm{cal}} = \{\mathbf{z}_i^{\mathrm{cal}}\}_{i=1}^k$.

$\mathbf{z}_{1:m}^{\mathrm{train}}$ is used exclusively to train a fixed predictive model $f_{\mathbf{z}_{1:m}^{\mathrm{train}}}$, while $\mathbf{z}_{1:k}^{\mathrm{cal}}$ is used exclusively to compute nonconformity scores. The nonconformity scores are given by:

$$s_i(y) = \tilde{s}(\mathbf{z}_i^{\mathrm{cal}}, \mathbf{z}_{1:m}^{\mathrm{train}}), \ \ i = 1, \ldots, k$$
$$s_{k+1}(y) = \tilde{s}((\mathbf{x}_{k+1}, y), \mathbf{z}_{1:m}^{\mathrm{train}}),$$

The remainder of the procedure is the same as full CP.

As the training and calibration sets are disjoint, and $(\mathbf{Z})_{i=1}^{k+1}$ is exchangeable, the computed scores $(S_i)_{i=1}^{k+1}$ are also exchangeable (conditioned on the trained model). This exchangeability preserves the validity of the coverage guarantee in (1).

**Example 2.** *The split version of the DTO nonconformity score is given by:*

$$s^{\mathrm{DTO}}(\mathbf{z}_{n+1}, \mathbf{z}_{1:m}^{train}) = |r_{n+1}| = |y_{n+1} - f_{\mathbf{z}_{1:m}^{train}}(\mathbf{x}_{n+1})|,$$

**Full Bayes–assisted conformal prediction.** Full Bayes–assisted conformal prediction follows the same procedure as full CP, with the difference being in the way the nonconformity scores are defined. More specifically, a Bayesian working model (BWM) is specified for the conditional data–generating process and the nonconformity score is taken to be the negative posterior predictive density of this model:

$$\tilde{s}(\mathbf{z}_{n+1}, \mathbf{z}_{1:n}) = -p(y_{n+1}|\mathbf{x}_{n+1}, \mathbf{z}_{1:n})$$
$$= -\int p(y_{n+1}|\mathbf{x}_{n+1}, \theta)p(\theta|\mathbf{z}_{1:n}) \ d\theta,$$

where $\theta$ denotes the parameters of the BWM. This gives nonconformity scores:

$$s_i(y) = -p(y_i|\mathbf{x}_i, \mathbf{z}_{1:n,-i} \cup \{(\mathbf{x}_{n+1}, y)\}), \ \ i = 1, \ldots, n$$
$$s_{n+1}(y) = -p(y|\mathbf{x}_{n+1}, \mathbf{z}_{1:n}).$$

Exchangeability is again preserved like in full CP, thus giving us the desired frequentist guarantee in (1). An example is given below.

**Example 3.** *Consider a Bayesian linear regression BWM with Gaussian noise:*

$$y_i = \mathbf{x}_i^\top \beta + \varepsilon, \qquad i = 1, \ldots, n,$$

*where* $\varepsilon \sim \mathcal{N}(0, \sigma^2)$, $\beta \in \mathbb{R}^d$ *denotes the regression coefficients and* $\sigma^2 > 0$ *the noise variance. We place a prior distribution on the parameters* $\theta = (\beta, \sigma^2)$, *for example a Gaussian prior on* $\beta$ *and an inverse-gamma prior on* $\sigma^2$.

*The Bayes–assisted nonconformity score for this BWM is given by:*

$$\tilde{s}(\mathbf{z}_{n+1}, \mathbf{z}_{1:n}) = -p(y_{n+1} \mid \mathbf{x}_{n+1}, \mathbf{z}_{1:n}),$$

*where*

$$p(y_{n+1} \mid \mathbf{x}_{n+1}, \mathbf{z}_{1:n}) = \int p(y_{n+1} \mid \mathbf{x}_{n+1}, \beta, \sigma^2) \, p(\beta, \sigma^2 \mid \mathbf{z}_{1:n}) \, d\beta \, d\sigma^2.$$

*In general, the posterior predictive distribution does not admit a closed–form expression. Consequently, the integral above is approximated using Monte Carlo methods, such as MCMC, by drawing samples* $\{\theta^{(s)}\}_{s=1}^S$ *from the posterior* $p(\theta \mid \mathbf{z}_{1:n})$ *and estimating*

$$p(y_{n+1} \mid \mathbf{x}_{n+1}, \mathbf{z}_{1:n}) \approx \frac{1}{S} \sum_{s=1}^S p(y_{n+1} \mid \mathbf{x}_{n+1}, \theta^{(s)}).$$

**Split Bayes–assisted conformal prediction:** Despite split CP being a popular and efficient alternative to standard full CP, to the best of our knowledge, the only work that has considered the split variant of Bayes–assisted CP is Deliu & Liseo (2025). There, the authors describe a range of nonconformity scores suitable for this framework. One strategy is to define the nonconformity score function as the negative posterior predictive density, with the posterior *conditioned on the training set*.

Using the same data split defined earlier, the nonconformity score function is:

$$\tilde{s}\left(\mathbf{z}_{n+1}, \mathbf{z}_{1:m}^{\text{train}}\right) = -p\left(y_{n+1} \mid \mathbf{x}_{n+1}, \mathbf{z}_{1:m}^{\text{train}}\right)$$
$$= -\int p(y_{n+1} \mid \mathbf{x}_{n+1}, \theta) p\left(\theta \mid \mathbf{z}_{1:m}^{\text{train}}\right) \, d\theta.$$

This gives the following nonconformity scores:

$$s_i(y) = -p(y_i^{\text{cal}} \mid \mathbf{x}_i^{\text{cal}}, \mathbf{z}_{1:m}^{\text{train}}), \quad i = 1, \ldots, k$$
$$s_{k+1}(y) = -p(y \mid \mathbf{x}_{k+1}, \mathbf{z}_{1:m}^{\text{train}}),$$

where exchangeability is again preserved like in the standard split CP case, ensuring the validity of the frequentist guarantee in (1). The key difference here is that the posterior is only fit on the training set, while the calibration set is used to find the nonconformity scores.

**Where does our approach fit in?** Our approach represents a middle ground between standard and Bayes–assisted conformal prediction. While we adopt the Bayes–assisted strategy of using a BWM to define nonconformity scores, we apply this model *solely* to the residuals of some predictor. This decoupling is crucial: whereas prior Bayes–assisted methods are restricted to strictly Bayesian models, our approach imposes no such constraint, allowing $f : \mathcal{X} \to \mathcal{Y}$ to be *any* predictive model. Moreover, our approach can also be considered as a middle ground between split and full CP. This is because we treat our predictor $f$ as fixed, like in the split CP setting, while carrying out full CP on the calibration set conditioned on $f$.

### E.3. Additional Details on BWM (12)

Here, we provide the derivation and motivation for the RoBAS–Full nonconformity score given by (13).

Let $\mathbf{r}_{1:n} = \{r_i\}_{i=1}^n$ denote the residuals of data $\mathbf{z}_{1:n}$ for some fixed predictor $f$. Consider the following BWM from §3.2:

$$R \mid \theta \sim \mathcal{N}(\theta, \sigma^2), \tag{26}$$
$$\theta \mid \tau^2 \sim \mathcal{N}(0, \gamma \tau^2), \tag{27}$$
$$\tau^2 \sim g_{\tau^2}(\tau^2), \tag{28}$$

where $\tau^2 > 0$, $\sigma^2$ and $\gamma$ are fixed hyperparameters and $g_{\tau^2}$ is a heavy–tailed prior on $\tau^2$. Specifically, we assume $g_{\tau^2}$ is regularly varying at infinity, i.e. $g_{\tau^2}(\tau^2) \sim C(\tau^2)^{-\delta}$ for some $C > 0, \delta > 1$ as $\tau^2 \to \infty$.

**Choice of $g_{\tau^2}$.** A convenient choice for $g_{\tau^2}$ that yields a simple expression for the marginal likelihood is a beta prime density for $\tau^2$:

$$g_{\tau^2}(t) = \frac{\Gamma(a+b)}{\Gamma(a)\Gamma(b)}\, t^{b-1}(1+t)^{-(a+b)}, \qquad t > 0, \quad a > 0,\ b > 0. \tag{29}$$

This prior is *regularly varying* at infinity:

$$g_{\tau^2}(t) \ \asymp\ t^{-(a+1)} \quad \text{as } t \to \infty,$$

so it matches our heavy–tail requirement.

**Choice of $\gamma$.** We take $\gamma = \frac{\sigma^2}{n}$. This choice is natural for two reasons: (i) it matches the scale of the sampling variance of the sample mean under (26), since $\bar{r}_n \mid \theta \sim \mathcal{N}(\theta, \sigma^2/n)$; (ii) it produces an exact cancellation of normalising constants after integrating out $\theta$, reducing the marginal likelihood to a single one–dimensional integral with a known form. See also Piironen & Vehtari (2017) for further discussion on this choice.

**Nonconformity score.** Below, we show that the marginal likelihood of this BWM with the above choices admits a simple and computationally tractable expression. Using Lemma D.1, we take this as our nonconformity score, which gives a tractable method for computing our conformal $p$–values without MCMC sampling. Moreover, we describe our choice of $a$ and $b$ for the beta prime prior on $\tau^2$, which leads to a horseshoe prior on $\theta$ (Carvalho et al., 2010).

Define the empirical mean and empirical variance:

$$\bar{r}_n = \frac{1}{n}\sum_{i=1}^{n} r_i, \qquad s_n^2 = \frac{1}{n}\sum_{i=1}^{n}(r_i - \bar{r}_n)^2.$$

We first note that the model likelihood can be written as:

$$
\begin{aligned}
p(\mathbf{r}_{1:n} \mid \theta) &= (2\pi\sigma^2)^{-n/2}\exp\left(-\frac{1}{2\sigma^2}\sum_{i=1}^{n}(r_i - \theta)^2\right)\\
&= (2\pi\sigma^2)^{-n/2}\exp\left(-\frac{ns_n^2}{2\sigma^2}\right)\exp\left(-\frac{n(\theta - \bar{r}_n)^2}{2\sigma^2}\right)\\
&= (2\pi\sigma^2)^{-n/2}\exp\left(-\frac{ns_n^2}{2\sigma^2}\right)(2\pi\sigma^2/n)^{1/2}\,\varphi\left(\bar{r}_n; \theta, \frac{\sigma^2}{n}\right),
\end{aligned}
$$

where $\varphi(\cdot; \mu, v)$ denotes the $\mathcal{N}(\mu, v)$ density. Now, integrating out $\theta$ given $\tau^2$ gives:

$$
\begin{aligned}
p(\mathbf{r}_{1:n} \mid \tau^2) &= \int p(\mathbf{r}_{1:n} \mid \theta)\, p(\theta \mid \tau^2)\, d\theta\\
&= (2\pi\sigma^2)^{-n/2}\exp\left(-\frac{ns_n^2}{2\sigma^2}\right)(2\pi\sigma^2/n)^{1/2}\int \varphi\left(\bar{r}_n; \theta, \frac{\sigma^2}{n}\right)\varphi(\theta; 0, \gamma\tau^2)\, d\theta.
\end{aligned}
$$

The integral is the convolution of two Gaussians:

$$\int \varphi\left(\bar{r}_n; \theta, \frac{\sigma^2}{n}\right)\varphi(\theta; 0, \gamma\tau^2)\, d\theta = \varphi\left(\bar{r}_n; 0, \frac{\sigma^2}{n} + \gamma\tau^2\right).$$

Thus we have:

$$p(\mathbf{r}_{1:n} \mid \tau^2) = (2\pi\sigma^2)^{-n/2}\exp\left(-\frac{ns_n^2}{2\sigma^2}\right)(2\pi\sigma^2/n)^{1/2}\,\varphi\left(\bar{r}_n; 0, \frac{\sigma^2}{n} + \gamma\tau^2\right).$$

This is further simplified by plugging in $\gamma = \sigma^2/n$:

$$p(\mathbf{r}_{1:n} \mid \tau^2) = (2\pi\sigma^2)^{-n/2}\exp\left(-\frac{ns_n^2}{2\sigma^2}\right)(1+\tau^2)^{-1/2}\exp\left(-\frac{n\bar{r}_n^2}{2\sigma^2(1+\tau^2)}\right).$$

We can obtain an expression for the marginal likelihood by integrating $\tau^2$ out as follows:

$$
\begin{aligned}
p(\mathbf{r}_{1:n}) &= \int_0^\infty p(\mathbf{r}_{1:n} \mid \tau^2)\, f_{\tau^2}(\tau^2)\, d\tau^2 \\
&= (2\pi\sigma^2)^{-n/2} \exp\left(-\frac{ns_n^2}{2\sigma^2}\right) \frac{\Gamma(a+b)}{\Gamma(a)\Gamma(b)} \int_0^\infty t^{b-1}(1+t)^{-(a+b+1/2)} \exp\left(-\frac{n\bar{r}_n^2}{2\sigma^2(1+t)}\right) dt,
\end{aligned}
$$

This can be further simplified by performing a change of variables $u = 1/(1+t)$ and recognising the integral as a representation of Kummer's confluent hypergeometric function, $_1F_1$ (Slater, 1960):

$$
p(\mathbf{r}_{1:n}) = (2\pi\sigma^2)^{-n/2} \exp\left(-\frac{ns_n^2}{2\sigma^2}\right) \frac{\Gamma(a+1/2)\Gamma(a+b)}{\Gamma(a)\Gamma(a+b+1/2)}\, _1F_1\left(a+\frac{1}{2};\ a+b+\frac{1}{2};\ -\frac{n\bar{r}_n^2}{2\sigma^2}\right).
$$

Using Lemma D.1, we can therefore use the above as our nonconformity score:

$$
s(r_{n+1}, \mathbf{r}_{1:n}) = p(\mathbf{r}_{1:n}) = (2\pi\sigma^2)^{-n/2} \exp\left(-\frac{ns_n^2}{2\sigma^2}\right) \frac{\Gamma(a+1/2)\Gamma(a+b)}{\Gamma(a)\Gamma(a+b+1/2)}\, _1F_1\left(a+\frac{1}{2};\ a+b+\frac{1}{2};\ -\frac{n\bar{r}_n^2}{2\sigma^2}\right),
$$

Finally, we can drop the constants to obtain the general form of the **RoBAS–Full** nonconformity score:

$$
s(r_{n+1}, \mathbf{r}_{1:n}) = p(\mathbf{r}_{1:n}) = \exp\left(-\frac{ns_n^2}{2\sigma^2}\right) {}_1F_1\left(a+\frac{1}{2};\ a+b+\frac{1}{2};\ -\frac{n\bar{r}_n^2}{2\sigma^2}\right). \tag{30}
$$

**Choices for $a, b$.**  A natural choice for $a, b$, as we describe below, is $a = b = 1/2$.

Setting $a = b = 1/2$ in (29) gives:

$$
g_{\tau^2}(t) = \frac{\Gamma(1)}{\Gamma(1/2)\Gamma(1/2)}\, t^{-1/2}(1+t)^{-1} = \frac{1}{\pi}\, t^{-1/2}(1+t)^{-1}, \qquad t > 0.
$$

This corresponds exactly to a half–Cauchy prior on the scale $\tau$, which induces a horseshoe prior on $\theta$ Carvalho et al. (2010). This is a natural default in our context where the $r_i$ are the residuals of some predictive model. This is because it provides:

- *strong shrinkage near zero*: the density on $\tau$ has substantial mass near $0$, encouraging $\theta$ to be close to $0$ when the data support it;

- *very heavy tails*: large values of $\tau$ are not overly penalized, so large signals in $\theta$ are not over–shrunk. This provides us with the desirable behaviour noted in Theorem 3.1.

### E.4. Additional Details on BWM (11)

Here, we describe the nonconformity score function corresponding to BWM (11) as well as the expression for its prediction interval.

**Nonconformity score.**  Firstly, recall that BWM (11) is given by:

$$
\begin{aligned}
R \mid \theta &\sim \mathcal{N}\left(\theta, \sigma^2\right), \\
\theta &\sim \mathcal{N}\left(0, \tau^2\sigma^2\right), \\
1/\sigma^2 &\sim \text{Gamma}(a/2, b/2), \tag{31}
\end{aligned}
$$

where $\tau^2, a, b$ are fixed hyperparameters. This has the following residual–based, Bayes–assisted nonconformity score:

$$
\begin{aligned}
\tilde{s}\left(\mathbf{z}_{n+1}, \mathbf{z}_{1:n}\right) &= s\left(r_{n+1}, \mathbf{r}_{1:n}\right) \\
&= -p\left(r_{n+1} | \mathbf{r}_{1:n}\right) \\
&= -\frac{\Gamma\left(\frac{a_\sigma+1}{2}\right)}{\sqrt{a_\sigma \pi}\,\Gamma\left(\frac{a_\sigma}{2}\right)} \left(\frac{1}{\sqrt{\sigma_t^2}} \left(1 + \frac{1}{a_\sigma} \frac{(r_{n+1} - \mu_\theta)^2}{\sigma_t^2}\right)^{-(a_\sigma+1)/2}\right),
\end{aligned}
$$

where $\sigma_t^2 = b_\sigma \left(1 + \tau_\theta^2\right)/a_\sigma$ and

$$a_\sigma = a + n, \quad b_\sigma = b + \sum_{i=1}^n r_i^2 - \frac{\mu_\theta^2}{\tau_\theta^2}, \quad \mu_\theta = \left(\sum_{i=1}^n r_i\right)\tau_\theta^2, \quad \tau_\theta^2 = \left(\frac{1}{\tau^2} + n\right)^{-1}.$$

**Prediction interval.** Bersson & Hoff (2024) showed that the prediction set under (31) is an interval and can be computed *exactly* via simple order statistics. Define, for each $i \in \{1, \ldots, n\}$,

$$g(r_i) = \frac{2\left(\sum_{k=1}^n r_k\right)\left(\frac{1}{\tau^2} + n + 1\right)^{-1} - r_i}{1 - 2\left(\frac{1}{\tau^2} + n + 1\right)^{-1}}.$$

Let

$$\mathbf{v} = \left(r_1, \ldots, r_n, g(r_1), \ldots, g(r_n)\right)^\top \in \mathbb{R}^{2n}, \qquad k = \lfloor \alpha(n+1) \rfloor,$$

and write $v_{(1)} \le \cdots \le v_{(2n)}$ for the order statistics of $\mathbf{v}$. Then the resulting prediction interval, in the space of the residuals, is given by $\left[ v_{(k)}, \ v_{(2n-k+1)} \right]$. This is easily transformed back into output space by adding the point prediction used to define the residuals. i.e.,

$$\mathcal{C}_\alpha(\mathbf{x}_{n+1}; \mathbf{z}_{1:n}) = \left[ f(\mathbf{x}_{n+1}) + v_{(k)}, \ f(\mathbf{x}_{n+1}) + v_{(2n-k+1)} \right].$$

**Connection to our approach.** Bersson & Hoff (2024) also use the working model (31) for Bayes–assisted conformal prediction. They do this by modelling the conditional distribution of the response given covariates, i.e. $Y|\mathbf{X}$. In contrast, we use (31) as a working model for the residuals induced by a fixed predictor $f$, and define our Bayes–assisted nonconformity score through the resulting posterior predictive density on these residuals.

### E.5. Computational Complexity

Here, we describe the computational complexity of Algorithm 1 and compare it with the complexity of standard grid–search.

**Computational complexity of Algorithm 1.** The computational cost of Algorithm 1 is dominated by evaluations of the conformal $p$-value $\rho(y)$. A single evaluation requires computing the test score and comparing it to $n$ calibration scores, which costs $T_s(n) = \mathcal{O}(n)$. The one-dimensional optimization and root–finding procedures used to locate $y^\star$, $l$, and $u$ require $K(\varepsilon) = \mathcal{O}(\log(1/\varepsilon))$ function evaluations[4] to achieve endpoint accuracy $\varepsilon$, independently of $n$. Hence the overall time complexity per test point is $\mathcal{O}(T_s(n)\log(1/\varepsilon)) = \mathcal{O}(n\log(1/\varepsilon))$.

**Computational complexity of standard grid–search.** In contrast, a grid-based method that evaluates $\rho(y)$ on $G$ grid points has complexity $\mathcal{O}(Gn)$, and achieving accuracy $\varepsilon$ typically requires $G = \mathcal{O}(1/\varepsilon)$, yielding $\mathcal{O}(n/\varepsilon)$ time. Thus our grid–free procedure improves the dependence on the target precision from $1/\varepsilon$ to $\log(1/\varepsilon)$, while eliminating discretisation error.

---

[4]Standard 1D bracketing methods (e.g., bisection, Brent) shrink the bracket containing the solution geometrically, so the number of function evaluations needed to reach accuracy $\varepsilon$ is $\mathcal{O}(\log(1/\varepsilon))$ (Brent, 1973; Press et al., 2007).

# F. Additional Results

## F.1. Additional Datasets

*Table 2.* Interval widths for different nonconformity scores at different calibration sizes, $n_{\text{cal}}$, for the `VentricularVolume` dataset. Results show the mean $\pm$ standard err. over 300 trials on the in–distribution and out–of–distribution subsets of the dataset.

| Scores | IN | | | | OUT | | | |
|---|---|---|---|---|---|---|---|---|
| | $n_{\text{cal}} = 5$ | $n_{\text{cal}} = 10$ | $n_{\text{cal}} = 25$ | $n_{\text{cal}} = 50$ | $n_{\text{cal}} = 5$ | $n_{\text{cal}} = 10$ | $n_{\text{cal}} = 25$ | $n_{\text{cal}} = 50$ |
| **DTO** | $2.554 \pm 0.105$ | $3.640 \pm 0.136$ | $3.202 \pm 0.081$ | $2.533 \pm 0.039$ | $4.664 \pm 0.168$ | $5.894 \pm 0.184$ | $5.571 \pm 0.099$ | $4.688 \pm 0.051$ |
| **DTA** | $2.716 \pm 0.113$ | $3.774 \pm 0.141$ | $3.251 \pm 0.079$ | $2.533 \pm 0.039$ | $4.381 \pm 0.155$ | $5.209 \pm 0.168$ | $4.774 \pm 0.088$ | $3.912 \pm 0.047$ |
| **NNG** | $2.557 \pm 0.108$ | $3.671 \pm 0.139$ | $3.211 \pm 0.080$ | $2.528 \pm 0.039$ | $4.379 \pm 0.162$ | $5.478 \pm 0.177$ | $5.062 \pm 0.093$ | $4.147 \pm 0.049$ |
| **LOCAL** | $2.397 \pm 0.094$ | $3.318 \pm 0.119$ | $2.924 \pm 0.072$ | $2.321 \pm 0.033$ | $4.743 \pm 0.167$ | $6.013 \pm 0.180$ | $5.726 \pm 0.107$ | $4.751 \pm 0.049$ |
| **CQR** | $2.410 \pm 0.113$ | $3.497 \pm 0.141$ | $3.021 \pm 0.085$ | $2.278 \pm 0.044$ | $5.253 \pm 0.172$ | $6.521 \pm 0.182$ | $6.213 \pm 0.100$ | $5.296 \pm 0.050$ |
| **RoBAS–Full** | $2.553 \pm 0.107$ | $3.666 \pm 0.138$ | $3.217 \pm 0.080$ | $2.531 \pm 0.039$ | $4.459 \pm 0.163$ | $5.521 \pm 0.179$ | $4.944 \pm 0.093$ | $3.950 \pm 0.048$ |
| **RoBAS–EB** | $2.622 \pm 0.111$ | $3.715 \pm 0.140$ | $3.231 \pm 0.080$ | $2.538 \pm 0.039$ | $4.362 \pm 0.154$ | $5.368 \pm 0.172$ | $4.842 \pm 0.090$ | $3.922 \pm 0.047$ |

Tables 2 and 6 show the interval widths and coverage on the `VentricularVolume` dataset with the setup described in Appendix C.3. We find that overall our approach is competitive with the best performing methods on the in–distribution subsets. On the out–of–distribution subsets we find that our approach remains robust and attains the smallest widths alongside **DTA.**

## F.2. Different Calibration Sizes

Here, we ablate with larger calibration sizes. Specifically, we use the full calibration set described in Appendix C. We use the same setup as §4.

**Tabular datasets.** Figure 3 shows the interval widths for the tabular datasets. Like in §4, we observe that the widths of all standard and Bayes–assisted approaches expand with increasing covariate shift. The notable exception is **RoBAS**, which remains robust and performs similarly to **DTA**, achieving the smallest widths. On the other hand, at lower levels of covariate shift, where $f$ is a better fit for the calibration set, **RoBAS** performs competitively or matches the methods with the smallest widths.

**Image datasets.** Table 3 shows the interval widths for the image datasets. On both datasets, we perform competitively on the in–distribution subset, while outperforming all other approaches on the out–of–distribution subset alongside **DTA**.

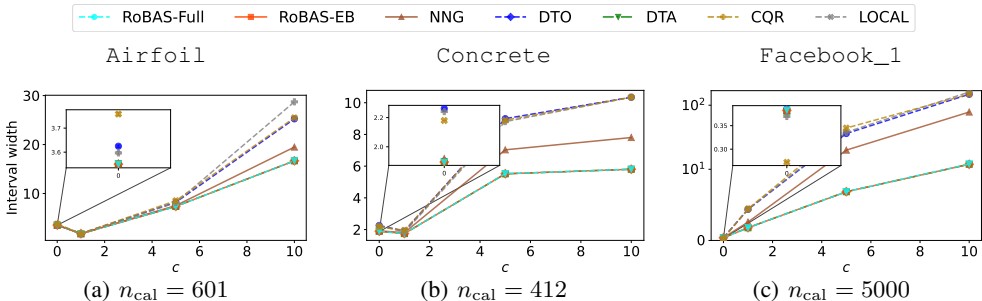

*Figure 3.* Interval width results for different nonconformity scores for different datasets with different levels of covariate shift. Results for standard calibration sizes are shown. Each of the plots also zooms in on the results for $c = 0$ to better illustrate the differences under lower levels of covariate shift. The `Facebook_1` results are displayed on a symlog scale to more clearly highlight the differences between the methods. Results show mean $\pm$ standard err. over 300 trials.

## F.3. Comparisons with CB and CBMA

Here, we compare **RoBAS–Full** and **RoBAS–EB** against **CB** and **CBMA** on the tabular datasets introduced in §4, following the experimental setup described in §C. Figures 4 and 5 show the interval widths and empirical coverage, respectively, for the various nonconformity scores.

*Table 3.* Interval widths for different nonconformity scores at standard calibration sizes for the `UTKFaces` and `VentricularVolume` datasets. Results show the mean $\pm$ standard err. over 300 trials on the in–distribution and out–of–distribution sets of the datasets.

| Scores | UTKFaces | | VentricularVolume | |
|---|---|---|---|---|
| | **IN** $n_{\text{cal}} = 2380$ | **OUT** $n_{\text{cal}} = 3387$ | **IN** $n_{\text{cal}} = 1031$ | **OUT** $n_{\text{cal}} = 1021$ |
| **DTO** | $3.021 \pm 0.002$ | $7.691 \pm 0.000$ | $2.345 \pm 0.003$ | $4.482 \pm 0.003$ |
| **DTA** | $3.029 \pm 0.002$ | $2.392 \pm 0.001$ | $2.337 \pm 0.003$ | $3.573 \pm 0.003$ |
| **NNG** | $3.025 \pm 0.002$ | $4.598 \pm 0.000$ | $2.343 \pm 0.003$ | $3.856 \pm 0.004$ |
| **LOCAL** | $3.005 \pm 0.002$ | $7.710 \pm 0.001$ | $2.180 \pm 0.003$ | $4.544 \pm 0.004$ |
| **CQR** | $3.049 \pm 0.001$ | $7.105 \pm 0.001$ | $2.075 \pm 0.004$ | $5.094 \pm 0.004$ |
| **RoBAS–Full** | $3.024 \pm 0.002$ | $2.392 \pm 0.001$ | $2.344 \pm 0.003$ | $3.572 \pm 0.003$ |
| **RoBAS–EB** | $3.022 \pm 0.002$ | $2.392 \pm 0.001$ | $2.345 \pm 0.003$ | $3.573 \pm 0.003$ |

As shown in Figure 4, both **RoBAS–EB** and **RoBAS–Full** consistently outperform **CB** and **CBMA** under distribution shift. In the absence of shift, our methods remain highly competitive or superior, with the single exception of the `Airfoil` dataset, where both **CB** and **CBMA** demonstrate strong performance. Furthermore, Figure 5 reveals that **CB** and **CBMA** frequently fail to achieve the target coverage rate – tending to either undercover or overcover – particularly as the level of distribution shift increases. This instability is likely a consequence of grid hyperparameter tuning.

Finally, we note that **CB** and **CBMA** exhibit significantly worse scalability than our approach, as they require averaging over typically high–dimensional posterior samples (see Table 4).

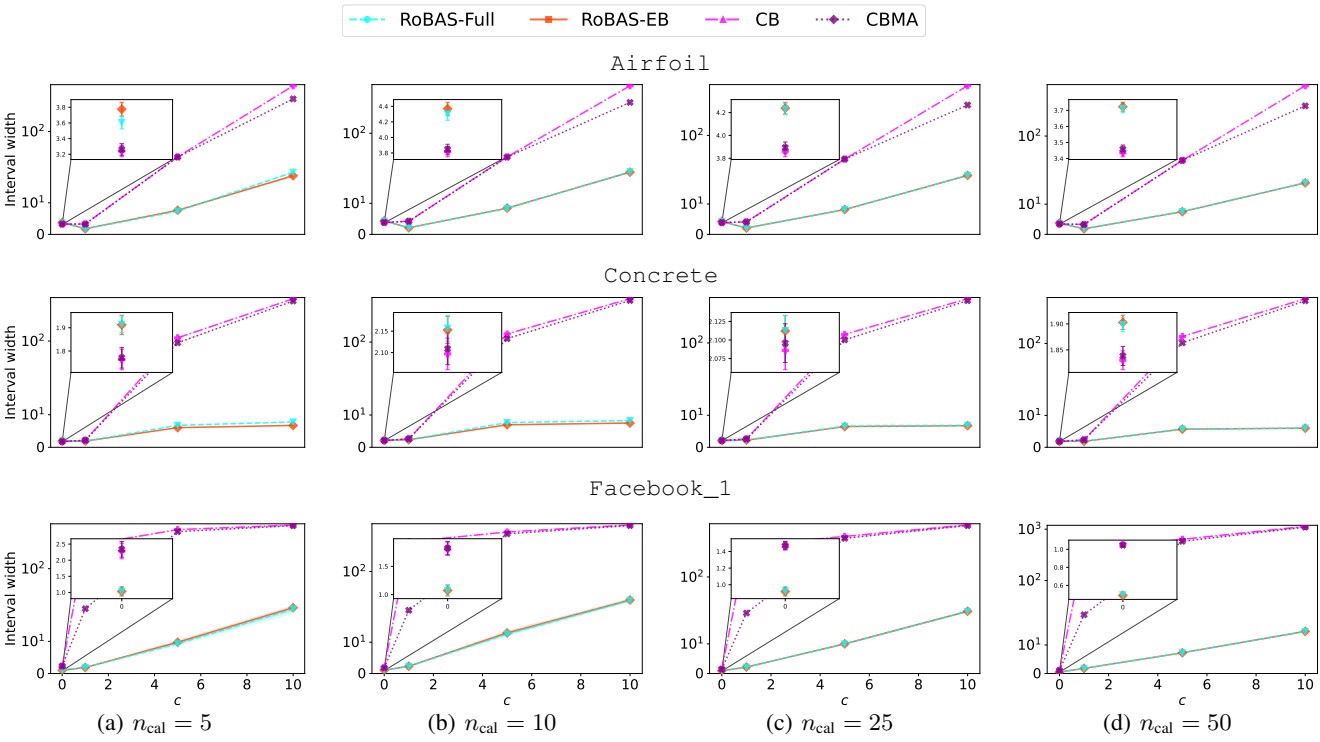

*Figure 4.* Interval width results for different nonconformity scores for different datasets with different levels of covariate shift. Results for different calibration sizes, $n_{\text{cal}}$ are shown. Each of the plots also zooms in on the results for $c = 0$ to better illustrate the differences under lower levels of covariate shift. The results are displayed on a symlog scale to more clearly highlight the differences between the methods. Results show mean $\pm$ standard err. over 300 trials.

## F.4. Computational Cost

Table 4 reports the computational cost for different nonconformity scores and different model choices on both our tabular and image datasets. We report performance as the mean throughput (trials per second) over 300 trials, where each trial

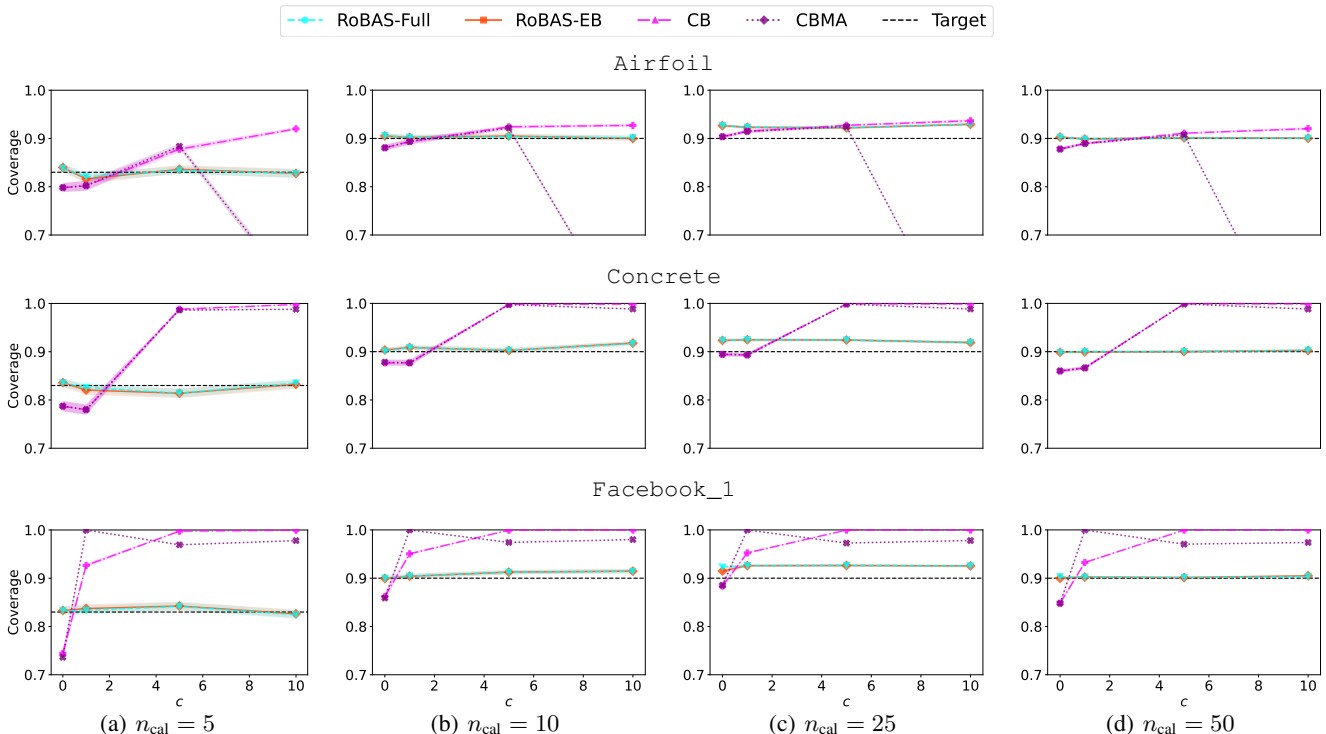

*Figure 5.* Coverage results for different nonconformity scores for different datasets with different levels of covariate shift. Results for different calibration sizes, $n_{cal}$ are shown. Results show mean $\pm$ standard err. over 300 trials.

corresponds to the computation of a single prediction interval.

We observe that **NNG**, **CQR**, and **LOCAL** incur the lowest computational costs across all datasets, primarily due to their closed–form expressions for their prediction intervals. In contrast, **CB** and **CBMA** have the highest computational cost.

*Table 4.* Computational cost of prediction interval computation for different nonconformity scores, measured in trials per second (over 300 trials). Each trial corresponds to the computation of a single prediction interval. Experiments use a calibration set size of $n_{cal} = 50$. For image datasets, results correspond to the in-distribution subsets; for tabular datasets, we report results for $c = 0$. "–" indicates that the experiments were not ran for this particular dataset and score.

| Scores | Facebook_1 | Airfoil | Concrete | VentricularVolume | UTKFaces |
|---|---|---|---|---|---|
| **DTO** | 4.32 | 5.37 | 5.62 | 15.92 | 16.85 |
| **DTA** | 3.98 | 5.29 | 5.25 | 12.63 | 13.08 |
| **NNG** | 7.39 | 11.31 | 12.79 | 3635.30 | 2607.25 |
| **LOCAL** | 4.11 | 42.84 | 45.74 | 22.41 | 16.74 |
| **CQR** | 4.40 | 41.70 | 46.65 | 22.50 | 18.70 |
| **CB** | 7.16 | 1.96 | 2.62 | – | – |
| **CBMA** | 7.25 | 1.96 | 2.62 | – | – |
| **RoBAS–Full** | 3.13 | 4.92 | 4.87 | 10.69 | 12.03 |
| **RoBAS–EB** | 4.02 | 5.20 | 5.35 | 12.69 | 13.81 |

## F.5. Coverage

**Synthetic dataset.** Figure 6 shows the coverage results for Figure 1.

**Tabular datasets.** Figures 7 and 8 show the coverage results for Figures 2 and 3 respectively.

**Image datasets.** Tables 5, 6, and 7 show the coverage results for Tables 1, 2 and 3 respectively.

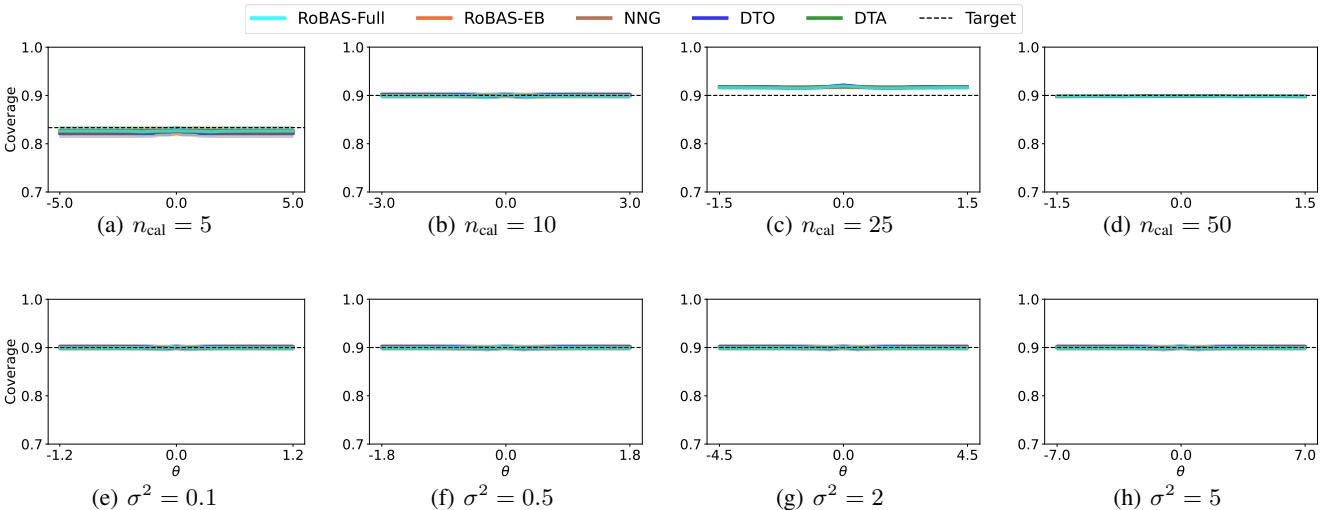

*Figure 6.* Coverage results for different nonconformity scores for data distributed as $\mathcal{N}(\theta, \sigma^2)$. **Top Row:** Results across different calibration sizes, $n_{\text{cal}}$, with $\sigma^2 = 1$. **Bottom Row:** Results across different noise levels, $\sigma^2$, with a fixed calibration size $n_{\text{cal}} = 10$. To make the differences between the different methods clear, the y–axis has been cut. Results show the mean $\pm$ standard err. over 300 trials.

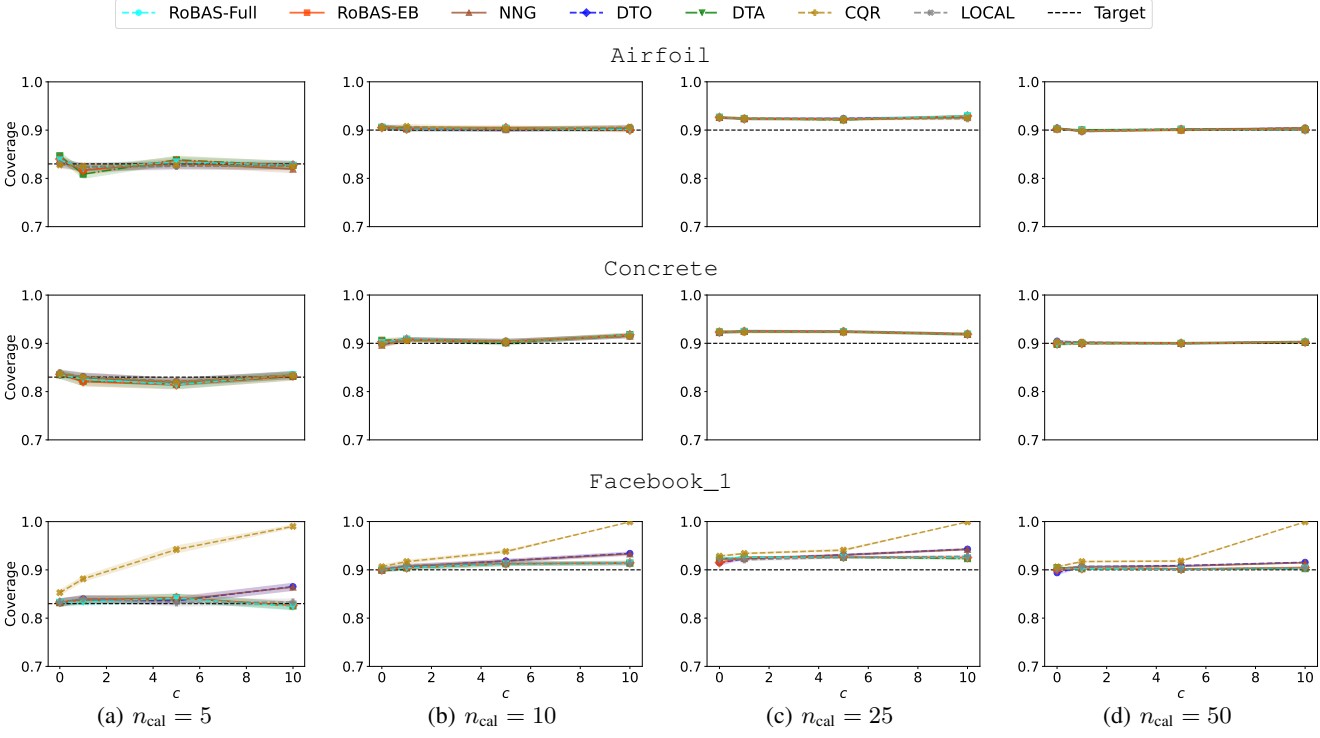

*Figure 7.* Coverage results for different nonconformity scores for different datasets with different levels of covariate shift. Results for different calibration sizes, $n_{\text{cal}}$ are shown. Results show mean $\pm$ standard err. over 300 trials.

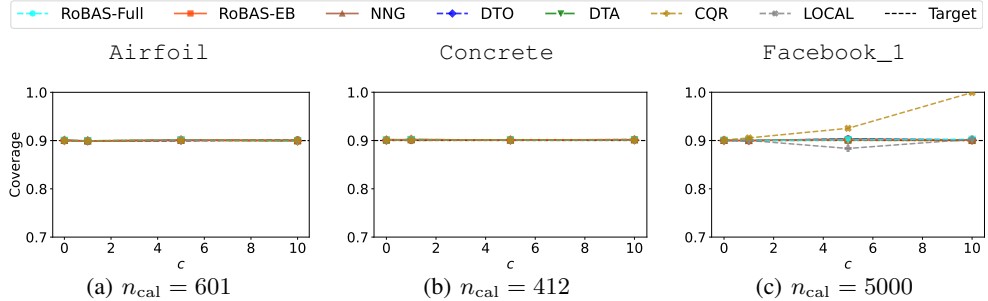

(a) $n_{cal} = 601$        (b) $n_{cal} = 412$        (c) $n_{cal} = 5000$

*Figure 8.* Coverage results for different nonconformity scores for different datasets with different levels of covariate shift. Results for standard calibration sizes are shown. Results show mean ± standard err. over 300 trials.

*Table 5.* Coverage for different nonconformity scores at different calibration sizes, $n_{cal}$, for the `UTKFaces` dataset. Results show the mean ± standard err. over 300 trials on the in–distribution and out–of–distribution subsets of the dataset. $\kappa$ denotes the target coverage level.

| Scores | IN | | | | OUT | | | |
|---|---|---|---|---|---|---|---|---|
| | $n_{cal} = 5$ $\kappa = 0.833$ | $n_{cal} = 10$ $\kappa = 0.900$ | $n_{cal} = 25$ $\kappa = 0.900$ | $n_{cal} = 50$ $\kappa = 0.900$ | $n_{cal} = 5$ $\kappa = 0.833$ | $n_{cal} = 10$ $\kappa = 0.900$ | $n_{cal} = 25$ $\kappa = 0.900$ | $n_{cal} = 50$ $\kappa = 0.900$ |
| DTO | $0.839 \pm 0.008$ | $0.912 \pm 0.004$ | $0.920 \pm 0.003$ | $0.905 \pm 0.002$ | $0.821 \pm 0.008$ | $0.908 \pm 0.005$ | $0.925 \pm 0.003$ | $0.900 \pm 0.002$ |
| DTA | $0.841 \pm 0.008$ | $0.907 \pm 0.005$ | $0.920 \pm 0.003$ | $0.903 \pm 0.002$ | $0.845 \pm 0.009$ | $0.912 \pm 0.005$ | $0.923 \pm 0.003$ | $0.902 \pm 0.002$ |
| NNG | $0.840 \pm 0.008$ | $0.910 \pm 0.005$ | $0.919 \pm 0.003$ | $0.904 \pm 0.002$ | $0.821 \pm 0.008$ | $0.907 \pm 0.005$ | $0.925 \pm 0.003$ | $0.900 \pm 0.002$ |
| LOCAL | $0.839 \pm 0.008$ | $0.913 \pm 0.004$ | $0.920 \pm 0.003$ | $0.904 \pm 0.002$ | $0.820 \pm 0.008$ | $0.909 \pm 0.005$ | $0.925 \pm 0.003$ | $0.901 \pm 0.002$ |
| CQR | $0.836 \pm 0.008$ | $0.916 \pm 0.004$ | $0.922 \pm 0.003$ | $0.905 \pm 0.002$ | $0.854 \pm 0.008$ | $0.936 \pm 0.004$ | $0.940 \pm 0.002$ | $0.922 \pm 0.001$ |
| RoBAS–Full | $0.842 \pm 0.008$ | $0.909 \pm 0.005$ | $0.919 \pm 0.003$ | $0.904 \pm 0.002$ | $0.847 \pm 0.009$ | $0.912 \pm 0.005$ | $0.923 \pm 0.003$ | $0.902 \pm 0.002$ |
| RoBAS–EB | $0.840 \pm 0.008$ | $0.908 \pm 0.005$ | $0.919 \pm 0.003$ | $0.904 \pm 0.002$ | $0.846 \pm 0.009$ | $0.912 \pm 0.005$ | $0.923 \pm 0.003$ | $0.902 \pm 0.002$ |

*Table 6.* Coverage for different nonconformity scores at different calibration sizes, $n_{cal}$, for the `VentricularVolume` dataset. Results show the mean ± standard err. over 300 trials on the in–distribution and out–of–distribution subsets of the dataset. $\kappa$ denotes the target coverage level.

| Scores | IN | | | | OUT | | | |
|---|---|---|---|---|---|---|---|---|
| | $n_{cal} = 5$ $\kappa = 0.833$ | $n_{cal} = 10$ $\kappa = 0.900$ | $n_{cal} = 25$ $\kappa = 0.900$ | $n_{cal} = 50$ $\kappa = 0.900$ | $n_{cal} = 5$ $\kappa = 0.833$ | $n_{cal} = 10$ $\kappa = 0.900$ | $n_{cal} = 25$ $\kappa = 0.900$ | $n_{cal} = 50$ $\kappa = 0.900$ |
| DTO | $0.834 \pm 0.008$ | $0.907 \pm 0.005$ | $0.920 \pm 0.003$ | $0.901 \pm 0.003$ | $0.837 \pm 0.008$ | $0.904 \pm 0.005$ | $0.923 \pm 0.003$ | $0.901 \pm 0.003$ |
| DTA | $0.831 \pm 0.009$ | $0.909 \pm 0.005$ | $0.922 \pm 0.003$ | $0.900 \pm 0.003$ | $0.845 \pm 0.008$ | $0.907 \pm 0.005$ | $0.924 \pm 0.003$ | $0.902 \pm 0.003$ |
| NNG | $0.836 \pm 0.008$ | $0.908 \pm 0.005$ | $0.921 \pm 0.003$ | $0.901 \pm 0.003$ | $0.839 \pm 0.008$ | $0.904 \pm 0.005$ | $0.924 \pm 0.003$ | $0.901 \pm 0.003$ |
| LOCAL | $0.835 \pm 0.008$ | $0.906 \pm 0.005$ | $0.918 \pm 0.003$ | $0.900 \pm 0.003$ | $0.839 \pm 0.008$ | $0.907 \pm 0.005$ | $0.924 \pm 0.003$ | $0.902 \pm 0.003$ |
| CQR | $0.833 \pm 0.007$ | $0.908 \pm 0.004$ | $0.917 \pm 0.003$ | $0.896 \pm 0.003$ | $0.829 \pm 0.009$ | $0.906 \pm 0.005$ | $0.923 \pm 0.003$ | $0.902 \pm 0.003$ |
| RoBAS–Full | $0.836 \pm 0.008$ | $0.909 \pm 0.005$ | $0.921 \pm 0.003$ | $0.901 \pm 0.003$ | $0.842 \pm 0.008$ | $0.905 \pm 0.005$ | $0.924 \pm 0.003$ | $0.902 \pm 0.003$ |
| RoBAS–EB | $0.833 \pm 0.008$ | $0.909 \pm 0.005$ | $0.922 \pm 0.003$ | $0.901 \pm 0.003$ | $0.844 \pm 0.008$ | $0.906 \pm 0.005$ | $0.924 \pm 0.003$ | $0.902 \pm 0.003$ |

*Table 7.* Coverage for different nonconformity scores at standard calibration sizes for the `UTKFaces` and `VentricularVolume` datasets. Results show the mean ± standard err. over 300 trials on the in–distribution and out–of–distribution sets of the datasets. The target coverage level here is 0.900.

| Scores | UTKFaces | | VentricularVolume | |
|---|---|---|---|---|
| | IN $n_{cal} = 2380$ | OUT $n_{cal} = 3387$ | IN $n_{cal} = 1031$ | OUT $n_{cal} = 1021$ |
| DTO | $0.900 \pm 0.001$ | $0.901 \pm 0.001$ | $0.900 \pm 0.001$ | $0.899 \pm 0.001$ |
| DTA | $0.900 \pm 0.001$ | $0.900 \pm 0.001$ | $0.900 \pm 0.001$ | $0.902 \pm 0.001$ |
| NNG | $0.900 \pm 0.001$ | $0.901 \pm 0.001$ | $0.900 \pm 0.001$ | $0.899 \pm 0.001$ |
| LOCAL | $0.900 \pm 0.001$ | $0.901 \pm 0.001$ | $0.900 \pm 0.001$ | $0.901 \pm 0.001$ |
| CQR | $0.902 \pm 0.001$ | $0.906 \pm 0.001$ | $0.899 \pm 0.001$ | $0.900 \pm 0.001$ |
| RoBAS–Full | $0.900 \pm 0.001$ | $0.900 \pm 0.001$ | $0.900 \pm 0.001$ | $0.902 \pm 0.001$ |
| RoBAS–EB | $0.900 \pm 0.001$ | $0.900 \pm 0.001$ | $0.900 \pm 0.001$ | $0.902 \pm 0.001$ |

