# OpenReview forum: "Robust Bayes-Assisted Conformal Prediction"
_ICML.cc/2026/Conference — ICML 2026 regular_

### Official Review · Reviewer_imNJ · 2026-02-28

**Soundness:** 2
**Presentation:** 2
**Significance:** 2
**Originality:** 2
**Overall Recommendation:** 4
**Confidence:** 2

**Summary:**

This paper addresses the degradation of prediction-set efficiency in Bayes–assisted conformal prediction when the prior is poorly aligned with the data. My main concerns are as follows.

**Compliance With Llm Reviewing Policy:**

Affirmed.

**Final Justification:**

As shown above, the author's rebuttal resolved my concern.

**Key Questions For Authors:**

How do we know the prior is misspecified?

**Limitations:**

yes

**Strengths And Weaknesses:**

Strength:

- The authors attempt to explore a general aspect of Bayes–assisted conformal prediction: robustness of efficiency to prior–data conflict.

Weaknesses:

- The third paragraph of the Introduction is a subjective assertion without supporting evidence. It is suggested to provide a visualization figure to support the argument.
- The fourth paragraph of the Introduction lacks motivation, i.e., why this approach is needed, rather than jumping straight to how we do it.
- The author did not provide reproducible code, making verification of the results difficult.

---

> ### Author Rebuttal · Authors · 2026-03-29
>
> We would like to thank the reviewer for their feedback on our submission. Below, we address the reviewer's weaknesses, answer their questions and also provide results for new experiments we have ran.
>
> # Weaknesses and Questions
>
> **Weaknesses 1-2:** To address your concerns, we have updated the third and fourth paragraphs of our introduction as below:
>
> _The validity [...] prior information in the model. In particular, if the data strongly disagrees with the prior, the resulting prediction sets can grow substantially **(e.g., see Figure 1 or [1, Figure 4 (b)-(c)])**. In such cases, the benefits of Bayesian modelling are effectively lost, and, to our knowledge, existing Bayes-assisted conformal methods do not explicitly target robustness to prior-data conflict._
>
> ***To address this**, we propose RoBAS: a novel Bayes--assisted nonconformity score motivated by a hierarchical Bayesian working model with heavy–tailed priors, and implemented in practice via a computationally tractable empirical Bayes instantiation. [...]*
>
>
> We believe that the revised introduction provides both evidence for the statements in the third paragraph and clearly motivates our method. Both prior literature [1, Figure 4] and Figure 1 in the main text, which we have moved to the first page to give it more visibility, support the observation that a misspecified prior leads to loss of efficiency in Bayes-assisted conformal prediction.
>
> **Weakness 3:** We agree that code availability is important. An anonymous repository for our code is provided here: https://anonymous.4open.science/r/RoBAS-D91D
>
> **Question 1:** In general, one cannot know ex ante whether the working prior is misspecified.
> This is precisely why we believe that Bayes-assisted conformal methods that effectively incorporate prior information while providing some form of built-in safety net against prior misspecification are valuable for the community.
>
> # New Experiments
>
> Below, we present experimental results for BWM (12), for which we make the choices discussed in Appendix E.3. In particular, we use a *horseshoe prior*. While specifying $\sigma^2$ for BWM (12) requires using additional data or prior information (see Appendix E.3), we overcome this issue by estimating $\sigma^2$ on the calibration data in such a way that allows us to still retain conformal validity. In particular, given calibration data $\\{r_i\\}\_{i=1}^n$ and a candidate test point $r_{n+1}$ we estimate $\sigma^2$ on the *augmented calibration set*, $\\{r_i\\}\_{i=1}^{n+1}$, using the unbiased estimate of the variance:
>
> $$\hat{\sigma}^2 = \frac{1}{n} \sum_{i=1}^{n+1} (r_i - \bar{r})^2, \quad \bar{r} = \frac{1}{n+1} \sum_{i=1}^{n+1} r_i.$$
>
> As $\hat{\sigma}^2$ is a symmetric function of all the $n+1$ points,  estimated on the augmented calibration set, we still retain the standard finite--sample marginal coverage guarantee of conformal prediction.
>
> **Synthetic experiments:**
> We follow the same experimental setup described in Section 4.1 of the manuscript. The approach based on BWM (12), described above, is referred to as **RoBAS-Full**, while the original approach from the paper is referred to as **RoBAS-EB**.
>
> Here, we present the results for calibration sizes $n_\text{cal}=5$ and $50$, corresponding to the first row of Figure 1 in the manuscript. Results for all calibration sizes and variance levels ($\sigma^2$) will be included in the revised version.
> - $n_\text{cal}=5$: https://ibb.co/mr5r7Yct
> - $n_\text{cal}=50$: https://ibb.co/LzNVc6MK
>
> As expected, these results show that BWM (12) and BWM (13) exhibit similar qualitative behaviour. However, there are two main differences:
> - BWM (12) outperforms BWM (13) when the residual means are $\approx 0$ (i.e., when the predictor is accurate or unbiased).
> - Compared to BWM (13), BWM (12) reverts to DTA for larger absolute values of the residuals (i.e., when the bias of the predictor is larger).
>
>
> **Real datasets:**
> We present results the airfoil dataset at calibration sizes $n_\text{cal}=5$ and $50$. Results for all other datasets and calibration sizes will be included in the revised manuscript.
>
> - $n_\text{cal}=5$: https://ibb.co/JR42Ztb9
> - $n_\text{cal}=50$: https://ibb.co/gbtsm4Rj
>
> These findings further reinforce our observations from the synthetic experiments. Specifically, both RoBAS-EB and RoBAS-Full remain efficient when there is little to no distribution shift, and they revert to the robust DTA baseline as the distribution shift increases. Furthermore, at $n_\text{cal}=5$, RoBAS-Full does not fully revert to the DTA baseline. This aligns with its behaviour in the synthetic experiments, where RoBAS-Full requires higher absolute residual values to revert to DTA.
>
> # References
> [1] Bersson, Elizabeth, and Peter D. Hoff. "Optimal conformal prediction for small areas." Journal of Survey Statistics and Methodology.

---

> > ### Author Rebuttal · Reviewer_imNJ · 2026-04-01
> >
> > Thank you for the author's reply. The reply resolved all my concerns, so I updated my score.

---

> > > ### Author Response · Authors · 2026-04-02
> > >
> > > Thank you for your feedback and for updating your score. We are glad to have resolved your concerns.

---

### Official Review · Reviewer_eK65 · 2026-03-05

**Soundness:** 3
**Presentation:** 4
**Significance:** 3
**Originality:** 3
**Overall Recommendation:** 5
**Confidence:** 3

**Summary:**

Bayes-assisted conformal prediction (CP) is a CP method that uses as the nonconformity score the log-likelihood a Bayesian working model (BWM) assigns to data. The paper extends Bayes-assisted CP with the aim of increasing its efficiency (decrease the width of the prediction intervals) in cases of distribution shift. The key components of Robust Bayes-Assisted Conformal Prediction (RoBAS) are a particular specification of the BWM and method for computation of the prediction intervals. The authors show mathematically and empirically that RoBAS adapts to various degrees of distribution shift.

**Compliance With Llm Reviewing Policy:**

Affirmed.

**Final Justification:**

In my initial review, I raised concerns regarding the paper's technical novelty. As part of the rebuttal, the authors provided additional theoretical and empirical results which have addressed these concerns. I have raised my score to reflect that including these additional results would strengthen the significance and originality of the paper's contributions.

**Key Questions For Authors:**

1. Could you please elaborate on the connections between the BWMs in Eqs. (12) and (13)? In particular,
- Does Theorem 3.1 apply also to the Bayes-assisted score function based on the BWM in Eq. (13)?
- If Eqs. (11), (12) and (13) are interpreted as generative models, how do they compare in terms of their accounts of the data-generating process?
- Could you please elaborate on the requirement that $g_{\tau^2}$ is regularly varying at infinity? If this requirement was not met, would this affect the validity of the proposed Empirical Bayesian approach?
2. In your experimental results, do all the tested methods satisfy the target marginal coverage guarantee?

**Limitations:**

I would have found it interesting to see more discussion of the limitations of the Empirical Bayesian approach as opposed to directly working with the BWM in Eq. (12).

**Strengths And Weaknesses:**

**Soundness:** The results appear sound, although I did not carefully check the math. The authors assess the behavior of RoBAS in a range of empirical settings and as a function of the degree of distribution shift.

**Clarity:** The paper is well-written.

**Significance and originality:** The paper tackles an important problem: how to maintain calibrated uncertainty quantification in the presence of distribution shift. CP methods like RoBAS that can cope with distribution shift without requiring substantial prior knowledge could offer a lot of practical value in settings where one encounters unknown degrees of both distribution shift and misspecification.

The paper's technical contributions seem to be principled but somewhat incremental: To the best of my understanding, the technical novelty consists of (1) the use of a heavy-tailed prior as the BWM, (2) an Empirical Bayesian approach to this model's estimation, and (3) a root-finding procedure to compute the prediction intervals.

---

> ### Author Rebuttal · Authors · 2026-03-29
>
> Thank you for your thorough review. Please see our responses below.
>
> # Questions
>
> **Q1.i.** Below we clarify how the different BWMs relate to one another (including BWM (11)).
>
> BWM (11) is the residual-based version of the model used in [1] and, as such, it represents a natural choice of informative BWM in our setup. However, as shown in Figure 1, BWM (11) leads to arbitrarily large prediction sets in the presence of prior-data conflict, motivating the search for BWMs that are both informative *and* robust to the latter.
>
> BWM (12) represents the first such choice. It is defined by mixing the variance of a Gaussian prior for the residual with a heavy-tailed (i.e., regularly varying at infinity) density.
> While remaining informative when residuals are close to zero, Theorem 3.1 shows that BWM (12) satisfies asymptotic robustness, with DTA-like behaviour in the presence of prior-data conflict.
>
> BWM (13), instead, defines a Normal-Normal model for the residuals, for which hyperparameters $(\sigma^2, \nu^2)$ are estimated using empirical Bayes (EB).
> Proposition 3.3 shows that BWM (13) with EB estimation directly interpolates between DTO and DTA based on the data. Note that the EB estimation step is crucial in obtaining robustness: specifying fixed hyperparameters $(\sigma^2, \nu^2)$ a priori would lead to a similar behaviour to BWM (11) in case of prior--data conflict. Theorem 3.1, as written, does not apply to BWM (13).
> In particular, as we are estimating $\sigma$, the condition $|\bar{r}_n^{(m)}|\to \infty$ used in Theorem 3.1 is not sufficient anymore to revert to DTA for BWM (13). Nonetheless, by additionally requiring that the sample variance of the residuals remains small compared to its sample mean, one obtains a robustness result similar to that of Theorem 3.1, as stated in the below Proposition. We will include this proposition and its proof in the revised manuscript.
>
> *Proposition. Fix $n \ge 1$. Let $(r\_{1:n}^{(m)})\_{m \ge 1}$, where $r\_{1:n}^{(m)}\in\mathbb R^n$, be a sequence of residuals such that $|\bar{r}\_n^{(m)}|\to\infty$ and $|\bar{r}\_n^{(m)}/\widehat\sigma^2(r\_{1:n}^{(m)})| \to \infty$ as $m \to \infty$. Then, for the nonconformity score in Eq. (14), we have, for every fixed $t \in \mathbb{R}$,
> $$s(\bar{r}\_n^{(m)} + t, r\_{1:n}^{(m)}) =  \longrightarrow |t| \qquad \text{as } m \to \infty.$$*
>
> As mentioned in the main text and Appendix E.3, while both BWMs (12) and (13) are robust to prior--data conflict, BWM (12) suffers from two main limitations compared to BWM (13): BWM (12) requires specifying $\sigma^2$ (and $g_{\tau^2}$), which may be difficult in certain settings, and the resulting scores involve integrating over $\tau^2$, increasing the computational overhead. As a result, we originally used BWM (12) simply as a motivating example of a fully-Bayesian BWM that leads to prediction sets that are both informative and robust. However, we have now been able to address the limitation of specifying $\sigma^2$ for BWM (12) and present the experiments with this BWM in our response to reviewer imNJ.
>
> **Q1.ii.** We would firstly like to clarify that BWMs (11)-(13) are specified as working models for the residuals $R = Y - f(\mathbf{X})$, where $f$ is fixed. Equivalently, since $Y = f(\mathbf{X}) + R$, they induce working models for $Y \mid \mathbf{X}$, but only through this residual decomposition. With this interpretation, we can make the following remarks about BWMs (11)--(13):
>
> - **BWM (11):** informative near zero residual means, but not robust because the prior scale is fixed.
>
> - **BWM (12):** uses a heavy-tailed prior scale, so it behaves like DTO when the residual means are close to 0 and reverts to DTA for large residual means.
>
> - **BWM (13):** uses EB-estimated hyperparameters to achieve the same qualitative behaviour as (12) in a data-adaptive way.
>
> **Q1.iii.** As BWM (13) is not an EB approximation to BWM (12) (please see our previous responses), the regular variation of $g_{\tau^2}$ does **not** affect the validity of BWM (13) and its EB approximation. Moreover, the validity of BWM (13) follows immediately from the standard exchangeability assumption of conformal prediction (see paragraphs 1-2 of Section 3.1).
>
>
> **Q2.** Yes, this is indeed the case - please see plots of coverage in Appendix F.5 and also our response to reviewer oRKW (Q1).
>
> # Limitations
>
> Firstly, we would like to mention that we discuss the limitations of our approach in Appendix B. Secondly, recall from our previous response that BWM (12) and (13) are distinct BWMs.
>
> A limitation of BWM (13) relative to BWM (12) is that its robustness is obtained through EB estimates, so it may be less reliable when the calibration set is very small and the residual variance is large. By contrast, BWM (12) achieves robustness directly through its heavy-tailed prior scale, at the cost of extra specification and computation.
>
>
> # References
> [1] Peter Hoff. "Bayes-optimal prediction with frequentist coverage control." Bernoulli.

---

> > ### Author Rebuttal · Reviewer_eK65 · 2026-04-01
> >
> > Thanks to the authors for the detailed rebuttal. The authors' rebuttal has clarified that part of the paper's contribution is the introduction of two distinct robust BWMs ((12) and (13)). As part of the rebuttal, the authors provide a theoretical robustness result for BWM (13), and empirical results on the behaviour of RoBAS with BWM (12). Revising the paper to include these results, which provide a more thorough set of guarantees and results on the behaviour of RoBAS under different BWMs, would to some extent address my concerns regarding the paper's technical novelty. I will raise my score accordingly.

---

> > > ### Author Response · Authors · 2026-04-02
> > >
> > > We sincerely thank you for your constructive feedback and for raising your score. We are glad that our rebuttal adequately addressed your concerns.

---

### Official Review · Reviewer_oRKW · 2026-03-10

**Soundness:** 3
**Presentation:** 3
**Significance:** 2
**Originality:** 2
**Overall Recommendation:** 4
**Confidence:** 2

**Summary:**

This paper introduces RoBAS, a robust Bayes-assisted nonconformity score for conformal prediction that addresses the sensitivity of prior Bayes-assisted methods to prior-data mismatch. By placing a heavy-tailed hierarchical prior only on the residuals of a fixed predictor and implementing it via a closed-form empirical Bayes shrinkage estimator, RoBAS automatically interpolates between the efficient Distance-To-Origin score (when residuals are near zero) and the robust Distance-To-Average score (when the residual mean is large). The authors derive theoretical asymptotic robustness (Theorem 3.1), provide an efficient grid-free interval computation algorithm, and evaluate on both synthetic data and real tabular/image regression tasks under covariate and distribution shift. They demonstrate competitive performance in the in-distribution regime and substantial gains (often the smallest widths) under shift, while remaining computationally lighter than full Bayesian baselines. The work offers a practical and theoretically grounded solution for conformal prediction in realistic deployment settings where training and calibration distributions may diverge.

**Compliance With Llm Reviewing Policy:**

Affirmed.

**Key Questions For Authors:**

1. The main text reports only interval widths; coverage rates appear solely in Appendix F.5. In the strong-shift + small-sample regime (especially n_cal = 5), does RoBAS maintain the nominal coverage level across all trials, or are there occasional violations?

2. Appendix F.2 shows only a brief selection of calibration sizes and omits full curves for n_cal = 100 or 500. Does the width advantage of RoBAS disappear at larger calibration sizes, or does it remain competitive?

3. The empirical-Bayes formulation completely removes the explicit choice of τ², yet the motivating hierarchical model (12) still implicitly requires specification of the tail index δ of g_τ². Does the value of δ meaningfully affect performance at small sample sizes?

4. The baseline comparison omits several recent conformal methods specifically engineered for covariate shift (e.g., weighted CP and adaptive conformal inference). How do these methods perform in width and coverage under the same covariate-shift intensities used in the paper?

**Limitations:**

No, the authors did not discuss the limitations and potential negative societal impact of their work.

**Strengths And Weaknesses:**

**Strengths:**
• Clean theoretical explanation of automatic DTO ↔ DTA switching via heavy tails and shrinkage.
• Closed-form score + grid-free intervals make the method immediately deployable and faster than MCMC-based Bayes-assisted baselines.
• Strong empirical gains precisely where prior methods fail (distribution shift + small calibration sets).

**Weaknesses:**
• Coverage is only systematically shown in the appendix (F.5); the main text focuses exclusively on width.
• Large-calibration behavior (n_cal ≥ 100) is unexplored, leaving open whether the adaptive advantage persists or vanishes as sample size grows.
• The heavy-tailed tail index δ is eliminated in the empirical-Bayes version but remains an implicit hyperparameter in the motivating model (12); its influence on small-sample performance is not analyzed.
• Comparison misses several recent conformal methods specifically designed for covariate shift.

---

> ### Author Rebuttal · Authors · 2026-03-29
>
> Thank you for your insightful review. Please see our responses below.
>
> # Questions
>
> **Q1:** We agree that coverage should be more visible. In our setup, the shift is between the predictor’s training distribution and the calibration/test distribution; the calibration and test data themselves are exchangeable, so the standard finite-sample marginal coverage guarantee applies to RoBAS and the other baseline methods  by construction. We therefore focused the main text on efficiency and reported coverage in Appendix F.5, but we will move a representative coverage result into the main paper.
>
> **Q2:** We first note that our desired behaviour holds theoretically for **any** calibration size -- see Theorem 3.1, and the discussion proceeding Proposition 3.3. Secondly, we did include an ablation with larger calibration sizes - standard calibration sizes > 100 - in Appendix F.2. There, we observe that:
> -  RoBAS still reverts to DTA with increasing levels of distribution shift between the data used to train the predictor $f$ and the calibration data.
> - RoBAS still achieves the desired shrinkage effect, retaining widths comparable, or even better, than DTO when there is no shift.
>
> We specifically decided to not include calibration sizes of 100, 500 etc. as the results were not qualitatively different from those observed at the standard calibration sizes that we ablated. We provide a representative example of this below by ablating with $n_\text{cal}=100$ for the *airfoil* dataset. Moreover, we will make the size of the standard calibration sets used in Appendix F.2 explicit in the manuscript. For convenience, we have also shown these sizes below.
>
> - Results for the *airfoil* dataset with $n_\text{cal}=100$: https://ibb.co/gZBNH4Rk
> - Standard calibration sizes for different datasets:
> | Dataset | Standard calibration size |
> | :--- | :--- |
> | Airfoil | 601 |
> | Concrete | 412 |
> | Facebook_1 | 16379 |
> | UTKFaces | 2445 |
> | VentricularVolume | 1061 |
>
>
> **Q3:**  Firstly, we would like to note:
> - BWM (12) and (13) are distinct BWMs:
>     - For BWM (12), we consider the score derived from the full Bayes posterior predictive.
>     - For BWM (13), we consider an empirical Bayes posterior predictive.
>     - The empirical Bayes approach of Section 3.3 is therefore is not based on model (12), which is why the exponent $\delta$ does not appear there.
>
> - In both cases, we achieve our desired behaviour: robustness in the case of prior misspecification; but the empirical Bayes approach is computationally more appealing, which is why we preferred the empirical Bayes approach in the experiments.
>
> We will make sure these differences are clear in the revised version of the manuscript.
>
>
> Secondly, while we did not experiment with BWM (12) in the uploaded manuscript as a result of the difficulty in specifying $g_{\tau^2}$, $\sigma^2$ and $\tau^2$ (as well as computational efficiency) (see also first paragraph of Section 3.3, and Appendix 3.3), we have now been able to
> address some of these limitations and
> obtain the results for BWM (12). Please see our response to reviewer imNJ for these results and their discussion.
>
> **Q4:** We intentionally decided not to cover these baselines because they work in a different setting to ours. Indeed, their setup assumes covariate shift between the calibration and test data, which we do not as explained in the answer to your question about coverage.
>
> We have, however, now compared our approach with a more recent Bayes--assisted baseline  (CBMA, [1]). We present representative results for this on the *airfoil* dataset at $n_\text{cal}=5, 50$ below:
> -  $n_\text{cal}=5$: https://ibb.co/gLKWyc8D
> -  $n_\text{cal}=50$: https://ibb.co/ds2gQnMC
>
> These results further reinforce that our approach is more robust than previous Bayes--assisted approaches.  Indeed, our approach attains stable widths  comparable to the robust DTA baseline with increasing distribution shift, while other approaches all increase in width. On the other hand, when there exists no shift, we attain comparable, or smaller, widths than all other approaches. We will include this baseline in the final version of the manuscript.
>
> # Limitations
>
> We have described the limitations of our approach and directions for future work in Appendix B. We will revise the manuscript to summarise this in the main text. Thank you for pointing out potentially negative societal impacts - we will update our Impact Statement to include these.

---

### Official Review · Reviewer_Wqmk · 2026-03-10

**Soundness:** 3
**Presentation:** 3
**Significance:** 3
**Originality:** 4
**Overall Recommendation:** 5
**Confidence:** 3

**Summary:**

The paper introduces RoBAS, a method of conformalizing (somewhat) Bayesian prediction algorithms. A good description is given in Section 3: RoBAS (i) retains the efficiency benefits of Bayes–assisted procedures, (ii) remains robust when based on inaccurate prior information, (iii) is computationally efficient. The Bayesian element of the proposed procedures only covers the model for the residuals, and the proposed algorithm interpolates nicely between DTO and DTA.

**Compliance With Llm Reviewing Policy:**

Affirmed.

**Final Justification:**

The authors addressed all my concerns, and I have raised my score.

**Key Questions For Authors:**

1. Your hierarchical BWM (12) for the residuals appears very inflexible, since $\sigma$ is a fixed hyperparameter. Is this because conformal p-values only depend on the order of the nonconformity scores and not on there precise values? If so, it would be good to explain it.
2. There are some problems with Theorem 3.1. You have both $\bar r_n^{(m)}\to\infty$ and $\left|\bar r_n^{(m)}\right|\to\infty$. Remove the contradiction by choosing one of those; drop the other. Or did you in fact mean $m\to\infty$ when you wrote $\left|\bar r_n^{(m)}\right|\to\infty$? (It appears likely.) Besides, understanding the order of quantifiers requires more than pass over the statement; making the order "for any $n$ and $r$ there exists $h$ such that for any $t$" explicit would make it easier to read. (Or is $n$ allowed to depend on $m$?)
3. You have three Bayesian models for the residuals, (11), (12), and (13), and different mathematical statements and computational experiments make different choices of the models. Is Theorem 3.1 true for (13) and is Proposition 3.3 true for (12)? And why don't you use (12) (which you introduce like your main choice) in Section 4?
4. It appears that in Figure 2 the ranking of different algorithms at $c=0$ is important; probably it is here that DTO shines. Was it your deliberate decision not to include this ranking? (Now it's impossible to see.)

**Limitations:**

The prior in (12) seems to be chosen for its mathematical convenience and is not supposed to reflect Bayesian's beliefs.

**Strengths And Weaknesses:**

There are interesting theoretical results, especially Proposition 3.3; the latter shows how the proposed algorithm interpolates between DTO and DTA via shrinkage. Theorem 3.1 is an asymptotic result (imperfectly stated, as explained below), and Proposition 3.2 shows how DTA fits this paper's Bayesian setting.

There are interesting simulation and empirical studies demonstrating advantages of the proposed algorithms (especially RoBAS).

The presentation is mostly good, but there are some lapses. Nice discussion of DTO and DTA scores. Theorem 2.1 is from Hoff (2023), but it's a big paper with 6 theorems; it would have been useful for me as reader to have an exact reference to the right theorem in that paper (I wanted to check how mild the "mild regularity conditions" are). Proposition 3.2 is somewhat informal, involving an improper prior. The Appendix Table of Contents is very helpful.

---

> ### Author Rebuttal · Authors · 2026-03-29
>
> Thank you for your thorough review. Please see our responses below.
>
> # Strengths & Weaknesses
>
> > Theorem 2.1 is from Hoff (2023), but it's a big paper with 6 theorems; it would have been useful for me as reader to have an exact reference [...]
>
> Theorem 2.1 is paraphrased from Theorem 4.1 in [1] (see also the discussion in the proceeding paragraphs of Theorem 4.1).
>
> # Questions
>
> **Q1:** For BWM (12), $\sigma^2$ is a fixed hyperparameter that needs to be specified (e.g. from prior information or validation data). This is not related to the ordering of the nonconformity scores - $\sigma^2$ is just a hyperparameter of the BWM.
>
> While the requirement to specify $\sigma^2$ from validation data or prior information is a limitation (see also Appendix E.3), we have now addressed this limitation with experiments using BWM (12) where $\sigma^2$ is estimated *only from calibration data.* See our response to reviewer imNJ for the results and their discussion.
>
> **Q2:** Thank you for pointing this out. The condition on the mean of the residuals is indeed $|\bar{r}_n^{(m)}| \to \infty$ as $m \to \infty$ and with $n$ fixed. We have revised the theorem statement to include these clarifications:
>
> *Theorem 3.1. Fix $n \ge 1$. Let $(r\_{1:n}^{(m)})\_{m \ge 1}$, where $r_{1:n}^{(m)}\in\mathbb R^n$, be a sequence of residuals such that $|\bar{r}_n^{(m)}| \to \infty$ as $m \to \infty$. Under BWM (12), for every fixed $t \in \mathbb{R}$,
> $$s(\bar{r}\_n^{(m)} + t, r\_{1:n}^{(m)}) = -p(\bar{r}\_n^{(m)} + t \mid r\_{1:n}^{(m)}) \longrightarrow h\_n(|t|) \qquad \text{as } m \to \infty,$$
> where $h_n$ is the strictly monotone increasing function
> $$h_n(u) = -\frac{1}{\sqrt{2\pi\sigma^2(1 + 1/n)}} \exp\left( -\frac{u^2}{2\sigma^2(1 + 1/n)} \right), \qquad u \ge 0.$$*
>
> **Q3:** Please see our response to reviewer eK65 (Q1.i).
>
> **Q4:** To more clearly show the ranking at $c=0$ we will update the manuscript with a section in the Appendix which zooms in on the differences at $c=0$. An example of the differences for the *airfoil* dataset at $n_\text{cal}=5, 50$ is given below:
> - $n_\text{cal}=5$: https://ibb.co/tMvm9LKW
> - $n_\text{cal}=50$: https://ibb.co/7t0cbw3s
>
> As you correctly point out, the links above more clearly shows that at $c=0$, DTO indeed achieves the smallest width/is within standard error of the method that achieves the smallest width. This is intuitive: because there is no distribution shift between the training data for $f$ and the calibration/test data, we expect the residuals of $f$ to be close to zero and DTO to serve as an informative nonconformity score.
>
> **Q5:** We would like to stress that while our choice of prior does simplify the final expression for the Bayes--assisted nonconformity corresponding to (12), it also reflects our prior beliefs. Indeed, the horseshoe prior [3] places large mass at residuals near zero while maintaining heavy tails for large (positive or negative) residuals. This prior accurately reflects our desired behaviour - that is DTO-like behaviour when the residuals are centred near zero and DTA-like behaviour when the residuals increasingly deviate from zero - and therefore reflects our prior beliefs. Please also refer to to Appendix E.3 for details on the choice of horseshoe prior for BWM (12).
>
> # References
> [1] Peter Hoff. "Bayes-optimal prediction with frequentist coverage control." Bernoulli.
> [2] Bersson, Elizabeth, and Peter D. Hoff. "Optimal conformal prediction for small areas." Journal of Survey Statistics and Methodology.
> [3] Carvalho, Carlos M., et al.“The Horseshoe Estimator for Sparse
> Signals.” Biometrika.

---

> > ### Author Rebuttal · Reviewer_Wqmk · 2026-04-03
> >
> > Thank you for your thoughtful rebuttal. I will update my score.

---

> > > ### Author Response · Authors · 2026-04-03
> > >
> > > We thank you for your constructive and thoughtful feedback and for raising your score. We are glad that our rebuttal adequately addressed your concerns.

---

### Decision · Program_Chairs · 2026-04-30

**Decision:**

Accept (regular)

**Comment:**

This paper studies the issue in Bayesian-assisted conformal prediction where mismatch between the prior and observed data leads to overly large prediction sets. To address this, the authors propose a new nonconformity score that adapts to the reliability of prior information. The method achieves smaller, more efficient prediction sets when the prior is reliable, while remaining robust when the prior is inaccurate. All reviewers acknowledge the novelty and effectiveness of the approach, making it a meaningful contribution to conformal prediction through improved utilization of prior information.